# Zoonotic *Streptococcus* imports glucose to inhibit stringent response and promote growth during meningitis

Chen Yuan[1], Karthik Hullahalli[2,3], Hao Huang[1], Siqi Zhao[1], Wenqing Wang[1], Xingyu Tian[1], Xin Li [1], Linya Xia[1], Yuchang Wang[1], Fei Pan[1], Ying Liang[1], Yurui Xie[1], Yue Li[1], Hongjie Fan [1,4], Matthew K. Waldor [2] & Zhe Ma [1,4] ✉

Proliferation of the emerging zoonotic pathogen *Streptococcus equi* subsp. *zooepidemicus* in the meninges is linked to mortality in pigs and morbidity in humans. The mechanisms underlying the remarkable capacity of hypervirulent *S. zooepidemicus* to proliferate in cerebrospinal fluid (CSF) are largely undefined. Here, using genetically barcoded *S. zooepidemicus*, we found that following systemic infection of mice, only ~1–10 *S. zooepidemicus* clones invade the meninges where they subsequently replicate ~$10^7$-fold. Subsequent transposon insertion sequencing experiments, plus validation work with bacterial mannose phosphotransferase system ($PTS_{man}$)-defective strains, identified the $PTS_{man}$, which imports glucose, as essential for *S. zooepidemicus* proliferation in CSF. The *S. zooepidemicus* $PTS_{man}$ promoter confers species-specific constitutive transcription of $PTS_{man}$, enabling glucose acquisition at low glucose concentrations and limiting activation of the stringent response, leading to pathogen replication in CSF. Our findings reveal how the rewiring of $PTS_{man}$ in the control of *S. zooepidemicus* metabolism enables this pathogen to adapt to and replicate in CSF during meningitis.

Bacterial meningitis occurs when bacteria invade the subarachnoid space, which contains cerebrospinal fluid (CSF) and blood vessels. Proliferation of bacteria in the CSF and the ensuing inflammation can also damage the brain parenchyma[1]. Bacterial meningitis results in high mortality and morbidity that often includes severe neurological sequelae[2,3]. Several *Streptococcus* species cause bacterial meningitis, with *Streptococcus pneumoniae* being predominant in adult community-acquired meningitis[1] and *Streptococcus agalactiae* (group B *Streptococcus* (GBS)) responsible for ~35% of early-onset neonatal meningitis[4]. Recently, the swine industry in North America and Europe has experienced an emerging pandemic of meningitis caused by *Streptococcus equi* subsp. *zooepidemicus*[5,6], with infected pigs showing high *S. zooepidemicus* colony-forming units (CFU; >$10^8$ ml$^{-1}$) within the CSF[7].

Notably, this pathogen has also been implicated in numerous human lethal meningitis cases globally[8,9]. Between 2021 and 2022, Italy experienced an *S. zooepidemicus* outbreak with 37 clinical cases, of which 5 patients died from meningitis[10].

To invade the central nervous system (CNS) from blood, pathogens must first penetrate the blood–CSF/brain barrier[11,12]. Although penetration of these barriers is challenging, pathogens may benefit from entering this immune-privileged niche. Contemporary studies into bacterial CNS invasion have primarily focused on elucidating strategies used by bacteria to penetrate the blood–CSF/brain barrier. For example, *S. zooepidemicus* strains in the ST194 clade encode factors such as *S. zooepidemicus* M protein (SzM) and BifA that facilitate the pathogen's traversal of the blood–brain barrier (BBB)[13,14]. By

[1]Ministry of Agriculture Key Laboratory of Animal Bacteriology, the International Joint Laboratory of Animal Health and Food Safety, and College of Veterinary Medicine, Nanjing Agricultural University, Nanjing, China. [2]Howard Hughes Medical Institute, Brigham and Women's Hospital Division of Infectious Diseases, and Department of Microbiology, Harvard Medical School, Boston, MA, USA. [3]Department of Microbiology and Immunology, Stritch School of Medicine, Loyola University Chicago, Maywood, IL, USA. [4]Jiangsu Co-innovation Center for Prevention and Control of Important Animal Infectious Diseases and Zoonoses, Yangzhou, China. ✉e-mail: mazhe@njau.edu.cn

contrast, there is a relative paucity of research addressing the adaptive responses of pathogens after CNS invasion, particularly within the CSF milieu. As the nutrients required to support bacterial replication in the CSF, such as glucose, can drop substantially during infection[15], to proliferate and reach high CFU levels in the later stages of infection, bacteria must adapt to nutrient limitation.

Glucose is the preferred carbon source for streptococci and is imported via the phosphoenolpyruvate (PEP)–carbohydrate phosphotransferase system (PTS), which also serves as a signalling mechanism by sensing the environmental carbohydrate availability[16]. In conditions in which glucose is scarce and insufficient for PTS-mediated import, the repression of alternative carbon source transporters is alleviated[17]. Growth in the CSF may also induce the bacterial stringent response, which is a broadly conserved bacterial stress response that controls adaptation to nutrient deprivation and coincides with the slowing of bacterial growth[18,19].

Using a mouse model, we investigated the mechanisms underlying the ability of *S. zooepidemicus* to adapt to and proliferate within the CSF. We show that the *S. zooepidemicus* mannose phosphotransferase system ($PTS_{man}$) is essential for its proliferation in the CSF, enabling continuous glucose uptake and preventing activation of the stringent response in a glucose-deprived milieu. Mechanistically, glucose acquisition in low-glucose CSF is due to *S. zooepidemicus*-specific polymorphisms within the $PTS_{man}$ promoter that render $PTS_{man}$ resistant to repression by stringent response-associated transcriptional repressors. Collectively, our findings unveil the molecular and genetic bases for *S. zooepidemicus* proliferation in CSF and suggest that targeting $PTS_{man}$ may represent a strategy to mitigate *S. zooepidemicus*-associated neurological damage.

## Results

### *S. zooepidemicus* accumulates to high CFU burden in CSF of infected mice

We first defined the kinetics of *S. zooepidemicus* accumulation in various tissues throughout the body following an intravenous (i.v.) challenge with $10^6$ CFU via the tail vein. At early time points (3–12 h post-inoculation (hpi)), the brain had a lower *S. zooepidemicus* burden compared with other organs. However, by 30 hpi, the *S. zooepidemicus* burden in the brain exceeded that of other organs, and by the time animals became moribund, the *S. zooepidemicus* brain burden was 10–100× greater than in other organs (Fig. 1a). Bioluminescence imaging of a luciferase-expressing strain of *S. zooepidemicus* ($SEZ_{luc}$) confirmed that the brain was the dominant site of *S. zooepidemicus* accumulation in moribund animals (Fig. 1b and Extended Data Fig. 1b). The brain also remained the predominant site of *S. zooepidemicus* accumulation when different doses were administered intravenously (Extended Data Fig. 1c,d). We note that perfusion did not impact CFU counts (Extended Data Fig. 1a), suggesting that the high pathogen burden is not due to intravascular *S. zooepidemicus*. In addition, the CFU counts in the brain of moribund mice after *S. agalactiae* or *S. pneumoniae* infection were lower than $10^8$ CFU and similar to those of other organs in those animals (Extended Data Fig. 1e,f). The rapid increase in the *S. zooepidemicus* burden in the brain suggests that the pathogen possesses specific strategies that facilitate its expansion in the CNS. Immunostaining of brain sections from moribund mice using monoclonal antibodies that recognize the SzM[20] revealed that *S. zooepidemicus* was primarily localized to CSF within the ventricles and not in the brain parenchyma (Fig. 1c). Remarkably, in most mice (8 out of 11), the bacterial burden in CSF exceeded $10^8$ CFU ml$^{-1}$; 6 out of 8 mice had $10^9$–$10^{10}$ CFU ml$^{-1}$ in the CSF (Fig. 1d). Together, these observations suggest that *S. zooepidemicus* has an exceptional capacity to accumulate in mouse CSF.

### Analysis of *S. zooepidemicus* infection dynamics with barcoded bacteria

Observations from CFU counting suggest that *S. zooepidemicus* can replicate locally in the brain. However, whether a high CNS burden

stems from in situ expansion or continued influx from other tissues remained unclear. To distinguish these possibilities and more broadly define the population dynamics of *S. zooepidemicus* CNS infection, we used barcoded isogenic *S. zooepidemicus* ($SEZ_{STAMP}$ library) coupled with the STAMPR analytic pipeline[21] to track individual *S. zooepidemicus* clones during infection[21,22]. The $SEZ_{STAMP}$ library was injected intravenously via the tail vein at a dose of $5 × 10^6$ CFU per mouse. Bacteria were collected from organs following cardiac perfusion at 6 hpi, 18 hpi or 36 hpi. Founding population (FP) values were relatively low throughout the infection, especially at 36 hpi (approximately <$10^2$), suggesting that there are notable host barriers that impede *S. zooepidemicus* access to and/or replication within these organs (Fig. 2a–c). Declining FP values over time suggest persistent host-imposed restrictions during infection. At 6 hpi, similar FP and CFU values within the same organ suggested minimal bacterial replication at this early stage of infection. The brain had the lowest number of founders (Ns ~$10^2$), suggesting that the BBB effectively impedes the traversal of *S. zooepidemicus* from the bloodstream to the brain (Fig. 2a)[14]. Although the number of founders in the brain was even lower at 36 hpi (Ns ~10), the CFU at that point was ~7 orders of magnitude higher (~$10^8$), suggesting that the 10 founders had undergone enormous replication.

To quantify the similarity between *S. zooepidemicus* populations from different organ samples and infer potential patterns of pathogen spread, we used a metric of genetic distance (GD, Cavalli-Sforza chord distance) derived from comparisons of the frequencies of barcodes in the bacterial populations from pairs of organs[21]. The GD is high for dissimilar samples and low for similar ones. As blood is the primary source and exchange route post-injection, we first assessed its GD with other organs during infection. The GDs between the blood and brain samples were relatively high, particularly at 6 hpi, indicating that the bacterial populations in most brains were initially dissimilar to those of the blood (high GD) (Fig. 2d and Extended Data Fig. 2e). Although there was a trend towards increasing similarity of the populations in the blood and brain during the course of the infection (Fig. 2d), the dominant clones in the brain remained largely distinct from those in the blood in most mice (Fig. 2e and Extended Data Fig. 2f,g). Examination of barcode distributions in individual mice showed that dominant brain clones were usually distinct from those in systemic sites (Fig. 2e and Extended Data Fig. 2f,g), indicating that these clones expanded in the brain in situ and were not derived from sources outside the brain. Unlike the brain, other tissues were more likely to share clones with the blood throughout the course of infection (Fig. 2d,e and Extended Data Fig. 2f,g). Unlike the brain, bacterial expansion in other organs more readily reflects systemic dissemination, with *S. zooepidemicus* spreading more readily between peripheral tissues than into the CNS. Together, the low *S. zooepidemicus* FP in the brain coupled with reduced similarity of the brain population with other tissues highlights that the brain and its CSF represent a privileged and specific compartment for *S. zooepidemicus* replication. Furthermore, by showing that *S. zooepidemicus* replicates massively within the CNS, these findings support the overarching hypothesis that *S. zooepidemicus* possesses specific mechanisms to support its proliferation in the CSF.

### Identifying *S. zooepidemicus* genes that promote its proliferation in CSF

During bacterial meningitis, nutrient concentrations in the CSF become limiting[23]. *S. zooepidemicus* must possess specific strategies to adapt to and grow in changing nutrient conditions. We performed a transposon insertion sequencing (Tn-seq) screen to define genetic requirements for *S. zooepidemicus* growth in the brain. We constructed a complex Himar transposon insertion library (referred to as $SEZ_{Tn}$) containing ~90,000 unique insertions, representing ~62% of all possible insertion sites (Extended Data Fig. 3a). To circumvent the stringent barriers presented by the bloodstream and BBB, which prevent pathogen access to the brain, we used a digital mouse stereotaxic instrument for direct

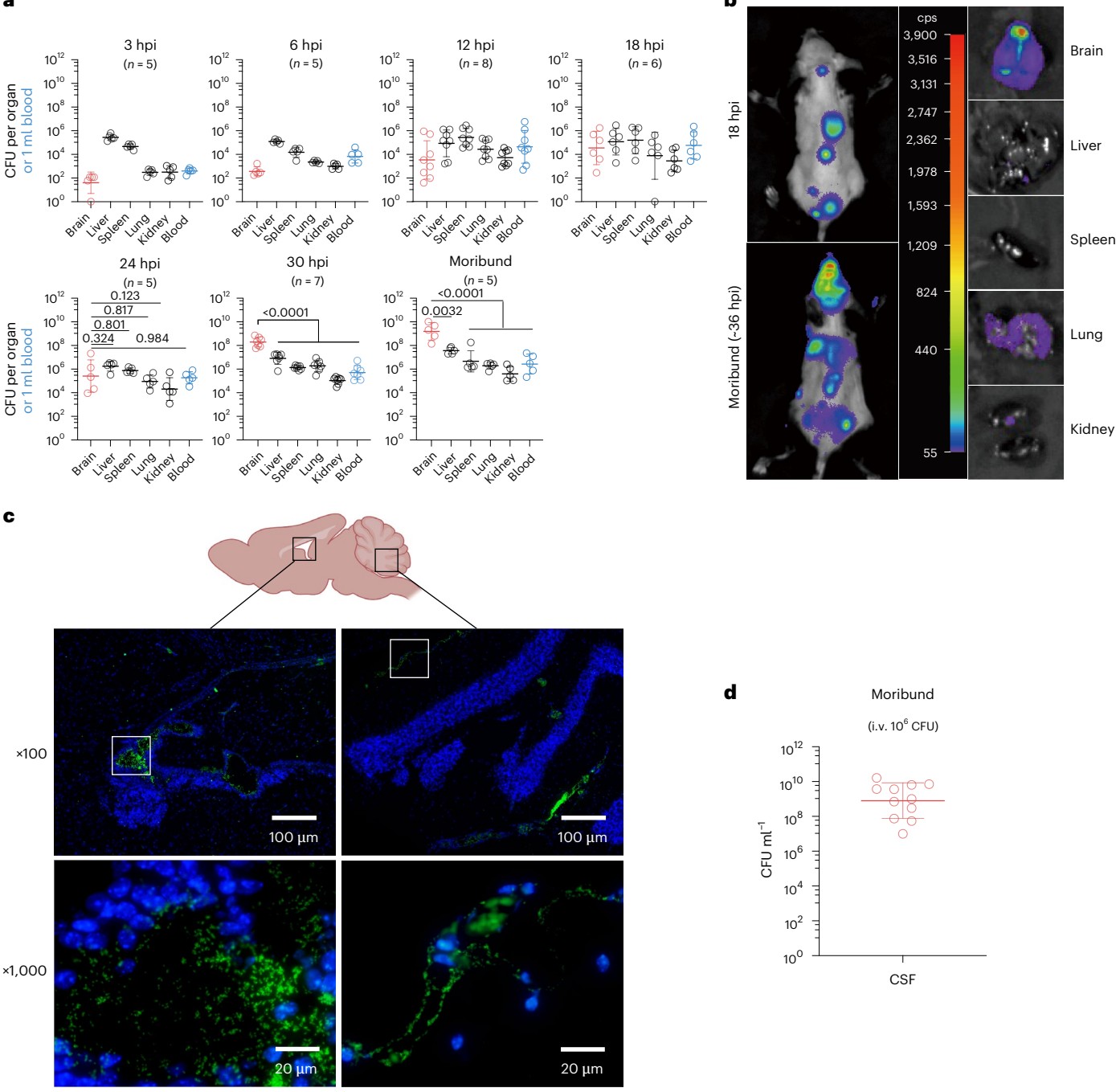

**Fig. 1 | *S. zooepidemicus* proliferates in the CSF and reaches high CFU in the brain. a**, CFU burden from organs over time after tail vein injection of $5 \times 10^6$ CFU *S. zooepidemicus* (geometric mean ± geometric s.d.; one-way ANOVA followed by Dunnett's multiple-comparison test on log-transformed CFU). Mice were moribund between 29 hpi and 39 hpi. **b**, C57BL/6J mice intravenously injected with $5 \times 10^6$ CFU of $SEZ_{luc}$. Left panels: in vivo bioluminescence imaging at 18 hpi reveals *S. zooepidemicus* presence and bacterial accumulation in the brain of moribund mice. Middle panel: quantification of the bioluminescence signals in counts per second (cps). Right panels: organ bioluminescence for moribund mice post-euthanasia (additional replicates are included in Extended

Data Fig. 1b). **c**,**d** C57BL/6J mice were infected with $10^6$ CFU of *S. zooepidemicus* intravenously, and the brains and CSF were collected at moribund stages for histological immunofluorescence or CFU counting. **c**, Immunofluorescence of mouse brain in the sagittal plane. Bacteria were labelled with anti-SzM monoclonal antibodies (green), and cell nuclei were stained with DAPI (blue). The black squares respectively indicate the third ventricle (left) and cerebral fissures (right). The ×1,000 images are magnified views of the content of the white boxes in the ×100 images. **d**, CFU burden in the CSF of moribund mice (geometric mean ± geometric s.d.; $n = 11$).

injection of the transposon library into the CSF in the lateral cerebral ventricle (Fig. 3a). The output $SEZ_{Tn}$ library was collected from brains of moribund mice at 13–18 hpi and subsequently prepared for sequencing and analysis. Genes showing a $\log_2$(fold change (FC)) less than −6 and *P* value less than 0.001 were defined as essential for *S. zooepidemicus*

survival and proliferation within the CSF (Extended Data Fig. 3b and Supplementary Table 1). We found that the majority of the top 30 under-represented genes could be categorized into four major functional groups, including ABC transporter systems, PEP–PTS, proline synthesis and transcription factors (TF) (Supplementary Table 2).

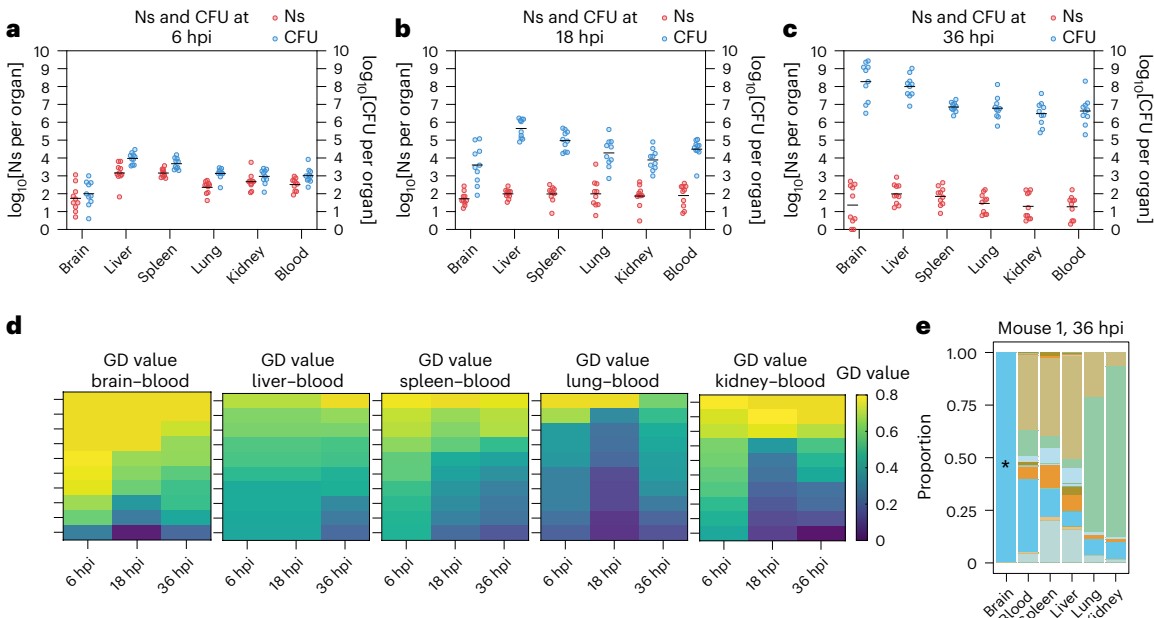

**Fig. 2 | Analysis of _S. zooepidemicus_ infection dynamics with barcoded bacteria. a–c**, $\log_{10}$(CFU) and $\log_{10}$(Ns) recovered from organs following tail vein injection of $5 \times 10^6$ CFU SEZ$_{STAMP}$ at 6 hpi (**a**), 18 hpi (**b**) and 36 hpi (**c**) ($n = 10$). After anaesthesia was induced with pentobarbital, cardiac perfusion was performed on all mice to ensure clear differentiation between blood and other organs. **d**, The GD heatmap illustrates GD in blood and various organs at 6 hpi, 18 hpi and 36 hpi. Each tile indicates one mouse. Higher GD values in the brain and blood indicate less similar bacterial populations compared with other organs and blood. Each time point includes data from 10 mice. **e**, Barcode frequency distribution of mouse 1 at 36 hpi. The proportion of each colour indicates the relative abundance of barcodes within each organ. Each colour represents one barcode occupying a percentage of the total barcodes in one organ. The asterisk indicates the primary abundant barcode in the brain that is distinct from the blood.

Four genes were linked to metabolism (marked with asterisks in Supplementary Table 2): _manY_ and _SESEC_RSO6210_, which belongs to the PTS system, and _proB_ and _proC_, which participate in proline biosynthesis. The Δ_proB_ and Δ_proC_ strains completely lost virulence owing to the impaired ability to resist early host immune clearance and survive in the bloodstream (Extended Data Fig. 4a,b). The Δ6210 had reduced mortality (Extended Data Fig. 4c), while the Δ_manY_ mutant showed similar lethality as wild-type (WT) _S. zooepidemicus_ (Extended Data Fig. 4e). Notably, the reduction in average CFU in both the Δ_manY_ and Δ6210 strains was highly specific to the brain. The average brain CFU count in moribund mice infected with Δ_manY_ was ~1,000-fold lower than that in the WT (Fig. 3b), and markedly more consistent than the Δ6210 strain (Extended Data Fig. 4d). Furthermore, transposon insertion distribution showed that the PTS operon containing the _SESEC_RSO6210_ gene is probably non-essential for proliferation in the brain, as other components (for example, EIIA) contained numerous insertions (Extended Data Fig. 3c), whereas insertions were depleted throughout the entire PTS$_{man}$ operon (Extended Data Fig. 3d). The requirement for _manY_ for growth in the brain and CSF was confirmed by complementation (CΔ_manY_) via i.v. (Fig. 3b) or stereotaxic injection (Fig. 3c). Complementation of _manY_ restored the high CFU burden specifically in the brain to WT levels, suggesting that _manY_ is specifically associated with _S. zooepidemicus_ proliferation in the CNS.

We further evaluated the severity of meningitis in moribund mice infected with WT or Δ_manY_ strains. Haematoxylin and eosin (H&E) staining showed that WT infection induced severe pathological changes in the brain. By contrast, Δ_manY_-infected mice showed only mild lesions (Fig. 3d), suggesting that the reduced bacterial burden is associated with diminished brain injury in bacterial meningitis. Despite the differing brain pathology, similar survival between WT- and Δ_manY_-infected mice suggests that death may still be associated with bacterial accumulation in non-CNS tissues. However, under penicillin treatment, Δ_manY_-infected mice had greater survival compared with WT-infected mice (Fig. 3e). This antibiotic impairs growth of WT and

Δ_manY_ similarly (minimum inhibitory concentration = 30 ng ml⁻¹) and reduced bacterial burden outside the CNS, while the surviving mice challenged with Δ_manY_ showed lower bacterial burden in the brain than the moribund ones challenged with either WT or Δ_manY_ after bacteraemia was controlled by antibiotic treatment (Fig. 3f). These findings suggest that reducing the bacterial burden in the brain could enhance the therapeutic benefits of antibiotics, suggesting that targeting nutrient uptake mechanisms during CNS infection may be a useful adjunctive therapy to antibiotics when bacteria invade the brain.

## PTS$_{man}$ is required for _S. zooepidemicus_ glucose utilization
The observation that _manY_ impacts _S. zooepidemicus_ growth in the brain suggests that carbohydrates imported by PTS$_{man}$ fuel _S. zooepidemicus_ replication in the CNS. To ascertain the carbohydrate specificity of PTS$_{man}$, we assessed _S. zooepidemicus_ utilization of 49 carbohydrates. _S. zooepidemicus_ was able to metabolize 14 of 49 (Extended Data Fig. 5a), with 6 supporting substantial growth in a chemically defined medium (CDM), achieving an OD$_{600}$ > 0.2 in a 2 g l⁻¹ concentration following 18 h of incubation at 37 °C (Fig. 4a and Extended Data Fig. 5b). Δ_manY_ showed reduced growth in glucose, fructose, mannose and _N_-acetylglucosamine (GlcNAc), but not in lactose or sucrose, compared with WT (Fig. 4a). WT _S. zooepidemicus_ and its derivatives showed similar growth phenotypes in swine CSF as in CDM following addition of fructose, mannose, GlcNAc and sucrose as sole carbon sources (Extended Data Fig. 5c). Using a PTS activity assay that measures nicotinamide adenine dinucleotide (NADH) as an indicator of PTS activity[24], we found that the deletion of _manY_ resulted in a significant reduction in NADH when glucose, fructose, mannose and GlcNAc were provided as the sole carbon source (Fig. 4b), suggesting that PTS$_{man}$ at least participates in the transportation of these four carbohydrates during _S. zooepidemicus_ growth in vitro.

In human CSF, glucose is the predominant carbohydrate, with fructose and mannose present in markedly lower concentrations[25]. When _S. zooepidemicus_ was cultured in CDM mimicking the physiological

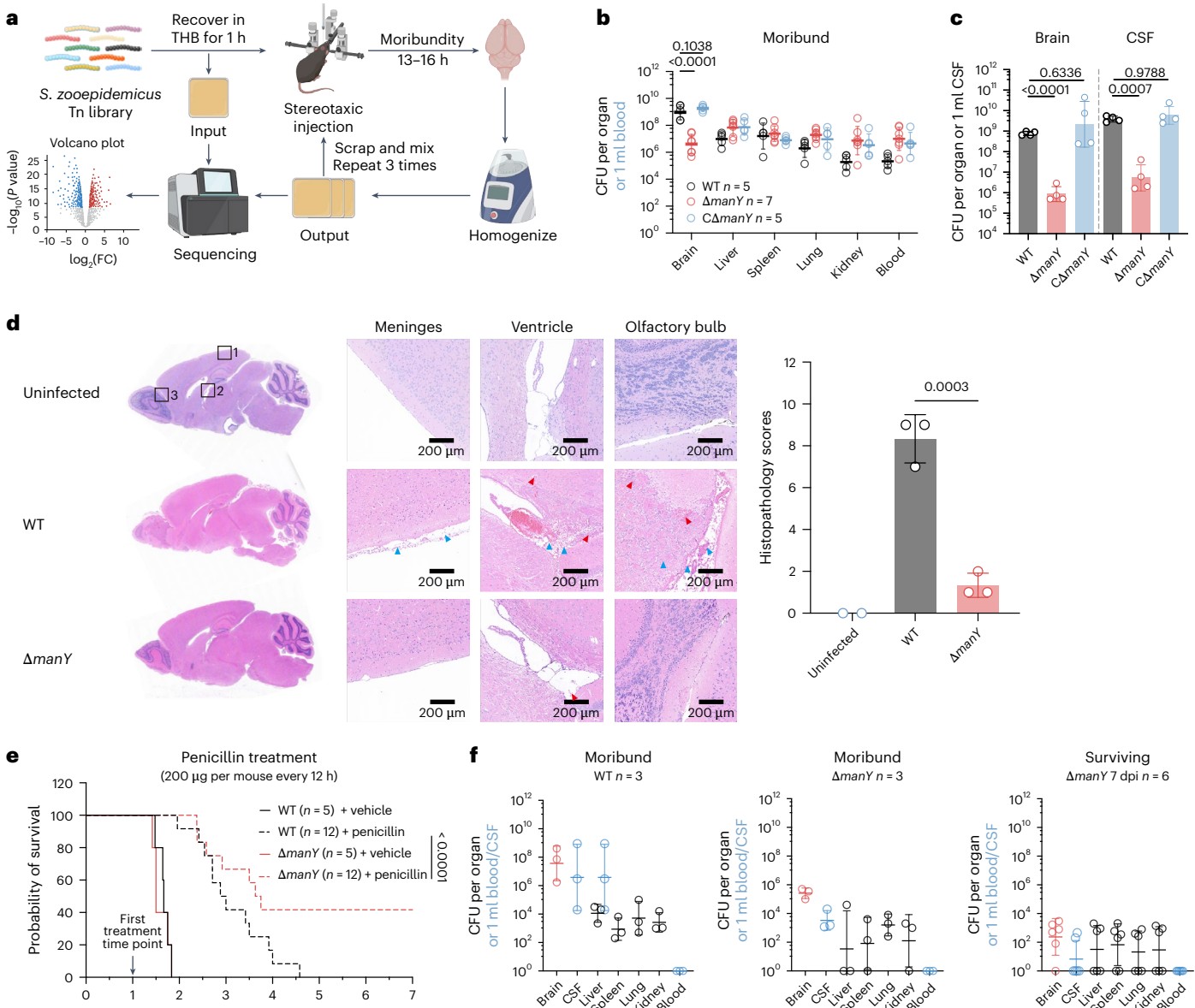

**Fig. 3 | Identification of essential genes contributing to *S. zooepidemicus* proliferation in the CSF. a**, Experimental schematic of the SEZ$_{Tn}$ library screening workflow. The SEZ$_{Tn}$ library, recovered in THB for 1 h, was directly injected into the lateral ventricles of mice using a stereotaxic device for initial screening. The library retrieved from the brain post-injection was processed to prepare for successive screening rounds. The libraries of three rounds of screening were collected from the brain designated as the output. **b**, CFU burden in the organs and blood of moribund mice. C57BL/6J mice were challenged with $5 \times 10^6$ CFU of WT, $\Delta manY$ or C$\Delta manY$ via tail vein injection (two-way ANOVA following Šídák's multiple-comparison test versus WT on log-transformed CFU). **c**, WT, $\Delta manY$ and C$\Delta manY$ were directly injected into the lateral cerebral ventricle of mice at $5 \times 10^6$ CFU using a digital mouse stereotaxic instrument. The brain and CSF CFU were counted at 12 hpi ($n = 4$; one-way ANOVA following Dunnett's multiple-comparison test versus WT on log-transformed CFU). **d**, H&E staining of brain sections from uninfected mice ($n = 2$) and moribund mice

intravenously infected with WT or $\Delta manY$ strains ($n = 3$). The left panel shows low-magnification views, with black squares indicating selected regions for detailed analysis: meninges, ventricle and olfactory bulb. The red arrows indicate haemorrhages. The blue arrows highlight neutrophil infiltration. WT shows obvious pathological features such as neutrophil infiltration and haemorrhage, while $\Delta manY$ shows mild pathology. Summarized histopathology scores of brain sections from indicated mice are shown in the right panel. **e**, Survival curve of C57BL/6J mice after i.v. administration of 200 μg penicillin every 12 h. Mice were challenged intravenously with $5 \times 10^6$ CFU WT or $\Delta manY$ and subsequently given penicillin or vehicle (PBS) treatment starting at 24 hpi (Kaplan–Meier methods following a log-rank test). **f**, CFU burden in the organs and blood of moribund and surviving mice following infection and penicillin treatment. In **b**, **c** and **f**, data are presented as geometric mean ± geometric s.d. In **d**, data are presented as mean ± s.d. Panel **a** created with BioRender.com.

carbohydrate concentrations of CSF, only glucose was sufficient to support growth (Fig. 4c). We hypothesized that PTS$_{man}$ promotes *S. zooepidemicus* proliferation in CSF by enabling glucose acquisition, as CSF glucose levels decline during infection (Fig. 4d) and the $\Delta manY$ showed reduced growth in CSF (Fig. 3c). To assess the impact of PTS$_{man}$ on *S. zooepidemicus* growth across varying glucose concentrations,

we measured the CFU of WT, $\Delta manY$ and C$\Delta manY$ at 12 hpi (stationary phase) in CDM. The $\Delta manY$ strain had significantly fewer CFU compared with WT and C$\Delta manY$ at glucose concentrations between 0.2 g l$^{-1}$ and 2 g l$^{-1}$, but as glucose concentrations were increased, the discrepancy in CFU between the three strains diminished (Fig. 4e). Comparisons of the growth curves of WT and $\Delta manY$ also revealed that

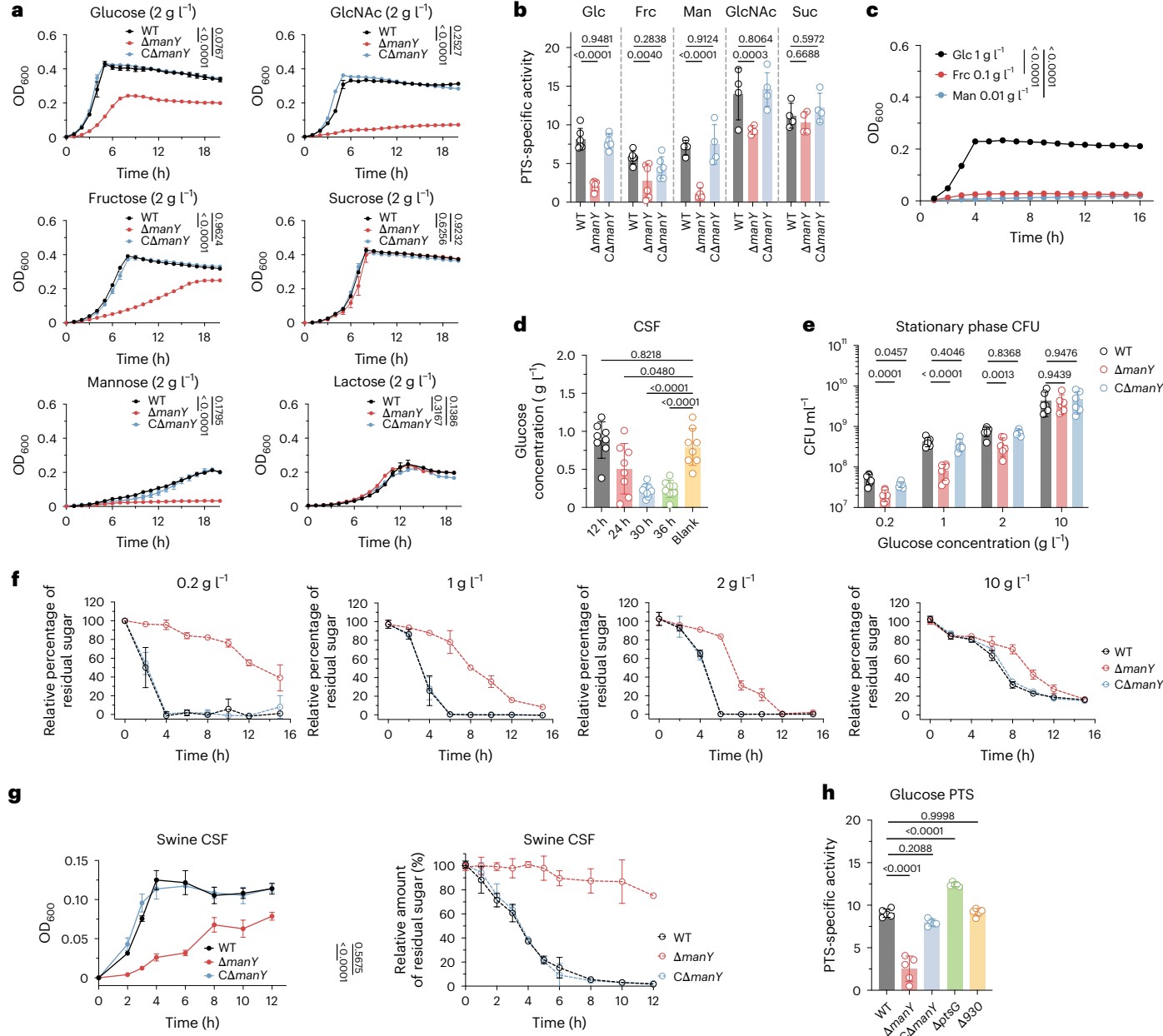

**Fig. 4 | PTS_man is critical for glucose utilization under low glucose concentrations. a**, Growth curves of WT, ΔmanY and CΔmanY with six preferred carbon sources in CDM (n = 4). **b**, PTS-specific activity was detected by measuring NADH decrease at OD₃₄₀ in toluene-treated suspensions of WT, ΔmanY and CΔmanY, using five carbohydrates as the sole carbon source in CDM. Carbon sources shown are those that led to significant growth defects in ΔmanY versus WT (Glc, glucose; Frc, frucose; Man, mannose) and one that does not lead to growth defects in ΔmanY (Suc, sucrose). n = 6 for Glc and Frc; n = 4 for others. **c**, Growth curve (n = 3) of WT in CDM with carbohydrates at physiological concentrations of CSF. **d**, Glucose concentrations in mouse CSF (n = 8) at 12 hpi, 24 hpi, 30 hpi and ~36 hpi (moribund) after i.v. infection with 5 × 10⁶ CFU WT. 'Blank' indicates CSF without infection. **e**, The CFU of WT, ΔmanY and CΔmanY (n = 6) at stationary phase (cultured for 12 h) in CDM with 0.2 g l⁻¹, 1 g l⁻¹, 2 g l⁻¹ and 10 g l⁻¹ glucose.

**f**, The relative percentage of residual glucose in CDM with 0.2 g l⁻¹, 1 g l⁻¹, 2 g l⁻¹ and 10 g l⁻¹ glucose after inoculation with WT, ΔmanY and CΔmanY (n = 3). The relative percentage of residual glucose is calculated by dividing the glucose concentration remaining in CDM at a specific time by the initial glucose concentration (0 h). **g**, Left: growth curve of WT, ΔmanY and CΔmanY cultured in swine CSF in vitro (n = 3). Right: the relative percentage of residual glucose in swine CSF after inoculation with WT, ΔmanY and CΔmanY in vitro (n = 3). **h**, PTS-specific activity in glucose of WT and indicated mutants (n = 5). All data in this figure are presented as mean ± s.d. In **a** and **c**, growth curves were modelled using a Gompertz nonlinear regression model; estimated parameters were compared across groups by one-way ANOVA with Dunnett's multiple-comparison test. In **b** and **e**, two-way ANOVA with Dunnett's multiple-comparison test was used. In **d** and **h**, one-way ANOVA with Dunnett's multiple-comparison test was used.

the mutant had slower growth, particularly in low glucose conditions (Extended Data Fig. 5d). We compared glucose utilization by WT and ΔmanY by determining the residual glucose concentration in media after growth in different glucose concentrations. In media containing 0.2 g l⁻¹ glucose, the WT strain consumed 100% of the glucose within 4 h, whereas the ΔmanY used only ~50% of the available glucose by 15 h.

Similar but less dramatic discrepancies in the glucose utilization of WT and ΔmanY were observed at 1 g l⁻¹ and 2 g l⁻¹ glucose, although the mutant achieved near-complete glucose utilization by 12 h. By contrast, at 10 g l⁻¹ glucose, the discrepancies were less pronounced, and all the strains used ~70% of the available glucose by 12 h (Fig. 4f). The defects in growth and glucose utilization in ΔmanY were also apparent in CSF

from swine, indicating that the role of *manY* is not likely to be specific to mouse CSF (Fig. 4g). These results suggest that the function of PTS$_{man}$ is critical to bacterial growth under the low glucose conditions that are physiologically relevant in the CSF.

To determine whether other PTS systems are activated during growth in glucose, we conducted RNA sequencing (RNA-seq) to characterize the transcriptomes of WT and Δ*manY* grown in different glucose concentrations. At 2 g l$^{-1}$ glucose, we found that the PTS$_{man}$ (encoded by *manX*, *manY* and *manZ*) and the PTS$_{930}$ (encoded by *SESEC_RS00930*, *SESEC_RS00935* and *SESEC_RS00940*) were expressed at higher levels compared with other PTS genes (Extended Data Fig. 5e and Supplementary Table 3). However, growth of a Δ930 mutant was unaffected in glucose-supplemented media (Extended Data Fig. 5f). Furthermore, the *S. zooepidemicus ptsG* homologue, which in *Escherichia coli* encodes the glucose-inducible glucose-specific integral membrane EIIABC transporter, was not appreciably induced in response to glucose in *S. zooepidemicus* (Extended Data Fig. 5e). In addition, Δ*ptsG* did not show a growth defect in media with glucose as the sole carbohydrate (Extended Data Fig. 5f). Furthermore, deletions of *SESEC_RS00930* or *ptsG* did not impact PTS activity (Fig. 4h). By contrast, Δ*manY* showed a reduction in PTS activity, which was restored in CΔ*manY* (Fig. 4h). These observations indicate that *S. zooepidemicus* relies predominantly on PTS$_{man}$ for growth in media with glucose as the sole carbohydrate.

## PTS$_{man}$ promoter confers *S. zooepidemicus* growth advantage in low glucose

The *S. zooepidemicus* PTS$_{man}$ operon, consisting of *manX*, *manY* and *manZ*, is highly conserved across various streptococcal species (Extended Data Fig. 6a). The genes in this operon are constitutively expressed as a single cistron in *S. zooepidemicus* (Extended Data Fig. 6b). Using BPROM (http://linux1.softberry. com/berry.phtml), we identified a region upstream of *manX* containing the likely promoter sequence, which differs across species of streptococci (Extended Data Fig. 6c). Remarkably, the *S. zooepidemicus* promoter showed markedly higher transcriptional activity in media with 2 g l$^{-1}$ glucose compared with its counterparts from other streptococci when the respective transcriptional fusions were expressed in either the *S. zooepidemicus* or GBS background (Fig. 5a). Moreover, the PTS$_{man}$ promoter (P$_{manX}$) had the highest transcriptional activity compared with other *S. zooepidemicus* PTS promoters (Extended Data Fig. 6d), and the *manY* gene showed constitutive transcription regardless of glucose concentration (Fig. 5d). β-galactosidase assays with promoter fusions to lacZ revealed that the nucleic acid sites from −152 to −98 of *manX*, which contained the −35 and −10 regions, are critical for the promoter activity (Fig. 5b). As group A *Streptococcus* (GAS) had the most similar P$_{manX}$ to *S. zooepidemicus* (differing in only two nucleotides, Extended Data Fig. 6c), we constructed mutants in the P$_{manX}$ to introduce the GAS sequence into *S. zooepidemicus* P$_{manX}$. Mutations at position −110 and −133 markedly reduced the *S. zooepidemicus* P$_{manX}$ activity (Fig. 5c), suggesting that species-specific polymorphisms in P$_{manX}$ contribute to the high transcriptional activity of PTS$_{man}$.

To further investigate the relationship of P$_{manX}$ to bacterial growth and CFU burden during infection, we replaced the *S. zooepidemicus* P$_{manX}$ with the GBS A909 P$_{manX}$ (SEZ::P$_{manX(A909)}$). The GBS A909 promoter drove much lower *manY* expression levels than the native *S. zooepidemicus* promoter (Fig. 5d), and the growth of SEZ::P$_{manX(A909)}$ in low glucose was comparable to that of the Δ*manY* mutant (Fig. 5e,f). Moreover, the brain and CSF CFU burden in moribund mice inoculated with SEZ::P$_{manX(A909)}$ were more than 100× lower than those inoculated with *S. zooepidemicus* via tail vein or intracranial injection, while there was no significant difference in other organs (Fig. 5g,h and Extended Data Fig. 6e). Together, these observations suggest that the PTS$_{man}$ promoter, which drives high-level expression of the glucose-transporting PTS$_{man}$ system in

*S. zooepidemicus*, confers a species-specific growth advantage to *S. zooepidemicus* in low-glucose environments.

## PTS$_{man}$ prevents stringent response under limited glucose

Comparisons of the transcriptomes of WT and Δ*manY S. zooepidemicus* at varying glucose concentrations were carried out to further assess the impact of PTS$_{man}$ on *S. zooepidemicus* gene expression (Supplementary Table 4). The number of differentially expressed genes (DEGs) between WT and Δ*manY* decreased (from 894 to 179 DEGs) as glucose concentrations increased, suggesting that PTS$_{man}$ facilitates *S. zooepidemicus* glucose responsiveness particularly in low-glucose conditions (Extended Data Fig. 7a). By comparing the DEGs in the 1 g l$^{-1}$ glucose group (that is, influenced by *manY* in low glucose) with the non-DEGs in the 10 g l$^{-1}$ glucose group (that is, not influenced by *manY* in high glucose) (Extended Data Fig. 7c), we identified 773 DEGs that may influence bacterial growth specifically under low-glucose conditions in a PTS$_{man}$-dependent manner.

Kyoto Encyclopedia of Genes and Genomes (KEGG) Orthology pathway analysis revealed that more than half of the 773 DEGs could be classified in pathways. Notably, a series of KEGG pathways related to the stringent response were enriched (Extended Data Fig. 7d). The stringent response is mediated by the (p)ppGpp in response to nutrient deprivation stress signals[26]. Comparison between the (p)ppGpp$^0$ strain (Δ*relA*Δ*relQ*) and WT identified 887 (p)ppGpp-dependent genes, 432 of which overlapped with the 773 *manY*-dependent DEGs (Extended Data Fig. 7b,c). Most of these 432 genes showed opposite expression trends in Δ*manY* and Δ*relA*Δ*relQ*, consistent with their contrasting (p)ppGpp levels, elevated in Δ*manY* and depleted in Δ*relA*Δ*relQ* (Extended Data Fig. 7e). These results indicate that the constitutive activity of *manY* in *S. zooepidemicus* prevents transcriptional activation of the stringent response under low glucose.

The stringent response is orchestrated by the synthetase–hydrolase of the hyperphosphorylated guanosine nucleotide (p)ppGpp[27], mediated by the ribosome-associated protein RelQ and RelA in *S. zooepidemicus*. However, only *relQ* (synthetase) expression was elevated in Δ*manY* under low-glucose conditions, as shown by RNA-seq (Extended Data Fig. 7f) and confirmed by qPCR, whereas *relQ* expression in both WT and CΔ*manY* strains remained low and constant across different glucose concentrations (Fig. 6a). Overexpression of *relQ* in WT (*relQ*$^+$) led to elevated (p)ppGpp levels and impaired growth (Fig. 6b,d). The deletion of the *relQ* gene in Δ*manY* significantly reduced the intracellular (p)ppGpp levels and rescued the growth defect caused by the absence of *manY* under low-glucose conditions, indicating that the growth inhibition in Δ*manY* is dependent on *relQ* (Fig. 6b). Conversely, reintroducing *relQ* into the Δ*manY*Δ*relQ* restored (p)ppGpp (Fig. 6c and Extended Data Fig. 8a) production and led to the reappearance of the growth defect (Fig. 6b). These results suggest that the intracellular (p)ppGpp level is closely associated with the growth of *S. zooepidemicus* under low-glucose conditions. Moreover, an increase in (p)ppGpp was observed in Δ*manY* cultured under low-glucose conditions, whereas (p)ppGpp was less in WT or Δ*manY* when grown in higher glucose concentrations by either TLC (Fig. 6d and Extended Data Fig. 8b) or an RNA-based fluorescent sensor (Extended Data Fig. 8c). Together, these observations indicate that constitutive activation of *manY* suppresses activation of the stringent response.

Finally, we assessed the role of several key intermediate proteins that link PTS$_{man}$ to the stringent response. The PTS$_{man}$ component is part of the EII complex within the PTS system and is functionally associated with the EI complex, which includes HPr and the EI enzyme. HPr phosphorylation has been reported to reduce Crp expression in *Listeria monocytogenes*[28]. In 1 g l$^{-1}$ glucose CDM, both Δ*manY* and SEZ::P$_{manX(A909)}$ showed low phosphorylated HPr and high Crp expression levels. By contrast, the WT and CΔ*manY* strains showed the opposite phenotypes (high phosphorylated HPr and low Crp). In 10 g l$^{-1}$ glucose CDM, all

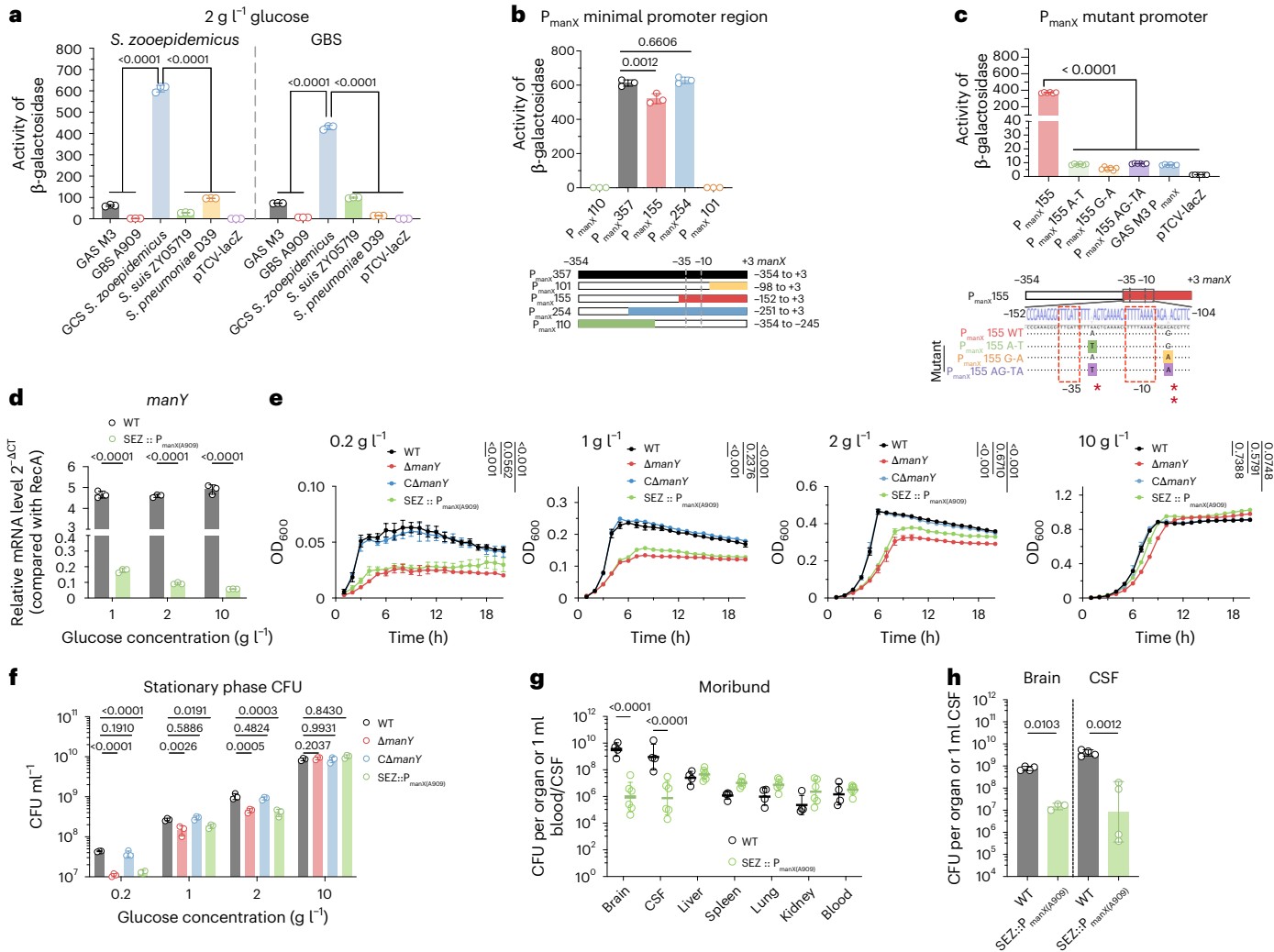

**Fig. 5 | The PTS_man promoter confers a growth advantage to *S. zooepidemicus* in low glucose. a**–**c**, Evaluation of promoter activities. **a**, P_manX from different *Streptococcus* were cloned into pTCV-lacZ and introduced into WT *S. zooepidemicus* or GBS A909 to evaluate transcriptional activity (*n* = 3). *S. suis* refers to *Streptococcus suis*, and S. pneumoniae refers to *Streptococcus pneumoniae*. **b**, Truncated region from upstream of *manX* was cloned into pTCV-lacZ to identify the potential promoter (*n* = 3). The diagram shows truncated regions, with bar chart colours matching the diagram. **c**, Nucleotide alignment of the −152 to −104 region in P_manX from GAS and GCS shows two nucleotide mutations (*n* = 6). Site-directed mutants were induced into the *S. zooepidemicus* P_manX promoter-based GAS mutants and cloned into the pTCV-lacZ to assess its transcriptional activity. Mutant bases at −133 and −110 of the *manX* gene are marked with '*' and '**' as shown in Extended Data Fig. 6c. **d**, Relative *manY* transcript levels in WT and SEZ::P_manX(A909) strains detected by qRT-PCR and normalized to *recA* (*n* = 3). **e**, Growth curve (*n* = 4) of WT, Δ*manY*, CΔ*manY* and

SEZ::P_manX(A909) in CDM with 0.2 g l⁻¹, 1 g l⁻¹, 2 g l⁻¹ and 10 g l⁻¹ glucose. **f**, CFU of WT, Δ*manY*, CΔ*manY* and SEZ::P_manX(A909) at stationary phase (12 h) in CDM with 0.2 g l⁻¹, 1 g l⁻¹, 2 g l⁻¹ and 10 g l⁻¹ glucose (*n* = 3, data modelled using a Gompertz nonlinear regression model, followed by one-way ANOVA with Dunnett's multiple-comparison test). **g**, C57BL/6J mice were intravenously injected with *S. zooepidemicus* or SEZ::P_manX(A909), CFU were counted from organs of moribund mice (*n* = 4; SEZ::P_manX(A909)-infected mice moribund between 31 hpi and 37 hpi; two-way ANOVA following Šídák's multiple-comparison test on log-transformed CFU). **h**, *S. zooepidemicus* or SEZ::P_manX(A909) was injected into the lateral cerebral ventricle (5 × 10⁶ CFU) using a digital mouse stereotaxic instrument. Brain and CSF CFU were counted at 12 hpi (*n* = 4; one-way ANOVA with Dunnett's multiple-comparison test on log-transformed CFU). *n* indicates biological replicates from different single bacterial clones (**a**–**f**) or mice (**g**, **h**). Data are shown as mean ± s.d. (**a**–**f**) or geometric mean ± geometric s.d. (**g**, **h**). In **a**–**d** and **f**, one-way ANOVA with Dunnett's multiple-comparison test was used.

four strains showed high phosphorylated HPr and low Crp expression (Fig. 6e and Extended Data Fig. 8d). Electrophoretic mobility shift assay (EMSA) experiments showed that Crp binds to the *relQ* promoter of *S. zooepidemicus* in the presence of cAMP (Fig. 6f). The deletion of *crp* significantly reduced *relQ* expression and (p)ppGpp levels in the Δ*manY* mutant under 1 g l⁻¹ glucose conditions (Fig. 6g,h and Extended Data Fig. 8e). These results indicate that the increase in phosphorylated HPr reduces the expression of the *relQ* transcriptional regulator cAMP–Crp complex, subsequently decreasing (p)ppGpp levels and preventing activation of the stringent response (Fig. 6i). These findings collectively suggest that PTS_man enables glucose acquisition of *S. zooepidemicus* at low glucose concentrations, which may in turn

limit activation of the stringent response and thereby enable its robust replication in the CSF during bacterial CNS infection.

## Discussion

*S. zooepidemicus* is capable of proliferating in CSF, in some cases even reaching densities observed in in vitro cultures in rich media. Our study reveals that *S. zooepidemicus* possesses a specialized glucose acquisition strategy within carbohydrate-limited environments mediated by its PTS_man, which is critical to the robust proliferation of *S. zooepidemicus* within CSF. Expression of *S. zooepidemicus* PTS_man is promoted by a strong promoter that distinguishes *S. zooepidemicus* from other streptococci because it remains active at the low glucose concentrations

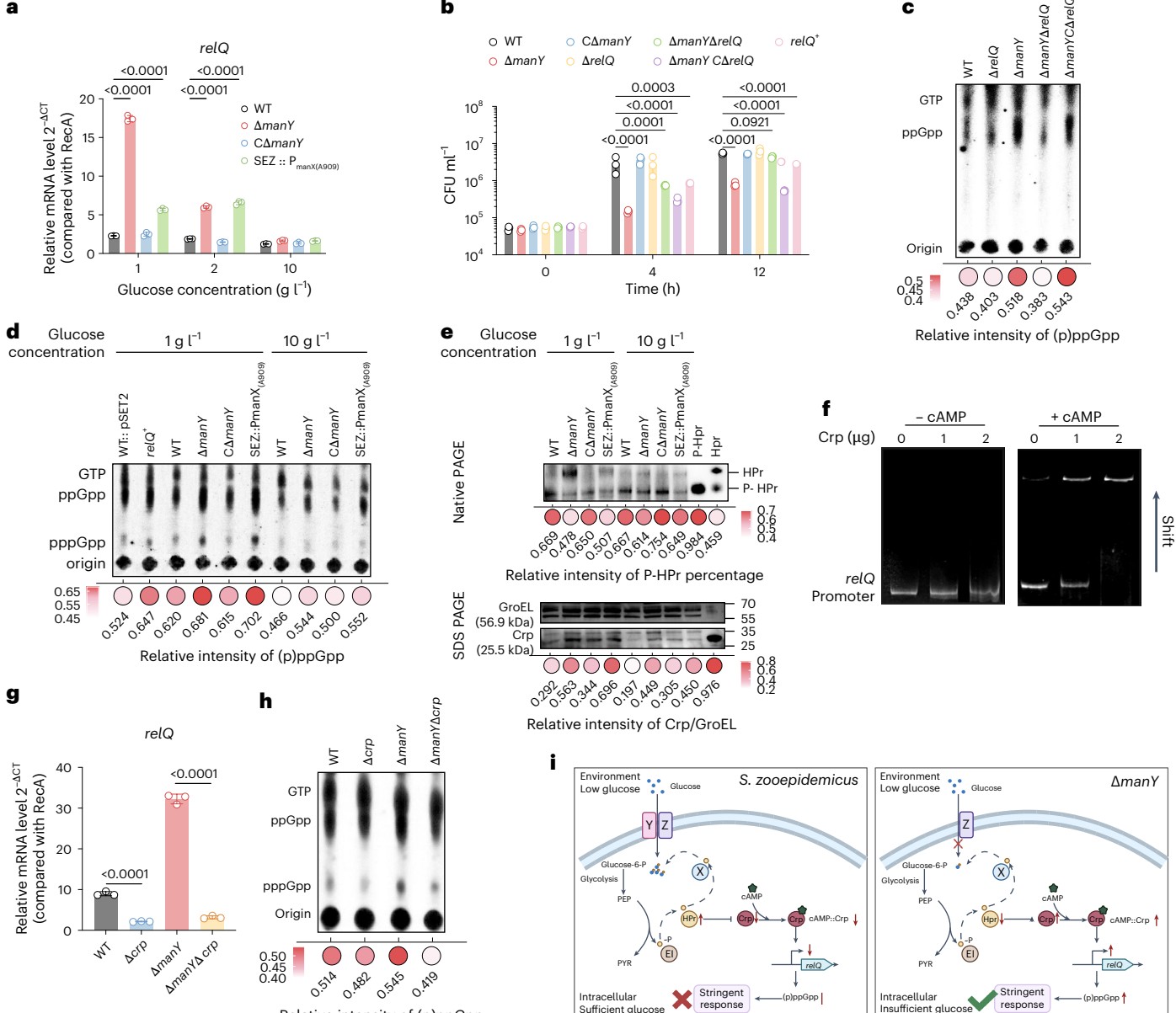

**Fig. 6 | PTS$_{man}$-dependent bypass of the stringent response contributes to *S. zooepidemicus* proliferation in low-glucose concentrations. a**, Relative *relQ* transcript levels in WT, Δ*manY*, CΔ*manY* and SEZ::P$_{manX(A909)}$ strains of *S. zooepidemicus*, detected by qRT-PCR and normalized to *recA* ($n = 3$). **b**, CFU of WT, Δ*manY*, CΔ*manY*, Δ*relQ*, Δ*manY*Δ*relQ*, Δ*manY*CΔ*relQ* and *relQ*⁺ strains in CDM with 1 g l⁻¹ glucose at 0 h, 4 h and 12 h ($n = 3$). *relQ*⁺ refers to *S. zooepidemicus* containing the *relQ* over-expressing plasmid (pSET2-*relQ*). **c,d**, Intracellular (p)ppGpp levels detected by TLC after nucleotide extraction. Bacteria were grown in CDM with 1 g l⁻¹ glucose (**c**) or 1 g l⁻¹ (low) versus 10 g l⁻¹ (high) glucose (**d**). **e**, Western blot analysis of HPr phosphorylation and Crp expression in WT, Δ*manY*, CΔ*manY* and SEZ::P$_{manX(A909)}$ strains grown in CDM with 1 g l⁻¹ or 10 g l⁻¹ glucose. GroEL as a loading control. SDS-PAGE (sodium dodecyl sulfate polyacrylamide gel electrophoresis) was used to separate Crp and GroEL. Native PAGE was used to separate HPr and P-HPr with their native conformation. **f**, EMSA assessing *relQ* promoter binding to Crp or Crp:cAMP complex. **g**, Relative transcript levels of *relQ* in WT, Δ*crp*, Δ*manY* and Δ*manY*Δ*crp* strains, detected by qRT-PCR ($n = 3$). Data were normalized to *recA*. **h**, Intracellular levels of (p)

ppGpp were detected by TLC after nucleotide extraction of *S. zooepidemicus* WT, Δ*crp*, Δ*manY* and Δ*manY*Δ*crp* strains. Bacteria were grown in CDM with 1 g l⁻¹ glucose. **i**, Schematic model illustrating the pathway between the PTS$_{man}$ system and stringent response. Under low-glucose conditions, *S. zooepidemicus* appears to transport glucose through PTS$_{man}$, leading to phosphorylation of HPr. The high ratio of P-HPr was associated with reduced Crp expression, which inhibited *relQ* activation by the Crp:cAMP complex, thereby preventing (p) ppGpp accumulation and the onset of the stringent response. By contrast, Δ*manY* impaired glucose uptake results in a lower P-HPr ratio and increased Crp expression, inducing *relQ* transcription and activating the stringent response. All samples used in qPCR and WB were collected at mid-exponential phase. The bubble plots below each panel represent the grey intensity corresponding to the results. The colour intensity of red indicates the levels of (p)ppGpp, the percentage of P-HPr or Crp expression. In **a**, **b** and **g**, data are presented as mean ± s.d.; one-way ANOVA with Dunnett's multiple-comparison test was used. Panel **i** created with BioRender.com.

that develop during meningitis. The constitutive expression of PTS$_{man}$ enables *S. zooepidemicus* to avoid activation of the stringent response in glucose-limited conditions in CSF (Extended Data Fig. 9). These results unveil an adaptation mechanism that enables efficient glucose

utilization in the CSF, thus promoting *S. zooepidemicus* proliferation under limiting nutrient conditions.

Barcoded bacteria have been widely used to explore the dynamics of pathogen infections[29,30]. Recent work using the STAMPR analytic

framework and barcoded *L. monocytogenes* uncovered patterns of pathogen dissemination to the CNS[31]. Following tail vein injection, in both *L. monocytogenes* and *S. zooepidemicus* models, the brain is seeded by very few founders. However, in contrast to *S. zooepidemicus* in which the number of founders was relatively constant, the *L. monocytogenes* founding populations increased in the brain over time, indicating that new clones were continually translocating across the BBB. Notably, the CFU of *S. zooepidemicus* in the brain at 1 day post-inoculation (dpi) is nearly five orders of magnitude greater than the CFU of *L. monocytogenes*. It is possible that there is continued *S. zooepidemicus* translocation across the BBB, but *S. zooepidemicus* replication is so rapid ($10^8$ CFU by 30 h) that it may be prevent the detection of transit of new clones.

Once bacteria penetrate the physical barriers of the CNS to enter the CSF, this fluid provides a relatively protected niche for bacterial growth, as the CNS is an immune-privileged site[32]. However, we and others have found that the CSF offers limited nutrient concentrations, which are rapidly depleted by bacterial replication, necessitating pathogen strategies for adaptation to and acquisition of sufficient nutrients[11,15]. To acquire carbon sources, bacterial PTS systems catalyse the uptake and concomitant phosphorylation of various carbohydrates. In addition, the PTS has a major role in bacterial carbon catabolite repression and the regulation of expression of many genes that are not directly linked to sugar import[16,33]. The EII components of streptococcal PTS$_{man}$ are sugar-specific transporters with three conserved domains, termed manLMN in *S. pneumoniae* or *manXYZ* in *S. zooepidemicus*. Although our study focused on the glucose in the CSF, the transcription of PTS EII could be controlled by additional carbohydrates and vary in other conditions. For example, expression of *ptsG* in *E. coli* is regulated by a global repressor Mlc that monitors glucose levels[34]. However, we did not observe glucose-dependent repression of the *S. zooepidemicus manXYZ* operon. Even though PTS$_{man}$ in *S. zooepidemicus* is crucial for glucose transport, its transcriptional regulation remains unaffected by glucose levels. The glucose insensitivity of PTS$_{man}$ expression could be due to the specific PTS$_{man}$ promoter sequence of *S. zooepidemicus*, which differs across streptococcal species.

The stringent response is a broadly conserved bacterial stress response that controls adaptation to nutrient deprivation[26]. The molecular hallmark of the stringent response is synthesis of (p)ppGpp, which accumulates quickly in a RelQ- or RelA-dependent manner upon starvation and causes rapid cessation of cell growth[35]. During bacterial adaptation to carbon source limitation, the stringent response can enable the production of catabolic proteins necessary for the utilization of alternative carbon sources[36]. However, in the context of the CSF, where glucose is the sole carbohydrate capable of supporting bacterial growth, utilization of additional effective carbon sources is not an effective growth strategy. Our study suggests that through the constitutive expression of PTS$_{man}$, *S. zooepidemicus* maintains its ability to import glucose even at low concentrations, effectively circumventing the stringent response. While this mechanism may primarily enhance the resilience of *S. zooepidemicus* to glucose scarcity, it appears to enable *S. zooepidemicus* to achieve substantially higher CFU burden in the CNS compared with other *Streptococcus* species. The relationship between PTS and the stringent response has been extensively investigated in *E. coli*, in which the σ factor regulator Rsd can sense PTS-imported carbon sources to regulate the stringent response through interaction with SpoT, a bifunctional (p)ppGpp hydrolase–synthetase enzyme[18,19]. However, *Streptococcus* species lack a homologue of Rsd, leaving the relationship between the PTS and stringent response in these bacteria a fruitful topic for future research.

Together, our insights into the mechanisms underlying the adaptations of *S. zooepidemicus* to and proliferation within the CSF highlight the complexity of bacterial survival strategies in the CNS. Furthermore, our findings suggest that therapeutics targeting PTS$_{man}$ could disrupt the ability of some meningeal pathogens to acquire glucose, thereby limiting their proliferation and mitigating the deleterious consequences of CNS infections, which could increase the therapeutic effect of antibiotic treatment and extend therapeutic windows in acute infection.

## Methods

### Ethics statement

All mice used in this study were female C57BL/6J mice (GemPharmatech) aged between 6 weeks and 8 weeks to minimize variability and ensure consistency across all experimental conditions. All mice 7 days before the experiment were housed in specific-pathogen-free conditions at Nanjing Agricultural University Laboratory Animal Center. All mice were kept under 12-h light–dark cycles, with temperature controlled at 22–24 °C and humidity at 40–60%. Mice had free access to food and water. All animal experiments were performed with protocols approved by the Laboratory Animal Welfare and Ethics Committee of Nanjing Agricultural University (protocol number NJAU. No20220311038) in accordance with the Laboratory Animal Guideline for ethical review of animal welfare (GB/T 35892-2018). The mice were randomly assigned to different experimental groups, with each group including at least three biological replicates. No animals or data points were excluded from the analyses.

### Bacterial strains, plasmids and culture conditions

Strains and plasmids used in this study are listed in Supplementary Tables 5 and 6. *S. zooepidemicus* ATCC35246 was cultured in Todd Hewitt Broth (THB) medium at 37 °C with shaking at 180 rpm. For glucose utilization experiments, strains were cultured in chemical-defined medium (CDM). The formulation of CDM is shown in Supplementary Table 7. *E. coli* DH5α and BL21 (DE3) were grown in Luria–Bertani (LB) medium at 37 °C with shaking at 180 rpm. The following concentrations of antibiotics were added as needed: spectinomycin (Spc, 100 µg ml⁻¹), kanamycin (Kan, 50 µg ml⁻¹ in *E. coli*, 100 µg ml⁻¹ in *Streptococcus*, 300 µg ml⁻¹ in SEZ$_{STAMP}$ and SEZ$_{Tn}$), erythromycin (Erm, 5 µg ml⁻¹) and ampicillin (Amp, 100 µg ml⁻¹).

The fragment used to construct plasmids was amplified by PCR, and the vector was digested with two unique clone QuickCut enzymes (Takara). All PCR products and the lined vector were resolved by agarose gel electrophoresis and purified using a E.Z.N.A.® Gel eExtraction Kit (Omega, D2500-2). All plasmids were constructed by standard PCR-based cloning procedures using the ClonExpress II One Step Cloning Kit (Vazyme, C112-01-ab) and verified by sequencing (Azenta). Primers are provided in Supplementary Table 8. Plasmids constructed in this study are available upon reasonable request from the corresponding author.

### Construction of deletion and complement strains

Gene deletion mutant strains of *S. zooepidemicus* and A909 were generated using homologous recombination. The upstream and downstream sequences of the deleted genes were amplified and cloned into pSET4s. All primers are listed in Supplementary Table 8. The recombinant plasmids were introduced into competent *S. zooepidemicus* cells by electroporation at 2.30 kV, 200 Ω and 25 µF using the GenePulser X cell electroporation system (Bio-Rad). Transformants were recovered in 3 ml THB at 28 °C for 4 h and then spread on THB (Spc⁺) plates and cultured at 28 °C for 36–48 h. Cells carrying the recombinant plasmids were transferred to THB (Spc⁺) and cultured at 37 °C through 2 passages to generate single-crossover mutants. The resulting mutants were spread on THB (Spc⁺) plates and cultured at 37 °C. Single clones of double-crossover mutants grown on THB (Spc⁺) plates were repeatedly passaged on THB at 28 °C to lose the plasmid. Gene deletions were verified by PCR and Sanger sequencing. To construct complementation strains of *S. zooepidemicus* and A909, genes carrying a synonymous point mutation were amplified by PCR from genomic DNA and inserted into pSET4s. The recombinant plasmids were electroporated into the corresponding deletion strains.

## Animal experiments

Bacteria were washed three times and resuspended in phosphate-buffered saline (PBS, pH = 7.4) before injection, and the control group was injected with the same volume of PBS alone.

To determine blood and organ burden, mice were deeply anaesthetized with isoflurane and an ophthalmic local anaesthetic was applied. Blood was collected from the postorbital venous plexus. Mice were drained of as much blood as possible until blood flow diminished naturally. Approximately 300–400 µl of blood was collected from moribund mice, while 500–700 µl was collected from mice at the early infection period. Following cervical dislocation, organs were collected, homogenized and diluted serially tenfold. The dilutions were plated on THB plates to enumerate bacterial CFU.

To collect CSF, mice were deeply anaesthetized through intramuscular injection of 10% pentobarbital sodium. After blood was collected, the mouse was laid prone, the foramen magnum was punctured with a capillary and CSF was collected with gentle elevation.

To collect larger volumes of blood in the SEZ$_{STAMP}$ screen, cardiac perfusion was performed after pentobarbital anaesthesia. PBS was injected into the left ventricle, and blood was collected from the right atrium until the outflow of the liver completely turned pale.

The mice were euthanized via isoflurane inhalation followed by cervical dislocation at the end of the study or when they showed clinical signs of moribundity. All figures depicting the organ burden of mice show the geometric mean and geometric standard deviation.

## In vivo bioluminescence imaging

Bioluminescence imaging of infected mice was performed as described with minor modifications[37]. Before the challenge, the mice were anaesthetized intramuscularly using 10% pentobarbital. Hair was removed from the back and head using a trimmer, followed by a mild depilatory cream. C57BL/6J mice were infected with $5 \times 10^6$ SEZ$_{luc}$ i.v. injection via the tail vein. Luciferin signal detection was recorded at 18 hpi and in moribund mice (~36 h). The substrate D-luciferin (NJDULY) was injected by intraperitoneal injection at a dose of 15 mg kg$^{-1}$ per mouse. For imaging, mice were anaesthetized by intramuscular injection and placed in the in vivo imaging system (NightSHADE, v2.0) for 2 min of exposure time. The mice were euthanized via isoflurane inhalation followed by cervical dislocation at the end of the study or when they showed clinical signs of moribundity.

## Indirect immunofluorescence antibody test

Mice were deeply anaesthetized via intramuscular injection of 10% pentobarbital sodium. The brains were then fixed in 4% paraformaldehyde for 24 h and processed into paraffin sections. After deparaffinization, the sections underwent antigen retrieval in 0.01 M sodium citrate buffer (pH = 6.0) and endogenous peroxidase and were blocked with 3% H$_2$O$_2$. A custom-made monoclonal antibody against the membrane protein SzM (anti-SzM monoclonal antibody) was diluted 1:500 in PBST and used as the primary antibody. Goat anti-mouse IgG H&L (Alexa Fluor 488; Abcam, ab150113) diluted 1:1,000 in PBST served as the secondary antibody. Sections were mounted with ProLong Diamond Antifade Mountant with DAPI (Invitrogen, P36966) and imaged using fluorescence microscopy (Zeiss, Axio Observer).

## H&E staining and histopathological scoring

Mice were deeply anaesthetized via intramuscular injection of 10% pentobarbital sodium and perfused transcardially with 4% paraformaldehyde until the liver turned pale. The brains were then fixed in 4% paraformaldehyde for 24 h, dehydrated through a graded ethanol series, cleared in xylene and embedded in paraffin. The sagittal plane was used to make paraffin sections.

All H&E-stained slides were scanned using a high-resolution digital slide scanner. Regions of interest were selected and cropped using QuPath software (v0.4.4). Histopathological scoring was based on the severity of haemorrhage and inflammatory cell infiltration in the meninges, choroid plexus, cerebral parenchyma and cerebellar parenchyma. The choroid plexus was additionally evaluated using the vacuolar degeneration scoring standard. (See Supplementary Table 9 for scoring details). A trained pathologist blinded to the experimental groups independently performed the scoring, ensuring the objectivity and reproducibility of the pathological assessment. Full-resolution scanned images are publicly available at Mendeley Data (https://data.mendeley.com/datasets/fkg44432ct/1).

## Penicillin treatment

Mice were initially infected intravenously with either the WT or Δ*manY* strain as previously described. Based on an average mouse weight of 16–18 g, a 200-µg dose of penicillin was administered intravenously at 24 hpi. Thereafter, 200 µg of penicillin was injected intravenously every 12 h until the mice succumbed to the infection.

The end-point for each mouse was recorded. Mice that survived the full course of treatment were euthanized at 6 dpi, 1 day after all penicillin-treated, WT-infected mice had succumbed. CFU counts were subsequently determined in all tissues from the surviving mice.

## Construction of the SEZ$_{STAMP}$ library

A STAMPR library was constructed in *S. zooepidemicus* with modifications from a previously described STAMPR library construction[21,38]. A 127-bp fragment of *galU* from *S. zooepidemicus* ATCC35246 was amplified with primer galU 127bp-F and galU 127bp-R. In addition, a 1,022-bp fragment was amplified from pMar4s with primer Kan 20N-F and Kan 20N-R, containing 20-bp random nucleotides. These fragments were inserted into pSET4s at EcoRI and BamHI sites, yielding pSET4s-STAMPR$_{127}$. The ligation product was transformed into DH5a and plated on LB (Kan$^+$, Spc$^+$), yielding over 10,000 colonies in the cloning reaction. Ten colonies were Sanger sequenced individually to confirm that each colony had a unique barcode.

Plasmids were extracted from total colonies grown on plates described above. Subsequently, 2.5 µg of plasmids was added to 80 µl of competent *S. zooepidemicus* and the mixture was kept on ice for 30 min. Cells were transferred to a 1-mm cuvette and pulsed at 2.30 kV, 200 Ω and 25 µF using the GenePulser X cell electroporation system (Bio-Rad). Then, 1 ml of THB was added to the cuvette immediately and bacteria were recovered at 28 °C for 3 h. The mixture was then spread onto 10 THB (Kan$^+$, Spc$^+$) plates.

A total of 2,112 colonies were picked and cultured in 200 µl THB (Kan$^+$, Spc$^+$) at 28 °C overnight. Subsequently, 1:100 dilutions of cultures were passaged successively 3 times at 37 °C for 18–24 h. All bacteria that grew at 37 °C after the passages were subjected to Sanger sequencing by primer P3 and P4. Cultures with single unique barcodes were collected and frozen in THB with 25% glycerol. Finally, 652 individual colonies were collected and aliquoted to 1 ml per tube stored at −80 °C.

To quantify the CFU of the frozen SEZ$_{STAMP}$ library, one tube of the library was thawed and resuspended in PBS to achieve a concentration of approximately $5 \times 10^7$ CFU ml$^{-1}$.

The SEZ$_{STAMP}$ library contains ~652 unique barcodes. Barcode sequences are provided in Supplementary Table 10.

## SEZ$_{STAMP}$ library stability test

To assess whether plasmid integration affected *S. zooepidemicus* growth, nine different SEZ$_{STAMP}$ strains were individually grown in a honeycomb plate with and without kanamycin and spectinomycin overnight. OD$_{600}$ was measured every 30 min using BIOSCREEN C. To assess barcode stability in vitro, the library was also cultured overnight in the absence of kanamycin. Then, 1:1,000 dilutions of the culture were serially passed overnight for 7 days. After each overnight passage, cultures were spread onto THB plates with or without kanamycin and incubated at 37 °C overnight for 1 day, 2 days, 3 days and 7 days. To

assess barcode stability in vivo, $5 \times 10^5$ CFU SEZ$_{STAMP}$ was intravenously inoculated into C57BL/6J mice. The mice were monitored and euthanized when they reached a moribund stage. Selected organs were collected, homogenized and subsequently plated on THB with or without kanamycin. The stability of the SEZ$_{STAMP}$ library is calculated by the ratio of log(CFU growth on THB with kanamycin) to log(CFU growth on THB lacking kanamycin).

Using the kanamycin resistance gene as a proxy for the presence of the barcode, we showed that the inserted barcode oligos in the SEZ$_{STAMP}$ genome were stable in vitro (at least seven passages), as well as in vivo after i.v. injection of mice (Extended Data Fig. 2a). Moreover, the growth of select barcoded strains from SEZ$_{STAMP}$ was not influenced by the insertion of oligos in vitro, with or without antibiotic selection (Extended Data Fig. 2b).

### Creation of the SEZ$_{STAMP}$ calibration curve

The barcoded library was diluted to concentrations ranging from $10^1$ cells to $10^5$ cells in 10-fold increments. Each dilution was spread on THB (Kan$^+$) at 37 °C for 12–14 h. Three replicates were prepared for each dilution. The cells from each plate were scraped separately and stored in PBS with 25% glycerol (PBSG) at −80 °C for future use. We plated defined fractions of an in vitro culture of the SEZ$_{STAMP}$ and compared the number of CFUs observed on the plates with the calculated Ns determined using STAMPR to generate a standard curve. This standard curve had high correlation ($R^2 = 0.9390$) over a range of 4 orders of magnitude, indicating that the SEZ$_{STAMP}$ library could be used to quantify the founding population size from 1 to $10^4$ (Extended Data Fig. 2c).

### In vivo STAMP sample preparation

The survival of mice infected with SEZ$_{STAMP}$ was similar to that of WT *S. zooepidemicus*, indicating that the insertion of barcodes does not impact the virulence of the SEZ$_{STAMP}$ library (Extended Data Fig. 2d). C57BL/6J mice were administered $5 \times 10^6$ CFU of SEZ$_{STAMP}$ intravenously. The mice were euthanized at 6 hpi, 18 hpi and the moribund stage (~36 hpi), with 10 mice at each time point. The brain, liver, spleen, lung, kidney and blood were collected from each mouse. Each organ was homogenized with three 2-mm stainless steel beads for 15 s with a 15-s interval, and this process was repeated for 5 cycles. Blood samples for all experiments were collected via cardiac perfusion after anaesthesia via intramuscular injection of 10% pentobarbital sodium. The samples were cooled on ice, and the homogenization step was repeated. The homogenates were plated on THB (Kan$^+$) plates and incubated at 37 °C for 9–12 h. Bacteria were collected from plates and prepared for sequencing. The barcode frequency of all mice is provided in Supplementary Table 11.

### Sample processing for sequencing analysis

All samples stored at −80 °C were thawed, diluted in 50 µl ddH$_2$O and boiled at 100 °C for 10 min. A 2-µl sample was used as template for PCR in a 50-µl reaction system. PCR cycling conditions were as follows: initial denaturation at 98 °C for 2 min, followed by 25 cycles of denaturation at 98 °C for 10 s, annealing at 55 °C for 10 s and extension at 72 °C for 15 s, with a final extension at 72 °C for 10 min. The PCR was performed in five replicates. Amplified products were run on 2% agarose gel, and the bands at ~350 bp were purified. Purified amplicons were quantified using a Qubit™ 4 fluorometer before sequencing. The STAMP sequencing data were analysed using the pipeline (https://github.com/hullahalli/stampr_rtisan/tree/main/STAMPR_Scripts)[21] based on R v3.8.3. All STAMP data were submitted to the NCBI database under accession code BioProject ID PRJNA1132155.

### Explanation of the STAMP terms

Comparisons of the number of barcodes in a tissue relative to the inoculum enable quantification of the bacterial FP, defined as the number of cells from the inoculum that gives rise to the bacterial population in an organ or tissue sample. The founders represent the bacterial clones that successfully passed through host bottlenecks, which are the chemical, physical and immune barriers that restrict the bacterial population; larger FP values indicate wider infection bottlenecks. In the STAMPR pipeline, the FP is estimated with calculation of the Ns, a simulation-based statistic that determines the sampling depth required to observe a specific number of unique barcodes in an output sample given the distribution of barcodes in the inoculum.

### Construction of the SEZ$_{Tn}$ library

A transposon (Tn) library was constructed in *S. zooepidemicus* using the pMar4s plasmid with modifications[39]. The suicide shuttle plasmid pMar4s was constructed in a previous study[40]. pMar4s is constructed by fusing the pSET4s plasmid with a mariner-based transposon from pMarA. The fragment from pSET4s confers temperature sensitivity and Spc resistance to pMar4s. The pMarA fragment provides pMar4s with the Himmar C9 transposase and transposon carrying the Kan resistance gene. Once transposition occurs, the Kan resistance gene will be inserted into the 5′-TA-3′ in the *S. zooepidemicus* genome.

The pMar4s plasmid was introduced into competent *S. zooepidemicus* by electroporation at 2.30 kV, 200 Ω and 25 µF using the GenePulser X cell electroporation system (Bio-Rad). Transformants were recovered following SEZ$_{Tn}$ construction, and the transformed cells were plated onto THB (Kan$^+$, Spc$^+$) plates at 28 °C for 12–14 h to collect large amounts of *S. zooepidemicus* carrying pMar4s. More than 10 colonies were picked into 200 µl PBSG per tube.

Colonies from each tube were then diluted, and 10 µl was spotted on THB, THB (Kan$^+$) and THB (Spc$^+$) plates for 14–16 h at 37 °C. The CFU on THB plates indicate the total live cell counts. The CFU on THB (Kan$^+$) plates represent the counts of cells with transposon insertions, while the CFU on THB (Spc$^+$) plates indicate the counts of the cells containing the pMar4s plasmid. The integration frequency was calculated as the ratio of the number of Spc colonies to the total number of colonies on the THB plate. The insertion frequency was calculated as the ratio of the number of Kan colonies to the total number of colonies on the THB plate. Tubes with integration frequency less than 0.1% and the insertion frequency less than 1% were selected as candidate for further processing[41].

The candidates were spread on THB (Kan$^+$) at an anticipated concentration of $1 \times 10^5$ to $3 \times 10^5$ colonies per 625-cm$^2$ square plate. Each square plate corresponded to one tube. After being cultured at 37 °C for 12–14 h, over $3.4 \times 10^6$ colonies were collected in THB with 25% glycerol and stored at −80 °C. The SEZ$_{Tn}$ library contains ~90,000 unique insertions, representing ~62% of all possible insertion sites (Extended Data Fig. 3a).

The construction and screening data of the SEZ$_{Tn}$ library were all submitted to the NCBI database under accession code BioProject ID PRJNA1132128.

### SEZ$_{Tn}$ screen for genes required for CNS infection

The SEZ$_{Tn}$ library was recovered in THB media at 37 °C for 1 h before injection, and $5 \times 10^6$ CFU were spread on plates to serve as the input control. After the mice have been anaesthetized intramuscularly using 10% pentobarbital, $5 \times 10^6$ CFU were directly injected into the lateral ventricles using digital mouse stereotaxic instruments (bregma as the origin, $X = 1.5$ mm, $Y = -1.7$ mm, $Z = 2.0$ mm). Brains from moribund mice (~13–18 hpi) were homogenized and plated on THB (Kan$^+$), and the bacterial growth on these plates was collected as an output library. This output library was injected into the lateral ventricles of mice for subsequent rounds of screening, for a total of three rounds.

Genomic DNA (25 µg) from input and output libraries was extracted using the MiniBEST Bacteria Genomic DNA Extraction kit (Takara, number 9763), following the manufacturer's instructions. The genomic DNA was diluted to a volume of 130 µl and sheared to 200–800-bp fragments by sonication (Covaris) at 50 W peak incident

power, 10% duty factor, 200 cycles per burst and 70-s treatment time. After shearing, the DNA was blunted with the Quick Blunting Kit (NEB, E1201) followed by ligation of Illumina adaptors (fork truncated NH2 primer: TACCACGACCA NH2-3′ and index fork primer: GTGACTG-GAGTTCAGACGTGTGCTCTTCCGATCTGGTCGTCGTGGTAT). DNA containing adaptors were then amplified with two steps of PCR; the first step amplified the fragment with Tn insertion (Himar 3 out A: CGCAACTGTCCATACTCTG and Index PCR R primer: GTGACTG-GAGTTCAGACGTGTG), and the second step added Illumina P5 and P7 sequences (P5 and P7 primer). The final DNA libraries were gel purified on a 2% agarose gel, where DNA fragments ranging from 300 bp to 500 bp were extracted for further sequencing. Sequencing was performed on Illumina HiSeq, and data analysis was conducted as described in a previous study[42]. Briefly, the adaptor sequences were trimmed with Cutadapt (v3.4). The trimmed sequences were mapped to the *S. zooepidemicus* ATCC35246 genome by Bowtie (v1.3). The python scripts were used to convert mapping files to a table of read count per insertion[43]. Finally, the ARTIST pipeline was used to identify conditionally essential genes[44].

The insertion counts of three rounds of screening at all TA sites are provided in Supplementary Table 12. The fold changes of all genes in three rounds of screening are provided in Supplementary Table 1.

### Carbohydrate utilization tests

The ability of *S. zooepidemicus* to grow on various carbohydrates was examined using the API® 50 CHL test following the manufacturer's protocol (Biomerieux S.A.). *S. zooepidemicus* was cultured on THB to mid-exponential phase and washed three times with PBS. The bacteria were diluted with a turbidity equivalent to 2 McFarland in an ampoule of API® 50 CHL medium (number 50410). The suspension was transferred to the reaction region to rehydrate the carbohydrate substrates. Results were observed at 12 hpi, 24 hpi and 36 hpi. The ability of *S. zooepidemicus* to ferment carbon sources was indicated by a pH-dependent colour change of the CHL medium, reflecting the acidification of the growth medium during fermentation.

### Standard PEP–PTS activity assay

PTS activities were assayed in permeabilized cells using a method of Kornberg and Reeves with modifications[24]. When the PTS system transports carbohydrates from the environment to the intracellular space, the phosphate from PEP is transferred to carbohydrates via EI components of PTS. This process converts PEP to pyruvic acid, which produces lactic acid through lactate dehydrogenase, which in turn converts NADH to NAD. Therefore, PTS activity is quantified by measuring the reduction of NADH.

The cells were collected at mid-exponential phase and washed twice with 0.1 M sodium–potassium phosphate buffer (pH = 7.2), containing 5 mM MgCl$_2$, before the pellet was frozen at −40 °C until use. Pellets were suspended in the ice-cold sodium–potassium phosphate buffer at 0.4 OD$_{600}$ per ml and mixed vigorously. Then, 50 μl of toluene–acetone (1:9) was added per ml of cell suspension and mixed for 2 min, repeated 3 times. The sample was kept on ice between vortexing steps to prevent the degradation of proteins. Samples were kept on ice until use in the assay.

Toluene-treated cells were added to 96-well plates. Each reaction contained 1 mol of PEP, 0.1 mol of NADH, 2 units of lactate dehydrogenase and sufficient buffer to yield a final volume of 180 μl. The plate was incubated at 30 °C for thermal equilibrium (15–25 min) to consume endogenous carbohydrate. OD$_{340}$ was measured in the Infinite 200 PRO microplate plate reader (TECAN), after adding 20 μl or 500 μM carbohydrate substrate. A control reaction lacking carbohydrate was included. Activity was calculated by correcting experimental readings for control activity using the molar extinction coefficient of NADH ($6.22 \times 10^3$ l mol$^{-1}$ cm$^{-1}$). The concentration of proteins was measured by the BCA Protein Qualification Kit (Vazyme, E112-02).

### Determination of β-galactosidase activity

The pTCV-lacZ plasmid is a recombinant β-galactosidase reporter used to study promoter activity. It contains the lacZ gene, which encodes β-galactosidase, an enzyme that cleaves lactose to produce a detectable colorimetric signal in the presence of a substrate. To assess the transcriptional activity of different promoters, plasmids were constructed by inserting the upstream ~500 bp of the start of the coding sequences into pTCV-lacZ, and empty plasmid pTCV-lacZ served as a control. Each promoter consists of the predicted −35 and −10 region. To determine β-galactosidase activity in cell extracts, cultures were grown in CDM with 2 g l$^{-1}$ glucose to mid-exponential phase. Then, 2 ml of the cultures was collected by centrifugation. The cells were washed and resuspended in 1/10 initial volume of Z-buffer with β-mercaptoethanol (35 μl per ml Z-buffer), and cells were lysed by adding 6.5 μl 0.1% SDS and 1.25 μl chloroform. The mixture was incubated at 30 °C for 5 min. Then, 100 μl *o*-nitrophenol-*b*-D-galactopyranoside (4 mg ml$^{-1}$) was added to assay β-galactosidase activity at 30 °C, and 250 μl 1 M Na$_2$CO$_3$ was added to terminate the reaction. The supernatants were moved to 96-well plates and recorded at 420 nm and 562 nm after centrifugation at 13,000 × *g* for 10 min. Galactosidase activity was measured in Miller units (MU) and calculated as follows: MU = 1,000 × (OD$_{420}$ − 1.75 × OD$_{550}$)/($T \times V \times$ OD$_{600}$), where *V* indicated the total sample volume and *T* indicated the reaction time.

### Growth curves

Bacteria were cultured in THB to mid-exponential phase. Cells were washed with PBS 3 times and diluted to OD$_{600}$ = 1.0. The cells were added to 300 μl of select cultured medium in HONEYCOMB® sterile microplates at a dilution ratio of 1:100. OD$_{600}$ was measured every 30 min using BIOSCREEN C. For the growth curve in CSF, 20 μl of culture was taken at a selected time point, and 3 μl of culture was used to measure OD$_{600}$ by using Eppendorf μCuvette® G1.0.

For growth curves quantified with CFU counts, we cultured cells in 2 ml medium with the same treatment as measured by BIOSCREEN C above. Every 2 h, 20 μl of culture was taken, serially diluted and spread on THB plates to count live cells at selected time points.

### Quantification of glucose concentrations in culture medium and CSF

Bacteria were cultured in CDM with different concentrations of glucose. At selected time points, 100 μl of cultured bacteria was collected. The supernatant was collected in 96-well plates after centrifugation at 13,000 × *g* for 2 min and stored at −20 °C until used for assay. Then, 2 μl of the supernatant and CSF sample was used to analyse the glucose concentration using the Hexokinase Method Glucose Kit (JianCheng Bio, A154-2-1). Glucose in the samples is first converted to glucose-6-phosphate by incubating in 200 μl of Buffer R1 (containing hexokinase) at 37 °C for 3 min. Subsequently, glucose-6-phosphate is oxidized by NAD$^+$ through the action of glucose-6-phosphate dehydrogenase, following the addition of 50 μl of Buffer R2. The formation of NADH from NAD$^+$ is measured at an absorbance of 340 nm (OD$_{340}$), which allows calculation of glucose concentration in the sample. The detectable range of glucose in the assay is 2.2–15 mmol l$^{-1}$, and thus, high-glucose samples were diluted before use in the assay.

### Protein expression and purification

His-tagged proteins were overexpressed in BL21 (DE3) and purified by immobilized metal affinity chromatography using TALON metal affinity resin. Bacteria containing pET28a constructed plasmids were grown in LB (Kan$^+$) at 37 °C until OD$_{600}$ to 0.6–0.8 and induced with 1 mM isopropyl β-D-1-thiogalactopyranoside (IPTG) for 5 h at 37 °C. Bacteria containing pcoldI constructed plasmids were induced for 20 h at 16 °C. The collected cells were resuspended in binding buffer (500 mM NaCl, 50 mM Tris base, 0.01% Tween-20, pH = 8.0) with PMSF (Biosharp, number BS071B). After lysis by ultrasonication, the soluble protein

was bound to a HisTrap™ HP column (Cytiva, number 17524801) on the ÄKTA system (GE Healthcare). Imidazole was removed using suitable kDa Amicon® Ultra 50 ml centrifugal filters (Millipore, UFC901024).

## RNA extraction and quantitative real-time reverse transcription PCR

RNA extraction and quantitative real-time reverse-transcription PCR (qRT-PCR) were performed as previously described[1]. Bacteria were cultured in CDM to mid-exponential phase. The bacteria were then incubated in 100 µl lysozyme (50 mg ml$^{-1}$) at 37 °C for 15 min and frozen in 1 ml Trizol (RNAiso Plus, Takara) at −80 °C until use. For RNA extraction, samples were homogenized in tubes containing 0.1–0.2-mm glass beads and for 4 min (15 s with a 15-s interval). Then, 200 µl of chloroform was added per ml Trizol, and samples were vortexed vigorously. The supernatant was added to an equal volume of 70% ethanol and mixed. RNA purification was performed with a E.Z.N.A.® Total RNA kit I (Omega, R6934-02) using the manufacturer's recommendations. The purity and integrity of RNA were verified by 2% agarose gel electrophoresis and Experion Automated Electrophoresis System (Bio-Rad Laboratories).

RNA from each sample was converted to cDNA using the HiScript III RT SuperMix for qPCR (Vazyme, R323-01). cDNA was diluted 20-fold and subjected to real-time PCR amplification using AceQ qPCR SYBR Green Master Mix (Vazyme, Q141-02) with specific primers (Supplementary Table 8). Each group comprises three technical replicates. Data were normalized to a reference housekeeping gene (*recA*).

## RNA-seq analysis

Strains were grown in CDM with different concentrations of glucose and collected at mid-exponential phase. Total RNA was isolated as described above and used to construct the library following removal of ribosomal RNA. qRT-PCR was used to quantify the concentration of the library. Libraries were sequenced by Illumina NovaSeq PE-150.

The clean reads were aligned with the reference genome (*S. zooepidemicus* ATCC35246) set by Bowtie2 (v2.2.5). FeatureCounts (v1.5.0-p3) was used to count the number of reads mapped to each gene, and the gene expression level transcripts per million value was calculated using RSEM (v1.2.12)[45]. Fold changes for the comparisons of Δ*manY* versus WT *S. zooepidemicus* (1 g l$^{-1}$ glucose), Δ*manY* versus WT *S. zooepidemicus* (10 g l$^{-1}$ glucose) and Δ*relA*Δ*relQ* versus WT *S. zooepidemicus* (10 g l$^{-1}$ glucose) were calculated using the DESeq2 package. Phyper (v3.6.0) was used to determine enriched KEGG pathways.

The regulated gene list related with both growth and stringent response is provided in Supplementary Table 13. RNA-seq clean data were submitted to the NCBI database under accession code BioProject ID PRJNA1131986.

## Quantification of intracellular (p)ppGpp concentration

Cultures at mid-exponential phase were washed with low-phosphate (0.2 mM) CDM-MOPs (40 mM) and transferred to 50 µl of low-phosphate CDM with different concentrations of glucose at a ratio of 1:100. Bacteria were then cultured at 37 °C to OD$_{600}$ = 0.05 and H$_3$$^{32}$PO$_4$ was added to a final concentration of 100 µCi ml$^{-1}$. Bacteria were continuously cultured at 37 °C for 3 h. Samples were quenched with 10 µl of ice-cold 13 M formic acid and freeze-thawed in liquid nitrogen 5 times. The sample was thawed at room temperature and centrifugated at 13,000 × *g* for 10 min. Then, 10 µl of the supernatant was analysed by TLC plates (Merck, Sigma). Nucleotides were resolved in 1 M potassium phosphate (pH = 3.4). After drying, the plates were placed overnight on the MS storage Phosphor Screen (GE Healthcare). The $^{32}$P signal on the phosphor screen was visualized using Amersham Typhoon (Typhoon FLA 9500 GE Healthcare). Relative intensity of (p)ppGpp was quantified by ImageJ 2.14.0. The experiment was repeated independently at least two times, with similar results obtained in each trial.

## Cellular imaging of (p)ppGpp

The imaging procedure was performed with modifications based on previously described methods[46]. The S2 sensor plasmid was reconstructed in pSET2 vector with CP25 promoter and its terminator, enabling expression in *Streptococcus*. Bacteria containing the (p)ppGpp RNA sensor plasmid (pSET2-S2) were grown in THB (Spc$^+$) at 37 °C to mid-exponential phase. The bacteria were then washed three times with PBS and the OD$_{600}$ was adjusted to 1.0, followed by incubation with 200 µM (5Z)-5-[(3,5-difluoro-4-hydroxyphenyl)methylene]-3,5-dihydro-2-methyl-3-(2,2,2-trifluoroethyl)-4H-imidazol-4-one (DFHBI-1T, Aladdin, D288318) in PBS at 37 °C for 30 min. After incubation, 10 µl of the bacterial suspension was fixed onto a glass slide. All sections were mounted using ProLong™ Diamond Antifade Mountant with DAPI (Invitrogen, P36966) and imaged using fluorescence microscopy (Zeiss, Axio Observer). The image exposed at 100 ms with 30% shift. All images were handled under the same parameter by ImageJ 2.14.0.

## EMSA

EMSA was performed as previously described[10]. The DNA fragments used in EMSAs were amplified by PCR (Supplementary Table 8), in which products were purified using a E.Z.N.A.® Gel Extraction Kit (Omega, D2500-02) and diluted to 80 ng µl$^{-1}$. The purified protein (1–2 µg) with or without cAMP was incubated with the DNA fragment (80 ng) in binding buffer (20 mM Tris–HCl (pH = 7.5), 30 mM KCl, 1 mM DTT, 1 mM EDTA (pH = 7.5), 10% (v/v) glycerol)) in a final volume of 20 µl for 30 min at 37 °C. Samples were loaded on native 8% polyacrylamide gel and electrophoresed in 0.5 × TBE (44.5 mM Tris, 44.5 mM boric acid, 1 mM EDTA, pH = 8.0) under 150 V with the ice bath for 2–2.5 h. Gels were stained in GoldView nucleic acid stain for 10 min, and the results were recorded under the UV Trans illumination by Gel Doc XR (Bio-Rad).

## Determination of HPr and P-HPr

Sample preparation was performed as previously described with modification[28], and three independent biological replicates were performed. Bacteria were grown to mid-exponential phase and washed with Tris-EDTA buffer (50 mM Tris–HCl, 50 mM EDTA, pH = 7.8) 3 times. Bacterial pellets (OD$_{600}$ between 40 and 50) were resuspended in 1 ml Tris-EDTA buffer with 0.01 M PMSF and transferred to Lysing Matrix (MP) for homogenizing (15 s on, 15 s off, 5 cycles repeated 4 times). Samples were kept on ice between homogenization cycles to maintain a low temperature. After homogenization, samples were centrifuged at 13,000 × *g* for 30 min at 4 °C. Then, 2 µg HPr was phosphorylated by adding 2 mM PEP and 1 µg of EI in Buffer A (10 mM phosphate; 2 mM MgCl$_2$; 1 mM EDTA; 10 mM KCl; 5 mM DTT), as an appropriate positive control included for comparison.

The supernatant was collected, and total protein concentrations were measured using a BCA Protein Quantification Kit (Vazyme, E112-02) before gel loading. Equal amounts of total protein from cell extracts were mixed with native sample loading buffer and separated on a 4–12% SurePAGE™ Precast gel (GenScript, M00652) in Tris-MOPS Native Running Buffer (GenScript, M00727C). Proteins were then transferred onto a polyvinylidene fluoride membrane and immunoblotted with specific rabbit polyclonal antibodies against HPr (CUSABIO, CSB-PA362538HA01BRJ, 1:2,000) at 4 °C overnight. An HRP-labelled goat anti-rabbit antibody was used as the secondary antibody at a 1:5,000 dilution. The membranes were treated with the SuperSignal™ West Atto ECL Chemiluminescence Kit (Thermo, A38555), and chemiluminescence signal was collected by using the ChemiDoc Touch Imaging system (Bio-Rad).

## Determination of cellular Crp levels

Sample preparation followed the same procedure as described for HPr determination, and three independent biological replicates were performed. Cell extracts were mixed with SDS-loading buffer, boiled for 10 min and then centrifuged at 13,000 × *g* for 2 min at

4 °C. Equal concentrations of total protein were loaded onto 4–12% SurePAGE™ Precast gel (GenScript, M00652) in Tris-MOPS SDS Running Buffer Powder (GenScript, M00138). Proteins were transferred onto a polyvinylidene fluoride membrane and immunoblotted using specific rabbit polyclonal antibodies against PrfA(Crp) (CUSABIO, CSB-PA325257XA01LPY, 1:2,000) or rabbit polyclonal antibodies against GroEL (made by our lab, 1:1,000) at 4 °C overnight. PrfA belongs to the Crp/Fnr family of transcriptional activators in *Listeria monocytogenes*, and sequence comparison shows that it shares over 70% homology with Crp (encoded by *SESEC_01595*) in *Streptococcus*. GroEL is a chaperonin protein commonly found in bacteria, often used as loading control in *Streptococcus*. HRP-labelled goat anti-rabbit antibodies were used as the secondary antibodies at a dilution of 1:5,000. The membranes were treated with the SuperSignal™ West Atto ECL Chemiluminescence Kit (Thermo, A38555), and the chemiluminescence signal was collected by using the ChemiDoc Touch Imaging system (Bio-Rad).

## Statistics and reproducibility

Unless otherwise specified, statistical analyses were performed using two-way ANOVA for multi-factor experiments and one-way ANOVA for single-factor experiments, each followed by Dunnett's multiple-comparison test. The in vivo CFU data were log transformed, and the normality assumption of ANOVA models was assessed using residual QQ plots. All log-transformed in vivo CFU data satisfied the normality assumption required for ANOVA. All CFU data are presented as geometric mean ± geometric s.d., and all other quantitative data are presented as mean ± s.d. unless otherwise specified.

Growth curves were modelled using a Gompertz nonlinear regression model, and the estimated parameters were compared across groups by one-way ANOVA, with Dunnett's multiple-comparison test. Survival curves were estimated using Kaplan–Meier methods, and group differences were tested with the log-rank test, using the WT group as the reference. The *t*-test was used in Tn-seq analysis to compare normalized reads between the average number of mapped reads for the input and output library.

No statistical methods were used to predetermine sample sizes, but our sample sizes are similar to those reported in previous publications[47,48]. Data were randomly collected when applicable. The animals and samples were randomly assigned to experimental groups to ensure unbiased allocation. Data collection and analysis were not performed blind to the conditions of the experiments, except for histopathology scoring. All reported *n*s indicate biological replicates derived from different single clones or mice, with no animals or data points excluded from the analyses.

## Reporting summary

Further information on research design is available in the Nature Portfolio Reporting Summary linked to this article.

## Data availability

The raw files for RNA-seq data generated in this study have been deposited in the NCBI database under accession code BioProject ID PRJNA1131986. The raw files for Tn-seq data generated in this study have been deposited in the NCBI database under accession code BioProject ID PRJNA1132128. The raw files for STAMP data generated in this study have been deposited in the NCBI database under accession code BioProject ID PRJNA1132155. The scanned HE-stained slide files are available on Mendeley Data and can be downloaded from the following link: https://data.mendeley.com/datasets/fkg44432ct/1. Source data are provided with this paper.

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

## Acknowledgements

This study was supported by the National Key Research and Development Program of China (2021YFD1800800) to Z.M., Natural Science Foundation of Jiangsu Province (BK20250095) to Z.M., the Fundamental Research Funds for the Central Universities (KJJQ2025003) to Z.M., the National Natural Science Foundation of China (32273009, 31973004) to Z.M., Jiangsu Agriculture Science and Technology Innovation Fund (CX(23)1041) to Z.M. and the Priority Academic Program Development of Jiangsu Higher Education Institutions to Z.M. M.K.W. is supported by HHMI and RO1-AI-042347. C.Y. is supported by the Postgraduate Research and Practice Innovation Program of Jiangsu Province (KYCX24_0996). We are grateful to Y. Lü for his contribution to the histopathological analysis and to D. A. Yang for his insightful guidance on the statistical analysis. We also appreciate the Radioisotope Laboratory of the College of Science at Nanjing Agricultural University for providing the venue for radioactive isotope experiments.

## Author contributions

All authors contributed to this paper. Z.M., M.K.W., H.F., K.H. and C.Y. designed the research study. C.Y., Y.W., X.T., X.L., S.Z., W.W., L.X., F.P., Y. Li, Y. Liang and Y.X. conducted the experiments. C.Y., K.H., H.H. and F.P. analysed the data. Z.M., C.Y., K.H. and H.H. conducted the bioinformatic analysis. Z.M., M.K.W., K.H. and C.Y. wrote the paper with contributions from the other authors. Z.M., M.K.W. and C.Y. acquired funding.

## Competing interests

The authors declare no competing interests.

## Additional information

**Extended data** is available for this paper at https://doi.org/10.1038/s41564-025-02194-2.

**Correspondence and requests for materials** should be addressed to Zhe Ma.

**Peer review information** *Nature Microbiology* thanks Thomas Hooven, Claudia Trappetti and the other, anonymous, reviewer(s) for their

contribution to the peer review of this work. Peer reviewer reports are available.

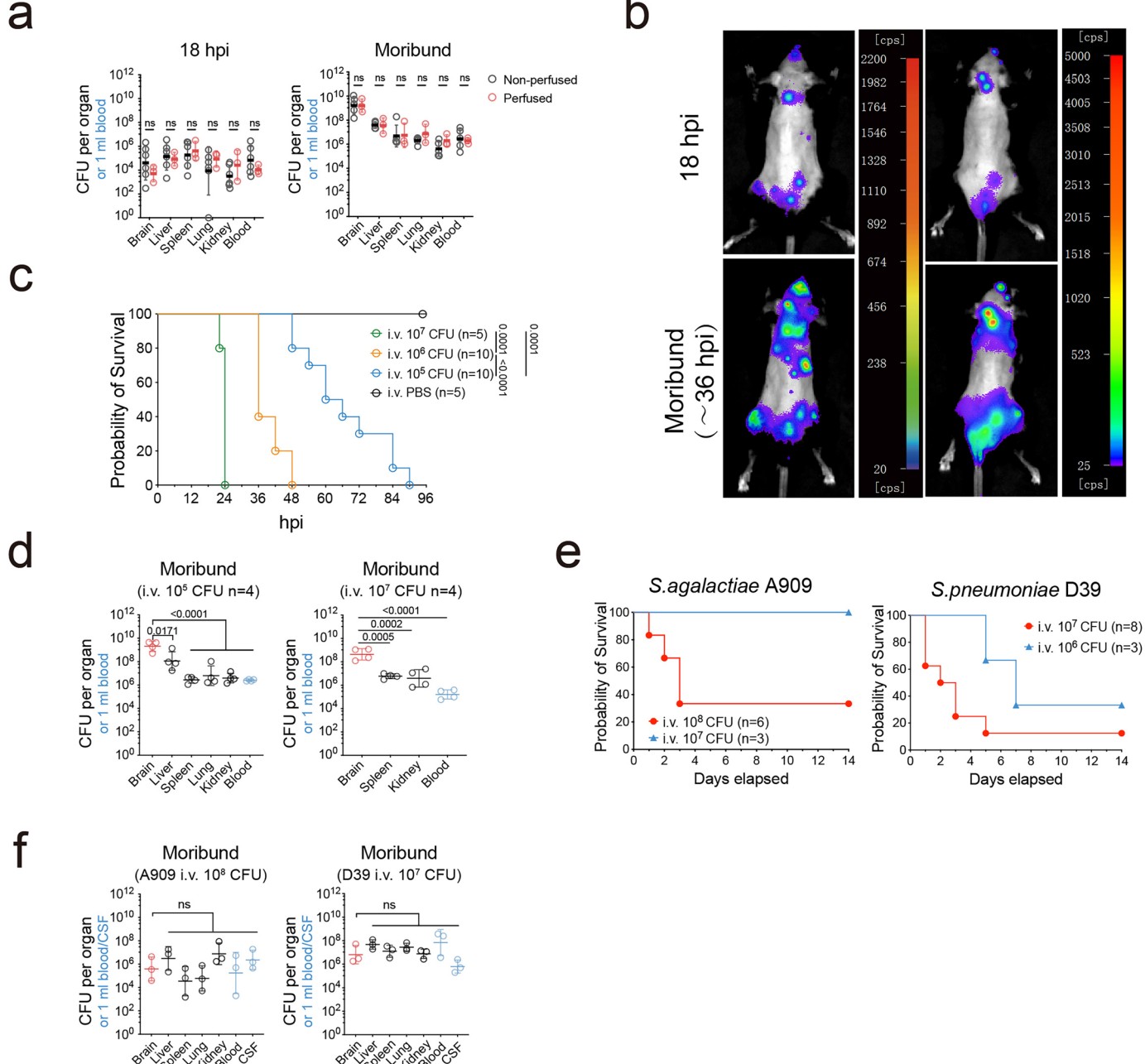

**Extended Data Fig. 1 | *S. zooepidemicus* accumulation in the brain and survival curves. a**, Organ CFU burden at 18 hpi and at moribund stage following tail vein injection of 5×10⁶ CFU *S. zooepidemicus*. Cardiac perfusion was performed until the liver appeared pale (over 20 mL perfusate) or blood was drained as fully as possible after anesthesia with pentobarbital (geometric mean ± geometric s.d.; n = 6 for no-perfusion; n = 3 for perfusion, Two-way ANOVA with Šídák's multiple comparisons test on log-transformed CFU). **b**, C57BL/6 J mice were i.v. injected with 5×10⁶ CFU of the luciferase expressing strain SEZ$_{Luc}$. Two additional replicates of bioluminescence imaging of mice at 18 and ~36 hpi are shown in Fig. 1. Quantification of the bioluminescence signals in counts per second (cps). **c**, Survival curves of C57BL/6 J mice i.v. injected with different doses of

*S. zooepidemicus* and monitored until moribund or 7 dpi. **d**, CFU of each organ in moribund mice following tail vein injection of *S. zooepidemicus* at 5×10⁵ and 5×10⁷ CFU. Mice were moribund between 54 and 72 hpi in 10⁵ dose group, and between 22 and 24 hpi in 10⁷ dose group. **e**, Survival curve of C57BL/6 J mice intravenously injected with *S. agalactiae* A909 (10⁷ and 10⁸ CFU) or *S. pneumoniae* D39 (10⁶ or 10⁷ CFU). **f**, Organ CFU counts in mice i.v. injected with 5×10⁸ CFU A909 (moribund between 24 and 72 hpi, n = 3) or 5×10⁷ CFU of D39 (moribund between 24 and 120 hpi, n = 3). Survival curves (c) were analyzed using the Kaplan-Meier methods following a log-rank test vs. WT. CFU burden (d, f) are presented as geometric mean ± geometric s.d. and analyzed using one-way ANOVA followed by Dunnett's multiple comparisons test on log-transformed CFU.

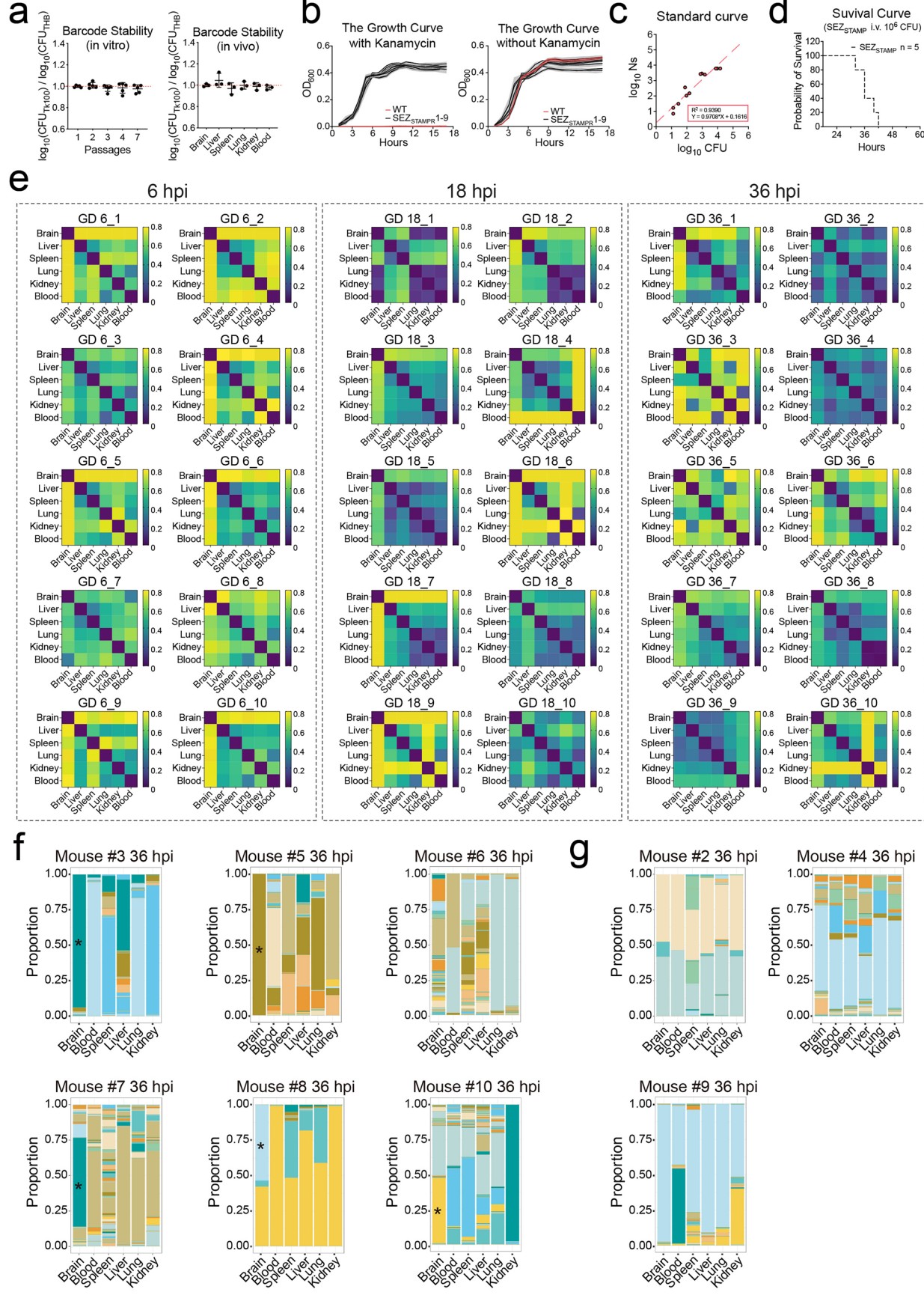

**Extended Data Fig. 2 | See next page for caption.**

**Extended Data Fig. 2 | Stability testing of SEZ$_{STAMP}$ and analysis of GD and barcode frequencies. a**, SEZ$_{STAMP}$ library stability calculated as the ratio of log$_{10}$ (CFU on THB with kanamycin) to log$_{10}$ (CFU on THB without kanamycin). Left: Barcoded library passaged in THB without antibiotic for 7 days (n = 5); The passaged culture quantified by growth on plates with or without kanamycin daily. Right: C57BL/6 J mice i.v. injected with 5×10$^5$ CFU SEZ$_{STAMP}$ library (n = 3). Bacteria from organs of moribund mice (~ 66 h) harvested and grown on plates with or without kanamycin. **b**, Growth curves (n = 3) of 9 randomly selected individual barcoded clones from the library and WT *S. zooepidemicus* cultured with or without Kanamycin. **c**, The standard curve of log$_{10}$Ns (reflecting founding population) and log$_{10}$CFU at different dilutions (that is, *in vitro* bottlenecks) of the SEZ$_{STAMP}$ library (n = 3). Founding population estimates are accurate up to

10$^4$. **d**, Survival curve of C57BL/6 J mice intravenous injected with 10$^6$ SEZ$_{STAMP}$ library. Mice were moribund between 32 and 42 hpi. **e**, Individual heatmaps of genetic distance (GD) of 30 mice at 6, 18 and 36 hpi. Each time point contains data from 10 mice. High GD values indicate dissimilar samples, while low GD values indicate similar samples. **f-g**, Barcode frequencies of 9 mice at 36 hpi. Each colour represents the relative abundance of each barcode. **(f)** Barcode frequencies of 6 mice where barcodes in the brain that are distinct from those in the blood (high GD). The asterisk indicates the dominant barcode in the brain that is less in the blood. **(g)** Barcode frequencies of 3 mice with clones in the brain are more similar to those in the blood (low GD). Total barcode frequency is provided in Supplementary Table S11. Panel a and b presented as mean ± s.d.

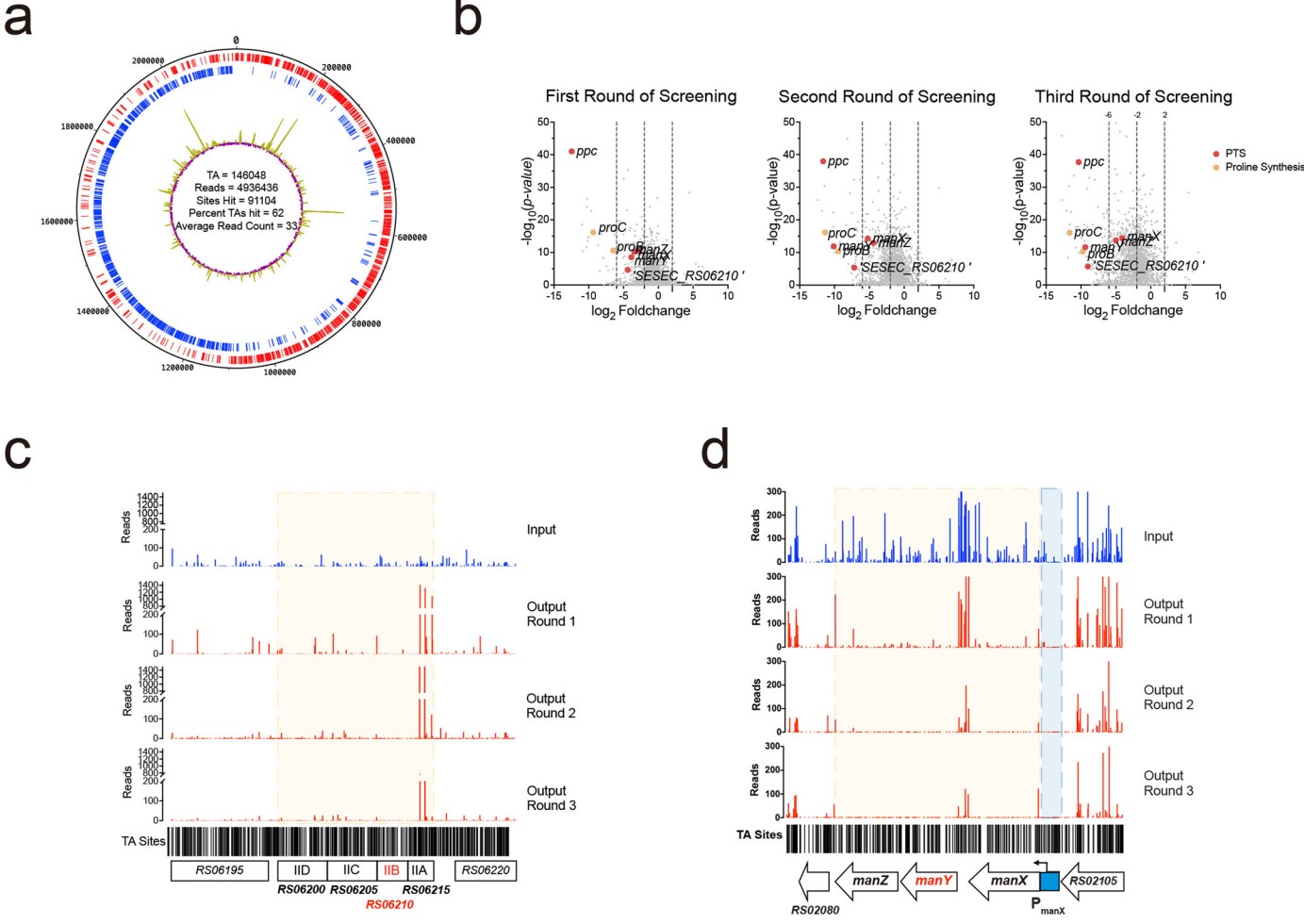

**Extended Data Fig. 3 | Construction and screening of transposon insertion library. a**, Circular visualization of SEZ$_{Tn}$ library. The purple and yellow rings represent unique transposon insertion counts at each TA dinucleotide site. The blue and red rings indicate coding regions on different DNA strands. Total TA site insertion counts are provided in Supplementary Table S12. **b**, Volcano plot of three rounds of SEZ$_{Tn}$ screening in the brain. Red points denote PTS components, and orange points denote select genes involved in proline synthesis. *t-test* was used to compare normalized reads between the average number of mapped reads for the input and output library to derive *P* value. All gene Fold change in three rounds of screening are provided in Supplementary Table S1. **c**, Comparison of the distributions of transposon insertions in another mannose PTS operon (*RS06200 - RS06215*) and the upstream and downstream gene in the input library, as well as in three rounds of the output samples. The *RS06210* gene is marked in red. The yellow background denotes the coding sequence (CDS) and intergenic regions. TA sites indicate potential transposon insertion site.

**d**, Comparison of the distributions of transposon insertions in PTS$_{man}$ operon and the upstream and downstream gene in the input library, as well as in three rounds of the output samples, which reveals underrepresentation of PTS$_{man}$ and its promoter (P$_{manX}$). The *manY* gene is marked in red. The yellow background denotes the CDS and intergenic regions of PTS$_{man}$, and the blue background highlights the promoter regions of PTS$_{man}$. TA sites indicate potential transposon insertion sites.

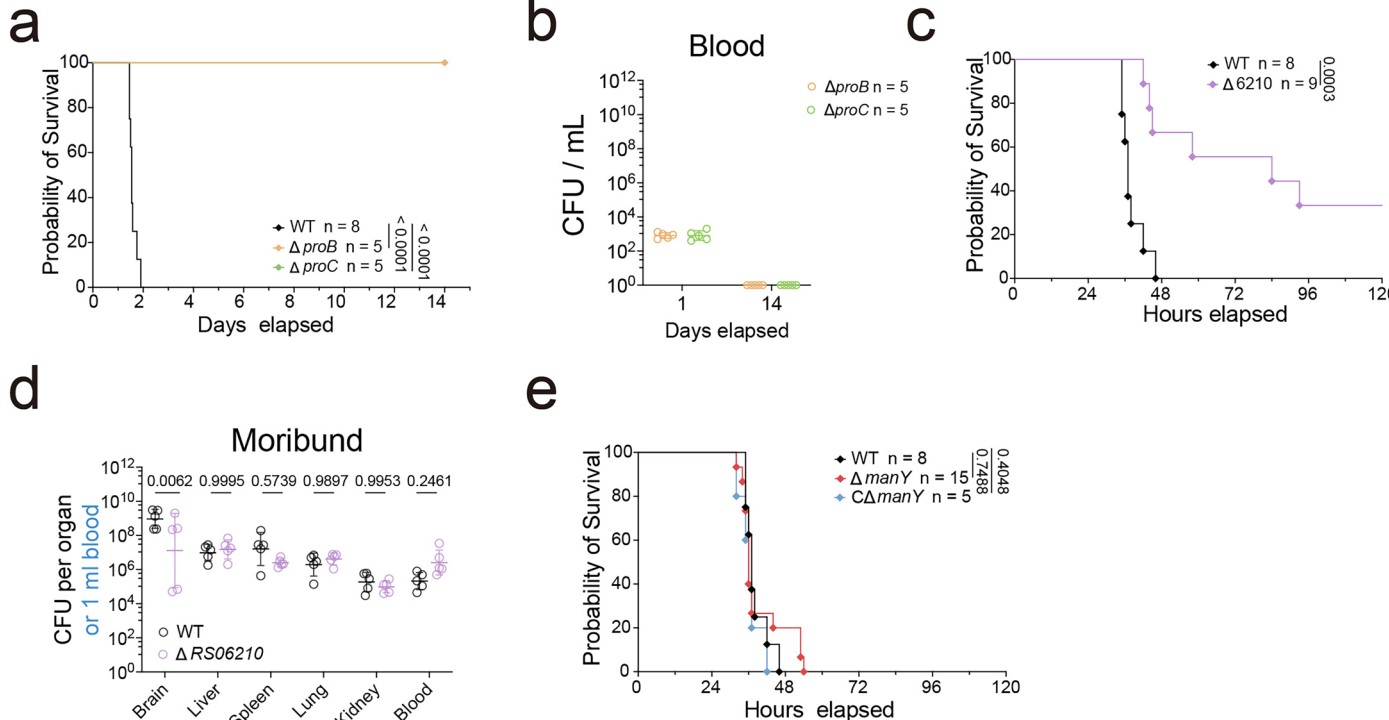

**Extended Data Fig. 4 | Infection phenotypes of other *S. zooepidemicus* mutants. a**, Survival curves of animals challenged with 5×10⁶ CFU of WT, Δ*proB* and Δ*proC*. Animals were monitored until moribund or 14 dpi. **b**, C57BL/6 J mice were intravenously injected with 5 × 10⁶ CFU of either Δ*proB* or Δ*proC*, and bacterial counts in the blood were measured. Δ*proB* and Δ*proC* persists at a low level in the blood and were eliminated at 14 dpi. **c**, Survival curves of animals challenged with 5×10⁶ CFU of Δ6210. Mice were monitored until moribund, or 120 hpi. **d**, Bacterial burden in organs and blood was assessed in moribund mice after i.v. infection with 5×10⁶ CFU Δ*6210* (n = 5) and WT (n = 6). **e**, Survival curves of animals challenged with 5×10⁶ CFU WT, Δ*manY* or CΔ*manY* via i.v. injection through tail vein. CFU data (**b,d**) are shown as geometric mean ± geometric s.d. Survival curves (**a**, **c**, **e**) were analyzed by log-rank test versus WT. CFU burden (**d**) was analyzed by two-way ANOVA followed by Šídák's multiple comparisons test on log-transformed CFU.

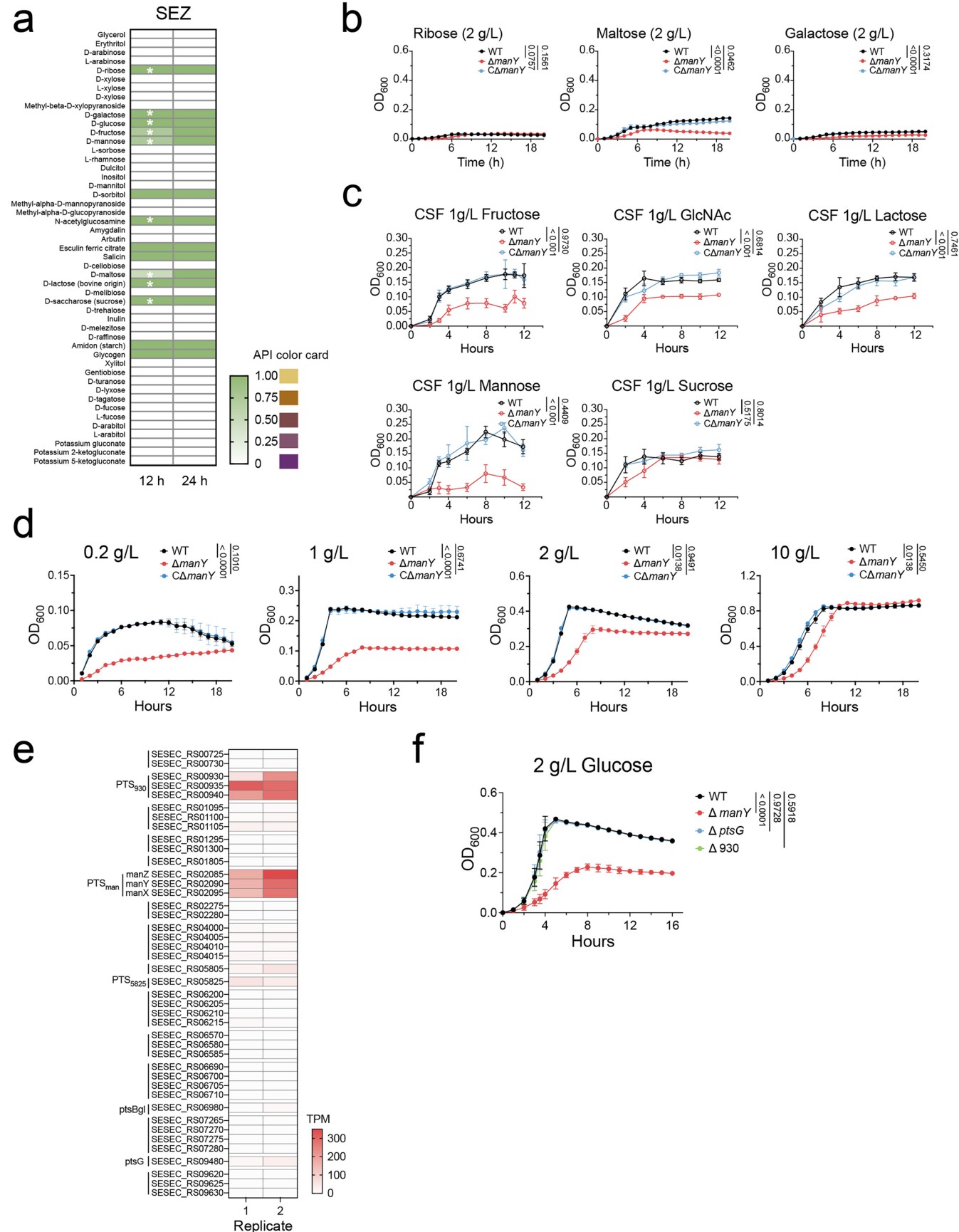

**Extended Data Fig. 5 | See next page for caption.**

**Extended Data Fig. 5 | Carbohydrate utilization of *S. zooepidemicus*.**
**a**, Heatmap demonstrating the ability of *S. zooepidemicus* to ferment 49 carbohydrates. White squares denote carbon sources that *S. zooepidemicus* cannot utilize. Green squares denote utilizable carbon sources, where the colour is scaled by the extent of utilization (fermentation colour change from purple to yellow as the API colour card on the left of the scale). White asterisks indicate the carbon sources present in animals which could be utilized by *S. zooepidemicus*. **b**, Growth curves of WT, Δ*manY* and CΔ*manY* with three non-preferred carbon sources utilizable for *S. zooepidemicus* in CDM (n = 4, mean ± s.d.). **c**, Growth curves of WT, Δ*manY* and CΔ*manY* cultured in swine CSF supplemented with five different carbohydrates *in vitro* (n = 3, mean ± s.d.). **d**, Growth curves of WT, Δ*manY* and CΔ*manY* in CDM with 0.2, 1, 2 and 10 g/L glucose (n = 4, mean ± s.d.). **e**, The Transcripts Per Million (TPM) of all PTS genes in *S. zooepidemicus*. Bacteria were cultured in 2 g/L glucose CDM at mid-exponential phase. Each row indicates one replicate. The PTS clusters are indicated as shown. **f**, Growth curve of WT, Δ*manY*, Δ*ptsG* and Δ*930* in CDM with 2 g/L glucose (n = 4, mean ± s.d.). Growth curves (panel b, d and f) were modeled using a Gompertz nonlinear regression model and the estimated parameters were compared across groups by one-way ANOVA, with Dunnett's multiple comparisons test.

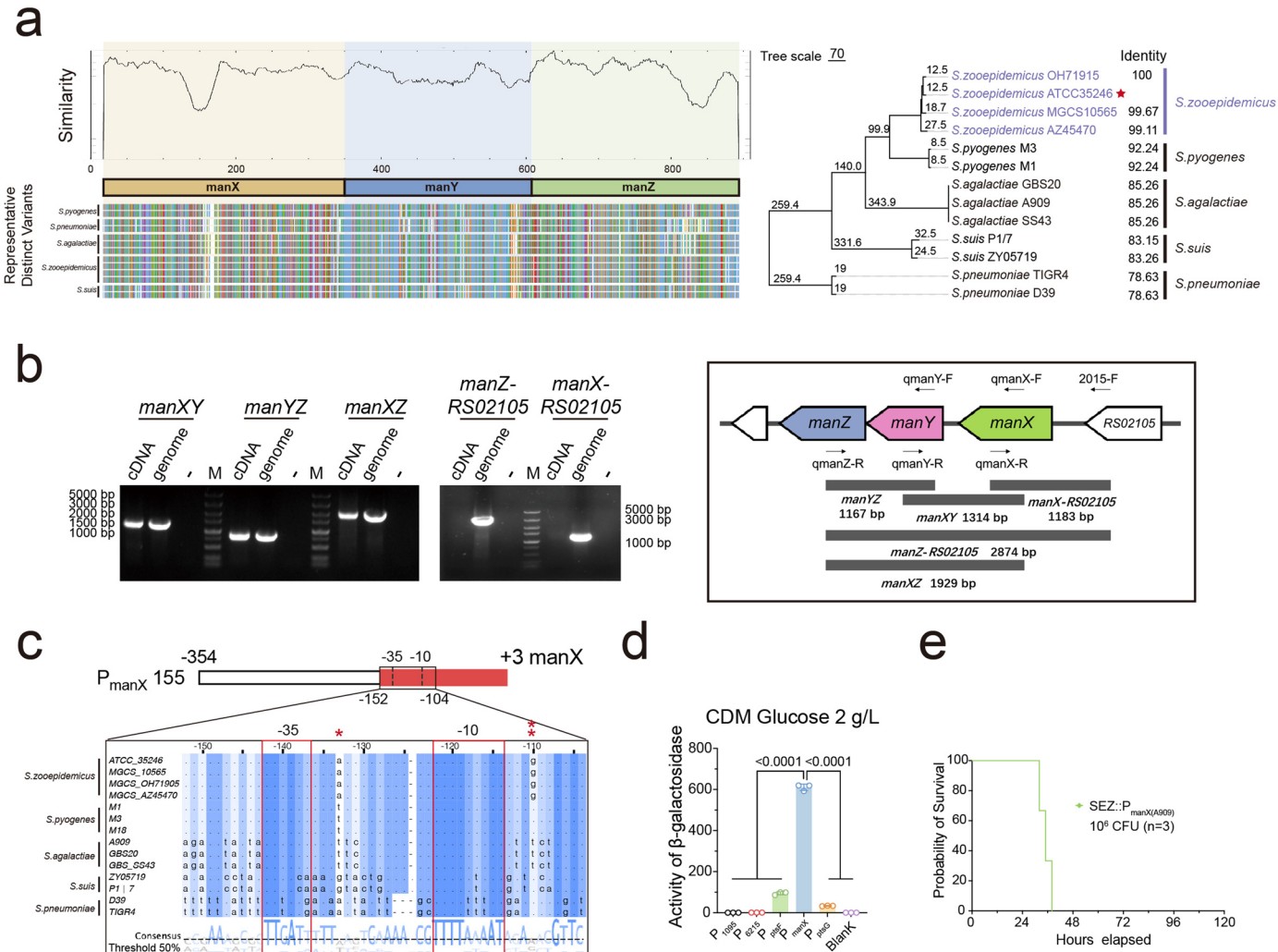

**Extended Data Fig. 6 | Comparative analysis of the PTS$_{man}$ operon in *S. zooepidemicus* and other streptococcal species. a**, Left: Amino acid sequence homology of PTS$_{man}$ components. Similarity plot depicts conservation across protein variants (y-axis: relative conservation over adjacent residues; higher values denote greater conservation). Bottom coloured lines: multiple sequence alignment of representative manX/manY/manZ conducted by MUSCLE v3.8.31. Blank spaces signify gaps or distinct amino acids across species. Colours follow Clustal X colour scheme, where the same type of amino acid has the same colour (for example, blue corresponds to hydrophobic amino acids). Right: evolutionary tree plot of PTS$_{man}$ across species via BLUSUM in Jalview 2.11.4.1. Tree branch shows distance. Red star: *S. zooepidemicus* ATCC35246. Purple highlight: *S. zooepidemicus*. **b**, PTS$_{man}$ genes amplified from genomic DNA or RT-cDNA with primers in Supplementary Table S8, amplicons detected by agarose gel electrophoresis. M: 5000 bp marker; "-": negative control with

no template. Right schematic: PTS$_{man}$ gene cluster with upstream/downstream genes; arrows: primer binding sites; black squares: amplified regions by the corresponding primers. **c**, Nucleotide homology of PTS$_{man}$ promoter (-152 to -104 bp, spanning 10 bp upstream of -35 to 10 bp downstream of -10) across *Streptococcus* strains (listed left). Red squares: -35 and -10 regions; Blue shades: nucleotide homology (threshold 50%); "-": a gap base; ".": consensus exceeding 50%. "*" and "**" respectively mark the mutant bases at -133 and -110 sites of *manX*. **d**, *S. zooepidemicus* PTS promoter activities evaluated by cloning them into the β-galactosidase reporter plasmid pTCV-lacZ. Bacteria were cultured in 2 g/L glucose in CDM to mid-exponential phase; empty plasmid pTCV-lacZ was used as a control (n = 3 biological replicates, mean ± s.d., one-way ANOVA following Dunnett's multiple comparisons test). **e**, Survival curves of C57BL/6 J mice i.v. injected with 10^6 CFU SEZ::P$_{manX(A909)}$ and monitored to moribund stage (n = 3).

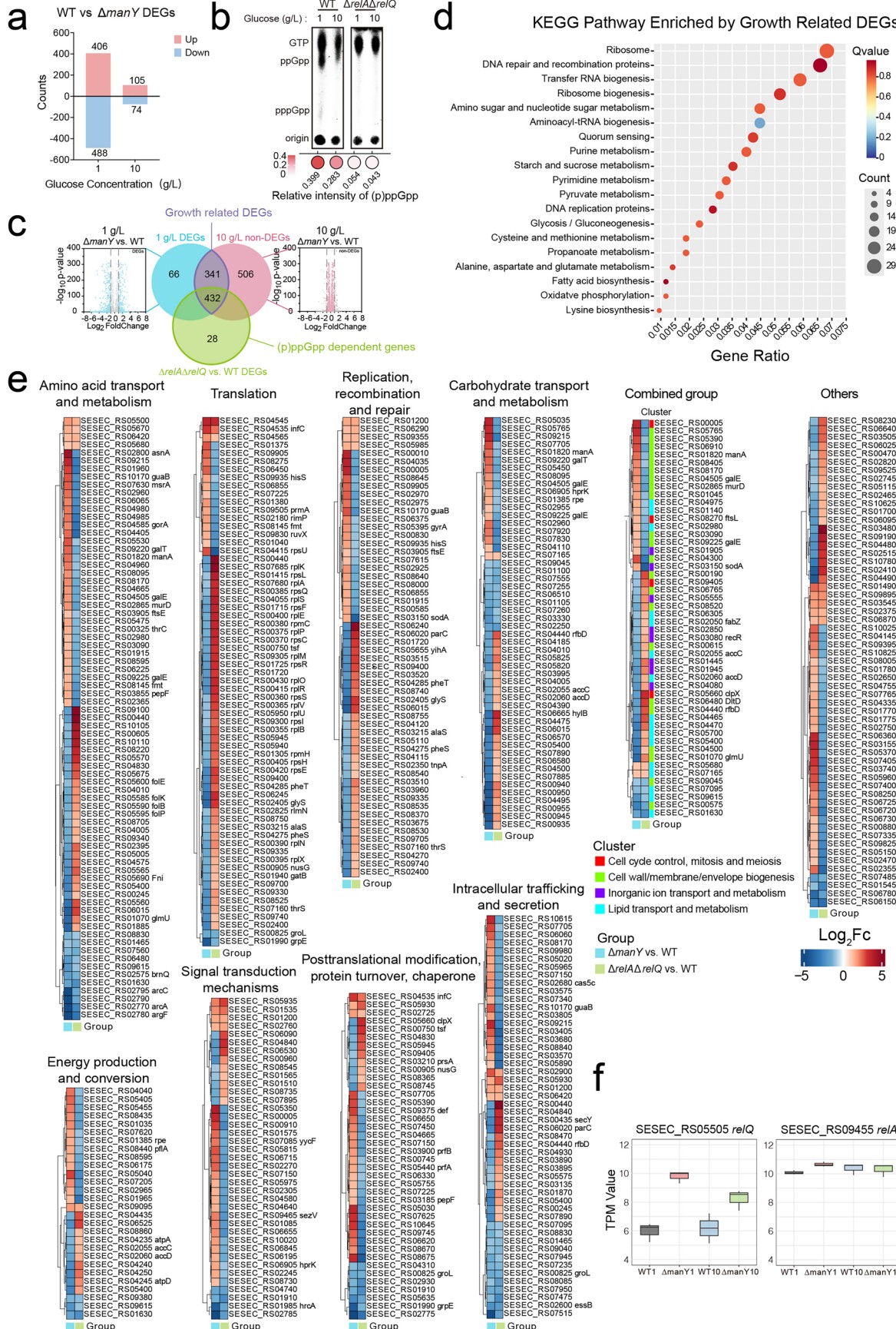

**Extended Data Fig. 7 | See next page for caption.**

**Extended Data Fig. 7 | RNA-seq analysis of WT, Δ*manY* and (p)ppGpp° strains.**
**a**, RNA-seq analysis shows the number of DEGs between Δ*manY* and WT cultured in CDM with 1 or 10 g/L glucose at mid-exponential phase (absolute log$_2$ fold change ≥ 1 as DEG threshold). **b**, Intracellular (p)ppGpp levels were detected by TLC after nucleotide extraction from WT, Δ*relA*Δ*relQ* strains grown in CDM with low (1 g/L) or high (10 g/L) glucose. **c**, The Venn diagram illustrates the intersection among DEGs (Δ*manY* vs. WT in 1 g/L glucose CDM), non-DEGs (Δ*manY* vs. WT in 10 g/L glucose CDM), and DEGs (Δ*relA*Δ*relQ* vs. WT). Growth related genes are marked in the purple lens-sharped region (intersection of Δ*manY* vs. WT in 1 g/L glucose and non-DEGs in 10 g/L glucose), while the (p) ppGpp dependent genes are shown in the green circle (Δ*relA*Δ*relQ* vs. WT DEGs). The central overlap (432 genes) among all three sets indicates candidates involved in both growth regulation and (p)ppGpp. **d**, KEGG pathway enrichment plot of growth regulated genes (773). Dot size represents DEG count per pathway; colour indicates Q-value. Rich factor = (DEGs in KEGG pathway) / (total enriched genes). **e**, The heatmap displays the Log$_2$Fc of the intersection genes in panel c in Δ*manY* vs. WT (influenced by *manY* in low glucose) and Δ*relA* Δ*relQ* vs. WT (influenced by (p)ppGpp) groups. The combined group consists of four distinct clusters, with the different coloured squares on the right side of the heatmap representing each cluster. The gene list related with growth regulation and (p)ppGpp is provided in Supplementary Tables S9 and S13. **f**, Box plots of normalized TPM for *relA* and *relQ* in WT and Δ*manY* cultured in CDM with 1 g/L (WT1 and Δ*manY*1) or 10 g/L (WT10 and Δ*manY*10) glucose (n = 3). Center line: median; box bounds: 25$^{th}$-75$^{th}$ percentiles; minima/maxima: smallest/largest values in dataset.

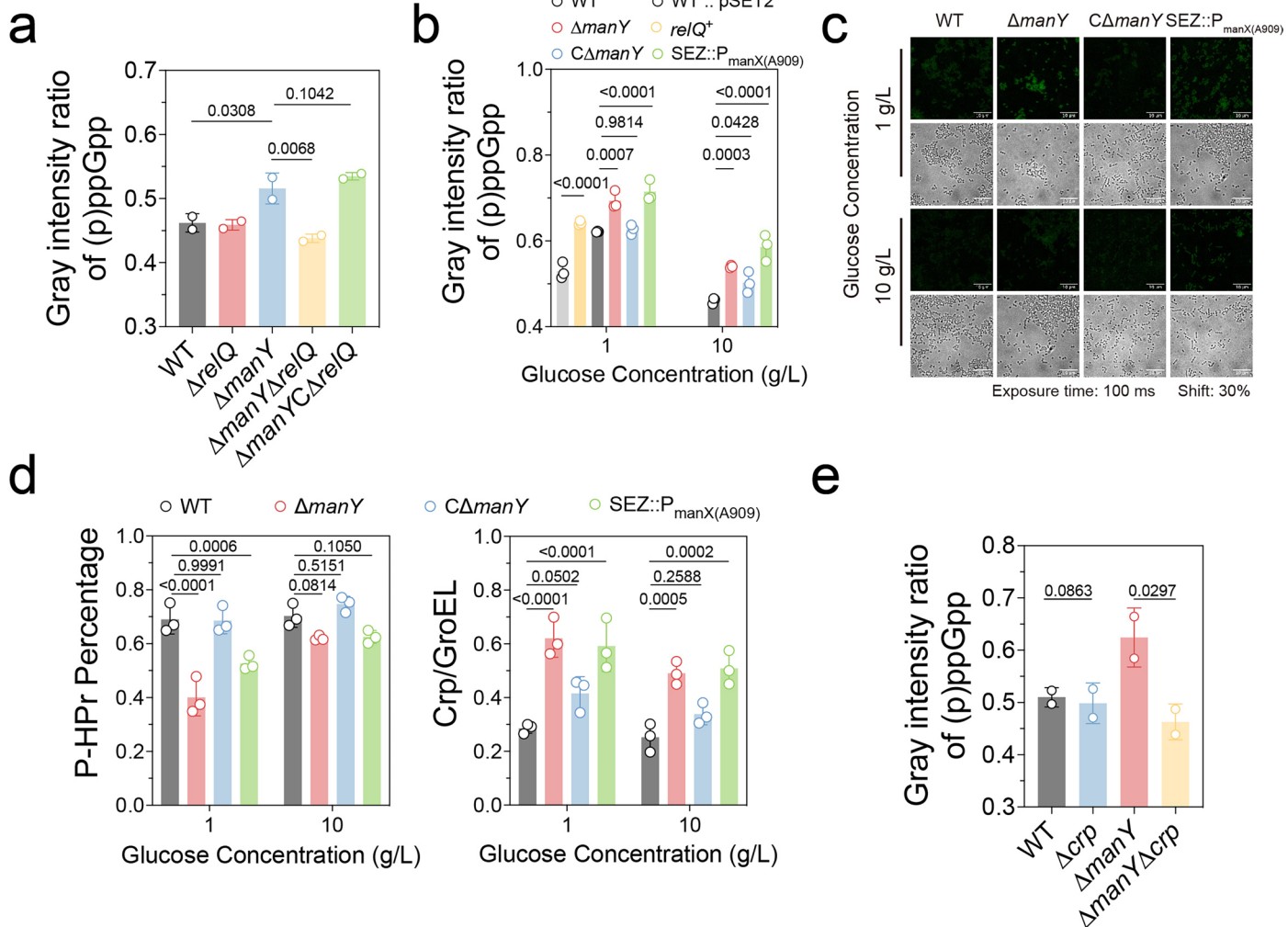

**Extended Data Fig. 8 | Quantitative analysis of (p)ppGpp, Crp and P-HPr level.**
**a, b,** Gray intensity analysis of intracellular levels of (p)ppGpp detected by TLC plates (n = 2 for a; n = 3 for b, biological replicates). The intracellular levels of (p)ppGpp = (ppGpp + pppGpp) / (GTP + ppGpp + pppGpp). Panel a and b correspond to Figs. 6c, d, respectively. **c,** Fluorescence imaging of (p)ppGpp biosynthesis in WT, ΔmanY, CΔmanY and SEZ::$P_{manX(A909)}$ strains under 1 g/L or 10 g/L glucose. All strains were grown to mid-exponential phase in CDM containing 1 or 10 g/L glucose, and fluorescence images were captured with a 100 ms exposure time at 30% shift. GFP signal ranged from 200 to 500 and bright field ranged from 800 to 4000. Identical imaging conditions and settings were applied across all figures,

as detailed in the supplementary materials. **d,** Gray intensity analysis of Crp expression levels and the percentage of P-HPr (n = 3, biological replicates). Crp expression level is calculated by measuring the gray intensity as the ratio of Crp to GroEL. The percentage of P-HPr is calculated by measuring the gray intensity as the ratio of P-HPr to total HPr. **e,** Gray intensity analysis of intracellular levels of (p)ppGpp detected by TLC plates in Fig. 6h (n = 2, biological replicates). Data in this figure presented as mean ± s.d. Panel a and e: one-way ANOVA following Šídák's multiple comparisons test. Panel b and c: two-way ANOVA following Šídák's multiple comparisons test.

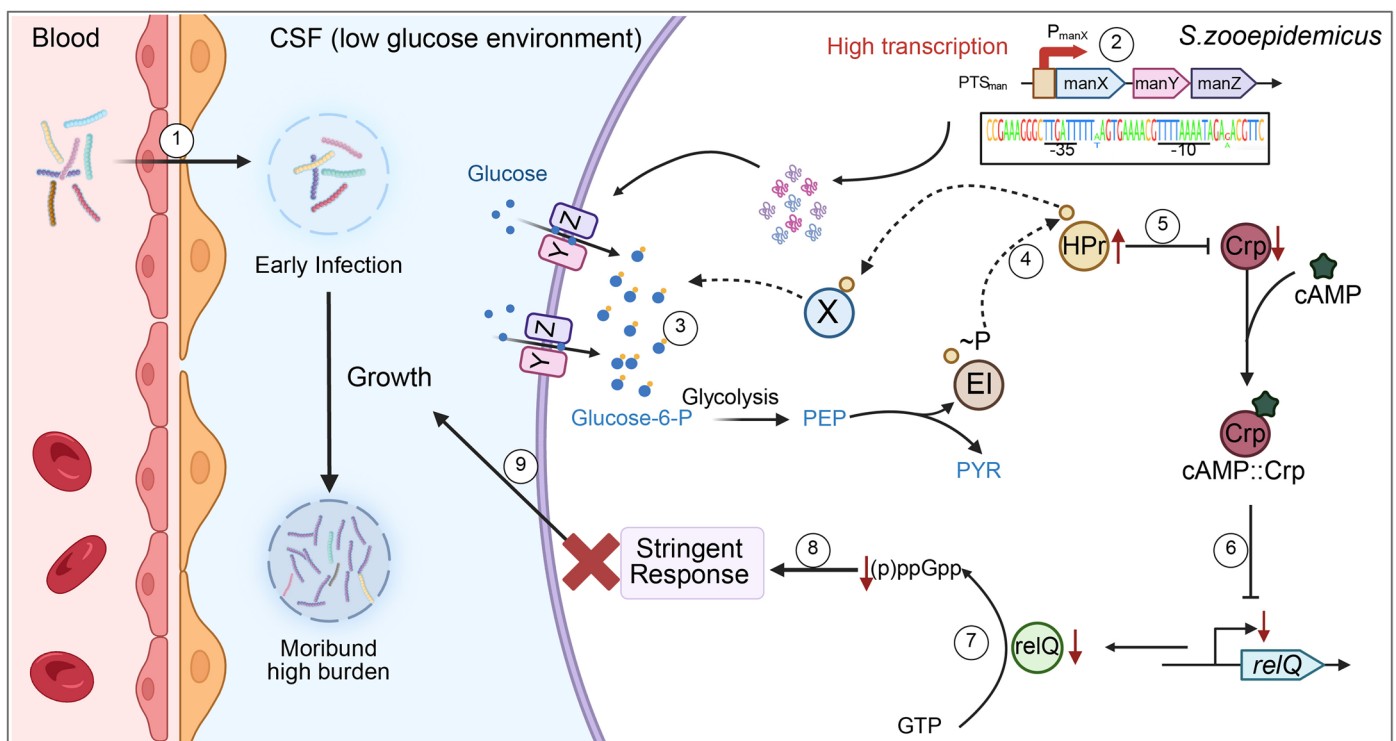

**Extended Data Fig. 9 | Schematic model of *S. zooepidemicus* proliferation under low glucose via PTS<sub>man</sub>-dependent bypass of the stringent response.** ① *S. zooepidemicus* enters the CSF through blood, establishing a small founding population in the CSF at the early infection stages and then proliferates to a high burden. ② The species-specific conserved promoter drives high transcription of **PTS_man**, ③ enabling sufficient glucose transport in low-glucose environments. ①-③ regarding the *in vivo* data from animal experiments. When *S. zooepidemicus* has access to adequate intracellular glucose, ④ the

**HPr** phosphorylation ratio increases, ⑤ leading to reduced **Crp** expression. As a result, ⑥ transcription of ***relQ*** will not be activated by cAMP::Crp complex, ⑦ resulting in the absence of **(p)ppGpp** accumulation. ⑧ Consequently, the stringent response is not triggered, ⑨ allowing *S. zooepidemicus* to continue replicating to high burden in low glucose environment, resembling the conditions of the CSF during bacterial CNS infection. ④-⑨ regarding the *in vitro* data. Figure created with BioRender.com.

# Reporting Summary

## Statistics

For all statistical analyses, confirm that the following items are present in the figure legend, table legend, main text, or Methods section.

| n/a | Confirmed | |
|---|---|---|
| ☐ | ☒ | The exact sample size (*n*) for each experimental group/condition, given as a discrete number and unit of measurement |
| ☐ | ☒ | A statement on whether measurements were taken from distinct samples or whether the same sample was measured repeatedly |
| ☐ | ☒ | The statistical test(s) used AND whether they are one- or two-sided *Only common tests should be described solely by name; describe more complex techniques in the Methods section.* |
| ☒ | ☐ | A description of all covariates tested |
| ☐ | ☒ | A description of any assumptions or corrections, such as tests of normality and adjustment for multiple comparisons |
| ☐ | ☒ | A full description of the statistical parameters including central tendency (e.g. means) or other basic estimates (e.g. regression coefficient) AND variation (e.g. standard deviation) or associated estimates of uncertainty (e.g. confidence intervals) |
| ☐ | ☒ | For null hypothesis testing, the test statistic (e.g. *F*, *t*, *r*) with confidence intervals, effect sizes, degrees of freedom and *P* value noted *Give P values as exact values whenever suitable.* |
| ☒ | ☐ | For Bayesian analysis, information on the choice of priors and Markov chain Monte Carlo settings |
| ☒ | ☐ | For hierarchical and complex designs, identification of the appropriate level for tests and full reporting of outcomes |
| ☒ | ☐ | Estimates of effect sizes (e.g. Cohen's *d*, Pearson's *r*), indicating how they were calculated |

*Our web collection on statistics for biologists contains articles on many of the points above.*

## Software and code

Policy information about availability of computer code

Data collection
> The following software was used to collect data:
> Image Lab (v 6.0.1);
> ZEISS software Zen (v 3.9.101.0000);
> Berthold Technologies software IndiGo in vivo imaging system (v2.0.2.0);
> illumina NextSeq 500/550 Control Software;
> Nano Drop 2000/2000c (v 1.6.198);
> Tecan i-control (v 3.9.1.0);
> StepOne Software (v 2.3);
> Amersham Typhoon software CytExpert (v 2.3);

Data analysis
> The following software and code was used to collect data:
> GraphPad Software 9.0;
> R Studio (v1.4.1717);
> ImageJ Software (v2.14.0);
> SnapGene Software(v4.3.4);
> Circular-Plot Software(v18.2.0);
> Biotie2 (v2.2.5);
> FeatureCounts (v1.5.0);
> RSEM (v1.2.12);
> Phyper (v3.6.0);
> MUSCLE (v3.8.31);

MMSeqs2 (15-6f452);
Blastn (v2.7.1, NCBI);
Jalview 2.11.4.1.

For manuscripts utilizing custom algorithms or software that are central to the research but not yet described in published literature, software must be made available to editors and reviewers. We strongly encourage code deposition in a community repository (e.g. GitHub). See the Nature Portfolio guidelines for submitting code & software for further information.

## Data

Policy information about availability of data

All manuscripts must include a data availability statement. This statement should provide the following information, where applicable:
- Accession codes, unique identifiers, or web links for publicly available datasets
- A description of any restrictions on data availability
- For clinical datasets or third party data, please ensure that the statement adheres to our policy

The raw files for RNAseq data generated in this study have been deposited in the NCBI database under accession code BioProject ID PRJNA1131986 (https://www.ncbi.nlm.nih.gov/bioproject/PRJNA1131986).
The raw files for Tn-seq data generated in this study have been deposited in the NCBI database under accession code BioProject ID PRJNA1132128 (https://www.ncbi.nlm.nih.gov/bioproject/PRJNA1132128).
The raw files for STAMP data generated in this study have been deposited in the NCBI database under accession code BioProject ID PRJNA1132155 (https://www.ncbi.nlm.nih.gov/bioproject/PRJNA1132155).
The scanned HE-stained slide files are available on Mendeley Data and can be downloaded from the following link: https://data.mendeley.com/datasets/fkg44432ct/1.
Source data are provided with this paper. The data that support the findings of this study are available from the corresponding author upon request.

## Research involving human participants, their data, or biological material

Policy information about studies with human participants or human data. See also policy information about sex, gender (identity/presentation), and sexual orientation and race, ethnicity and racism.

| Reporting on sex and gender | N/A |
| Reporting on race, ethnicity, or other socially relevant groupings | N/A |
| Population characteristics | N/A |
| Recruitment | N/A |
| Ethics oversight | N/A |

Note that full information on the approval of the study protocol must also be provided in the manuscript.

# Field-specific reporting

Please select the one below that is the best fit for your research. If you are not sure, read the appropriate sections before making your selection.

☒ Life sciences ☐ Behavioural & social sciences ☐ Ecological, evolutionary & environmental sciences

For a reference copy of the document with all sections, see nature.com/documents/nr-reporting-summary-flat.pdf

# Life sciences study design

All studies must disclose on these points even when the disclosure is negative.

| Sample size | There are at least three biological replicates in all of the experiments unless otherwise stated. No statistical methods were used to pre-determine sample sizes, but our sample sizes are similar to those reported in previous publications (Pletzer D, et al. mBio. 2017;8: e00140-17; Chang J, et al. Nat Commun. 2025;16: 6928z) |
| Data exclusions | No data were excluded. |
| Replication | All experiments were performed with at least three independent biological replicates and repeat at least for two times. |
| Randomization | The strains are separated to specific group according to their known single variables. Animals were randomly separated to different experimental groups. Microscopy observations were made from random selected vision, and multiple independent slices from different biological replications were taken for analysis. |
| Blinding | Data collection and analysis were not performed blind to the conditions of the experiments, except for histopathology scoring. |

# Reporting for specific materials, systems and methods

We require information from authors about some types of materials, experimental systems and methods used in many studies. Here, indicate whether each material, system or method listed is relevant to your study. If you are not sure if a list item applies to your research, read the appropriate section before selecting a response.

## Materials & experimental systems

| n/a | Involved in the study |
|---|---|
| ☐ | ☒ Antibodies |
| ☒ | ☐ Eukaryotic cell lines |
| ☒ | ☐ Palaeontology and archaeology |
| ☐ | ☒ Animals and other organisms |
| ☒ | ☐ Clinical data |
| ☒ | ☐ Dual use research of concern |
| ☒ | ☐ Plants |

## Methods

| n/a | Involved in the study |
|---|---|
| ☒ | ☐ ChIP-seq |
| ☒ | ☐ Flow cytometry |
| ☒ | ☐ MRI-based neuroimaging |

## Antibodies

| | |
|---|---|
| Antibodies used | Anti-SzM antibody was conducted by our lab in previous study (1:500 dilution). <br> Goat anti-mouse IgG H&L (Alexa Fluor® 488) (Abcam, ab150113, 1:1000 dilution). <br> Rabbit anti-prfA(crp) polyclonal antibody  (CUSABIO,CSB-PA325257XA01LPY, Lot: K1231A, 1:2000 dilution) <br> Anti-GroEL antibody was conducted by our lab in previous study (1:1000 dilution). <br> Rabbit anti-ptsH(HPr) polyclonal antibody  (CUSABIO,CSB-PA362538HA01BRJ, Lot:  CBF0621A, 1:2000 dilution) <br> Goat Anti-Rabbit IgG H&L (HRP) (Abcam, ab97051, 1:5000 dilution) |
| Validation | Specificity of SzM antibody is validated in IFA test in figure 1c to verify the distribution of SEZ (DOI: 10.1128/spectrum.01742-22). <br> Rabbit anti-prfA (crp) polyclonal antibody: Used for the detection of intracellular Crp levels in Figure 6e. <br> GroEL antibody: Used as an internal control for bacterial protein expression in Figure 6e. <br> Rabbit anti-ptsH (HPr) polyclonal antibody: Used for the analysis of intracellular HPr phosphorylation levels in Figure 6e. |

## Animals and other research organisms

Policy information about studies involving animals; ARRIVE guidelines recommended for reporting animal research, and Sex and Gender in Research

| | |
|---|---|
| Laboratory animals | All mice used in this study were female C57BL/6J mice (GemPharmatech, Nanjing) aged between 6 and 8 weeks.  All swine used in this study to collected CSF were Landrace pigs (Zhao Fenghua biotechnology, Nanjing) aged between 12 and 16 weeks. |
| Wild animals | No wild animals were used. |
| Reporting on sex | In this study, only female mice were used to minimize variability and ensure consistency across all experimental conditions. Female mice were chosen for easy to operate. Using single sex reduced biological variability and helped to achieve more consistent and reliable results. |
| Field-collected samples | No field-collected samples were used. |
| Ethics oversight | All mice before experiment are housed in specific pathogen free conditions at Nanjing Agricultural University Laboratory Animal Center for 7 days prior to the experiment. All mice were kept under a 12-hour light-dark cycles, with temperature controlled at 22-24°C and humidity at 40-60%. Mice had free access to food and water. All animal experiments were performed with protocols approved by the Laboratory Animal Welfare and Ethics Committee of Nanjing Agricultural University (protocol number: NJAU.No20220311038) in accordance with the Laboratory Animal Guideline for ethical review of animal welfare (GB/T 35892-2018). |

Note that full information on the approval of the study protocol must also be provided in the manuscript.

## Plants

Seed stocks

N/A

Novel plant genotypes

N/A

Authentication

N/A

