## [Peer Review File · Nature Microbiology]

Zoonotic Streptococcus imports glucose to inhibit stringent response and promote growth during meningitis

Corresponding Author: Professor Zhe Ma

Version 0:

Reviewer comments:

Reviewer #1

(Remarks to the Author)

GENERAL COMMENT

An interesting study by Yuan et al investigating the pathogenesis of *S. equi* subsp. *Zooepidemicus* meningitis, a disease of great societal and health impact in farm animals (pigs) and humans. Authors have interestingly studied the interplay between the metabolic properties of the bacteria within the host and disease pathogenesis. Yet the study presents technical and scientific unclear points. My comments, remarks and suggestions are the following:

TEXTUAL COMMENTS

Line 42: Streptococcus IS a common cause

Line 43: Streptococcus pneumonia (the pneumococcus)

Lines 51-52: blood-CNS-barrier, do the author mean blood-brain barrier here?

Line 52: authors state what contemporary studies focus on, but biologically speaking, what are the differences between these two routes?

METHODS

Animal experiments. Experimental meningitis can be induced in vivo either systemically (bacteremia-derived meningitis mouse model) or through intracisternal administration of bacteria directly into the csf. Even though bacteria can travel to the brain from the nose through the olfactory bulb, a minority of the bacteria in the nasal cavity reaches the CNS through this route. Therefore, it is not clear to me why the authors have chosen this in vivo model.

RESULTS

Fig 1a. Were animals perfused before organ collection? If not, it is impossible to distinguish between bacteria in the blood stream from bacteria within the tissue. Same remarks goes for the IVIS imaging results in 1b.

Fig 2. Why was the SEZstamp library injected iv, while the infection mouse model is based on an intranasal challenge? Again, if organs were collected from non-perfused mice, it is very hard to analyze SEZ infection dynamics with barcoded bacteria since it is not possible to distinguish the blood-borne bacteria that have infiltrated into tissues.

Fig 3. With a more typical meningitis model (iv or ic infection), instead of the intranasal model, would have the gene identification analysis led to different results?

Fig 4. Growth experiments should also be performed in swine CSF (like performed in 4g) with increasing concentrations of the five carbohydrates (each carbohydrate in separate growth experiments) to assess differences in growth phenotypes.

Fig 5. Higher or lower presence of bacteria in tissues is not necessarily linked to a higher or lower growth capability of the bacteria. A higher expression of virulence factors can also lead to a higher invasiveness without affecting growth rate. Have authors assessed this?

Fig S1. Bacteria in the brain has to be properly defined. If animals were not perfused, bacteria could have either been in the blood circulation within the BBB (so, not within the brain tissue yet) or within the brain tissue. IVIS imaging does not have such high resolution to distinguish the two groups.

Fig S3. Again, without animal perfusion, it is not possible to analyse the GD values since each organ could have had bacteria that were inside the blood vasculature (since each organ is vascularized). Bacteria that are in the blood stream are not yet in any other tissue, and gene expression of bacteria during the invasion process (from blood into other tissues) can change a lot.

Growth experiments, general comment. Have authors performed statistical analysis to assess differences in growths between the experimental groups?

Reviewer #2

(Remarks to the Author)

This manuscript by Yuan et al. explores the ability of *Streptococcus equi* subspecies *zooepidemicus* (SEZ) to proliferate in cerebrospinal fluid (CSF) and uncovers the mechanisms underlying its glucose acquisition strategies, particularly through the

phosphotransferase system (PTS_{man}). The study provides valuable insights into SEZ's adaptations in nutrient-limited environments and proposes potential therapeutic interventions targeting PTS_{man} to mitigate CNS infections.

The study addresses a critical gap in our understanding of how certain bacterial pathogens survive and proliferate in the nutrient-limited environment of the CSF. The focus on SEZ's glucose acquisition system, specifically the constitutive expression of PTS_{man}, is novel and offers important therapeutic implications.

While the findings are significant, the manuscript could benefit from a broader discussion on how these insights might be applied to other bacterial pathogens. The study's implications for therapeutic development, while mentioned, are not explored in sufficient depth. Additionally, comparisons with other *Streptococcus* species could be expanded to strengthen the claims about SEZ's unique adaptations.

Specific Comments:

- The use of barcoded bacteria and the STAMPR analytic framework is a strong methodological approach that allows for detailed tracking of pathogen dissemination in the CNS. The experimental design appears sound, with appropriate controls and replication to support the conclusions drawn. However, I found the study lacking novelty and excitement in certain areas. For example, the role of the immune system is never touched upon in this study, and I think it should be at least explored. Additionally, 10^{10} bacteria in the CSF seems like a very high number and might need further justification.
- The manuscript would benefit from more detailed explanations of certain experimental procedures. For example, the methods used to assess glucose levels and the expression of PTS_{man} in various environments could be more thoroughly described. Additionally, the statistical analyses, while present, need to be explicitly connected to the conclusions in a clearer manner.
- The discussion occasionally drifts into speculative territory, particularly regarding the potential ecological advantages of the PTS_{man} system outside of an animal host. These claims could be more cautiously presented or supported by additional data. The role of the stringent response in SEZ, while mentioned, could be more thoroughly explored in relation to other regulatory systems.
- Some references are clustered together without a clear connection to specific points in the text. These citations could be better integrated into the narrative to strengthen the arguments made.

Minor comments:

Citations: The introduction section includes some sentences with numerous citations (e.g., lines 39-40), which could be distracting. It might be helpful to either rephrase or split such sentences to reduce the density of citations.

Line 43: pneumoniae.

Lines 51-54: The sentence about contemporary studies focusing on bacterial invasion strategies could be expanded slightly to explain why this is relevant to the current study (i.e., contrasting with the lack of research on post-invasion adaptation).

Lines 103-161: The section detailing the SEZ accumulation in murine CSF is well explained, but it might benefit from more explicit connections between the experimental results and the overarching hypothesis of the study.

Potential Revisions:

- Further Exploration of Therapeutic Implications: Expand the discussion on how targeting PTS_{man} could be applied in therapeutic settings, perhaps considering challenges and future directions for research in this area.
- Broader Contextualization: While the focus on SEZ is appropriate, drawing connections to other pathogens and conditions could enhance the manuscript's relevance to a wider audience.

The manuscript presents novel and valuable insights into the mechanisms of SEZ proliferation in the CNS, with potential implications for therapeutic development. However, it would benefit from a deeper exploration of the therapeutic implications, and a clearer focus in the discussion. Addressing these points would strengthen the manuscript and make it more suitable for publication in *Nature*.

Reviewer #3

(Remarks to the Author)

This article by Yuan et al. presents a study that investigates the unique ability of zoonotic *Streptococcus equi* (SEZ) to grow to high abundance in cerebrospinal fluid (CSF) and attributes this ability to the activity of the mannose phosphotransferase system (PTS_{man}) system in SEZ, specifically at low glucose concentrations. Additionally, they attempt to make a connection between the activity of the PTS_{man} system and the lack of stringent response activity.

To understand the kinetics of colonization of the brain and further growth, the authors use a mouse model of IV infection of uniquely barcoded SEZ. This revealed that the founding population in the brain is very small (~10 cells) and distinct from the population involved in systemic infection and the blood.

Using a SEZ transposon library, a Tnseq screen identified bacterial genes important for growth in the brain, including those involved in proline synthesis and the PTS mannose system. The authors go on to investigate how the SEZ promoter for manXYZ (P_{man}), unlike other *Streptococcal* manXYZ promoters, is still active even at low glucose (2g/L) concentrations using a promoter fused to beta-galactosidase. They show that the P_{man} is specifically responsible for this by testing in a GBS species background as well. They show that with a GBS P_{man}, SEZ is not able to grow to a higher optical density like WT SEZ in low glucose conditions and, in fact, phenocopies the manY mutant.

Perhaps the most unclear conclusion in the paper is that the PTS_{man} system in SEZ prevents the stringent response, resulting

in an increased abundance in the brain and CSF in low glucose conditions. The authors show that the activity of the SEZ PTS mannose system at low glucose concentrations and the stringent response are associated by identifying the differentially expressed genes in SEZ Δ manY in comparison to WT SEZ and identify these genes as also being regulated by the stringent response.

****Main critiques:****

- Of the genes identified, the authors focused on manY as the manY mutant resulted in no change in mortality but a significant decrease in bacterial abundance in the brain. Although this gene may describe why SEZ can grow to such high abundance, the significance of this insight to potential meningitis remains weak, as there is no link between disease severity, complications, or outcome shown from increased bacterial abundance. It is interesting that the Δ 6210 mutant was not chosen for further study as it is also related to the PTSman system and in an i.v. inoculation resulted in a defect in growth in the brain only and also had decreased mortality.
- Validation of Tn-seq screen should have been performed in the same way as the screen with the stereotaxic model to confirm that the mutants had a defect in growth in the brain in case the mutant SEZ could not pass through the blood-brain barrier in the same way, which may also result in decreased abundance in the brain.
- The GBS Pman experiment data seem overinterpreted. The in vitro growth differences seem very slight with the optical density only increasing by a few hundredths with no statistics applied. The difference in abundance in vivo was more convincing, with i.v. injected mice having lower SEZ abundance in the brain with the GBS Pman promoter compared to WT. However, these differences were apparently not significant with an alpha level of 0.05.
- The stringent response conclusions are not well-supported. There are many regulators of metabolism genes. Additional evidence for this link to the stringent response included a suboptimal qualitative detection of ppGpp via thin-layer chromatography with no complement of the manY mutant and quantification of relA transcript in SEZ Δ manY. relA transcript may increase in manY mutants, but this is not specifically indicative of relA activity. Fig 6e might suggest that the stringent response is limiting bacterial abundance, but how that is related to the PTSman system is not defined. Mechanistically, this assertion seems underdeveloped, as there is minimal evidence to suggest that the activity of Pman decreases the stringent response. If it does, the mechanism is very unclear.

****Other specific critiques:****

****Figures:****

- Fig 1: Would benefit from CFU data from other meningeal pathogens to really prove that this observation is remarkable.
- Fig 3: Unclear why results were verified in i.v. injection model and not model used in Tnseq screen.
- Fig 5: Would have appreciated an alignment of Streptococcal promoters. What is different about the SEZ promoter specifically?
- Fig 5d: Very hard to see graphs inside other graphs. I understand the authors were trying to keep the axes the same, but it makes it harder to understand the conclusion.
- Fig 6a: Title is misleading. Your Venn diagram is not Δ manY vs. SEZ.
- Fig 6b: This figure is very confusing and does not support the association between the stringent response and PTS mannose system well.
- Fig 6f: TLC very blotchy, at least for 0.2g/L. No complement included.

****Text:****

- Line 37: change "with bacterial invasion" to "when bacteria invade."
- Line 42: italicize "Streptococcus."
- Line 44: change "whereas" to "and" – it is odd to contrast two pathogens you are trying to persuade your audience are both significant.
- Line 46: change "is" to "has."
- Line 46-47: take out "emerging pandemic" or "outbreak in death."
- Line 62: italicize "Streptococci."
- Line 351: change "promoted by a strong promoter."
- Line 368: change "entrance into" to "enter."

The final claim of significance for your findings is not well supported: If there were therapeutics developed, your findings suggest that lethal meningitis would still develop (Fig 3c tail vein injection of SEZ Δ manY), which is not "mitigating the deleterious consequences of CNS infection."

****General writing critiques:****

Make sure Streptococcus and Streptococci are capitalized and italicized.
Ensure gene names are italicized (e.g., manY), including Δ manY.

Decision Letter:

18th September 2024

Dear Professor Ma,

Thank you for your patience while your manuscript "Constitutive glucose import in zoonotic Streptococci enables proliferation in cerebrospinal fluid" was under peer review at Nature Microbiology. It has now been seen by our referees, whose expertise and comments you will find at the end of this email. In the light of their advice, we have decided that we cannot offer to publish your manuscript in Nature Microbiology.

From the reports, you will see that while they find your work of some potential interest, the referees raise concerns about the strength of the novel conclusions that can be drawn at this stage. In particular, several of the reviewers raised concerns about the inoculation routes and how this might affect the relevance of the model and interpretation of the data. In addition, there were concerns over the role of the PTSman system, how it might contribute to pathogenesis as opposed to replication, and the strength of the data supporting a link to the stringent response. Unfortunately, these criticisms are sufficiently important as to preclude publication of your work in Nature Microbiology. I am sorry that we cannot be more positive on this occasion, but hope that you find the referees' comments helpful when preparing your paper for resubmission elsewhere.

I have discussed your manuscript and the reviewers' comments with our colleagues at Nature Communications. They would send the appropriately revised version to the original reviewers (if they are available) and reserve the right to recruit a reviewer with expertise in barcoding. Should you wish to have your revised paper considered by Nature Communications, please use the link to the Springer Nature manuscript transfer service provided below once the revision is ready, and include a point-by-point response to the reviewers' concerns.

Your handling editor at Nature Communications would be Dr. Nino Iakobachvili (nino.iakobachvili@nature.com). If there is anything you would like to discuss before transferring the paper and its reviews, please don't hesitate to contact her by e-mail.

Please note that Nature Communications is a fully open access journal. For information about article processing charges, open access funding, and advice and support from Springer Nature, please consult the Nature Communications Open Access page (www.nature.com/ncomms/open_access/index.html).

To transfer your manuscript please use our manuscript transfer portal. You will not have to re-supply manuscript metadata and files, unless you wish to make modifications. For more information, please see our [manuscript transfer FAQ](http://www.nature.com/authors/author_resources/transfer_manuscripts.html?WT.mc_id=EMI_NPG_1511_AUTHORTRANSF&WT.ec_id=AUTHOR) page.

Yours sincerely,

Reviewer Expertise:

Referee #1: Streptococcal infection, meningitis

Referee #2: Streptococcus pathogenesis, host-pathogen interactions

Referee #3: Streptococcal pathogenesis, Tnseq and genomic approaches

Reviewers Comments:

Reviewer #1 (Remarks to the Author):

GENERAL COMMENT

An interesting study by Yuan et al investigating the pathogenesis of *S. equi* subsp. *Zooepidemicus* meningitis, a disease of great societal and health impact in farm animals (pigs) and humans. Authors have interestingly studied the interplay between the metabolic properties of the bacteria within the host and disease pathogenesis. Yet the study presents technical and scientific unclear points. My comments, remarks and suggestions are the following:

TEXTUAL COMMENTS

Line 42: Streptococcus IS a common cause

Line 43: Streptococcus pneumonia (the pneumococcus)

Lines 51-52: blood-CNS-barrier, do the author mean blood-brain barrier here?

Line 52: authors state what contemporary studies focus on, but biologically speaking, what are the differences between these two routes?

METHODS

Animal experiments. Experimental meningitis can be induced in vivo either systemically (bacteremia-derived meningitis mouse model) or through intracisternal administration of bacteria directly into the csf. Even though bacteria can travel to the brain from the nose through the olfactory bulb, a minority of the bacteria in the nasal cavity reaches the CNS through this route. Therefore, it is not clear to me why the authors have chosen this in vivo model.

RESULTS

Fig 1a. Were animals perfused before organ collection? If not, it is impossible to distinguish between bacteria in the blood stream from bacteria within the tissue. Same remarks goes for the IVIS imaging results in 1b.

Fig 2. Why was the SEZstamp library injected iv, while the infection mouse model is based on an intranasal challenge? Again, if organs were collected from non-perfused mice, it is very hard to analyze SEZ infection dynamics with barcoded bacteria since it is not possible to distinguish the blood-borne bacteria that have infiltrated into tissues.

Fig 3. With a more typical meningitis model (iv or ic infection), instead of the intranasal model, would have the gene identification analysis led to different results?

Fig 4. Growth experiments should also be performed in swine CSF (like performed in 4g) with increasing concentrations of the

five carbohydrates (each carbohydrate in separate growth experiments) to assess differences in growth phenotypes.

Fig 5. Higher or lower presence of bacteria in tissues is not necessarily linked to a higher or lower growth capability of the bacteria. A higher expression of virulence factors can also lead to a higher invasiveness without affecting growth rate. Have authors assessed this?

Fig S1. Bacteria in the brain has to be properly defined. If animals were not perfused, bacteria could have either been in the blood circulation within the BBB (so, not within the brain tissue yet) or within the brain tissue. IVIS imaging does not have such high resolution to distinguish the two groups.

Fig S3. Again, without animal perfusion, it is not possible to analyse the GD values since each organ could have had bacteria that were inside the blood vasculature (since each organ is vascularized). Bacteria that are in the blood stream are not yet in any other tissue, and gene expression of bacteria during the invasion process (from blood into other tissues) can change a lot.

Growth experiments, general comment. Have authors performed statistical analysis to assess differences in growths between the experimental groups?

Reviewer #2 (Remarks to the Author):

This manuscript by Yuan et al. explores the ability of *Streptococcus equi* subspecies *zooepidemicus* (SEZ) to proliferate in cerebrospinal fluid (CSF) and uncovers the mechanisms underlying its glucose acquisition strategies, particularly through the phosphotransferase system (PTS_{man}). The study provides valuable insights into SEZ's adaptations in nutrient-limited environments and proposes potential therapeutic interventions targeting PTS_{man} to mitigate CNS infections.

The study addresses a critical gap in our understanding of how certain bacterial pathogens survive and proliferate in the nutrient-limited environment of the CSF. The focus on SEZ's glucose acquisition system, specifically the constitutive expression of PTS_{man}, is novel and offers important therapeutic implications.

While the findings are significant, the manuscript could benefit from a broader discussion on how these insights might be applied to other bacterial pathogens. The study's implications for therapeutic development, while mentioned, are not explored in sufficient depth. Additionally, comparisons with other *Streptococcus* species could be expanded to strengthen the claims about SEZ's unique adaptations.

Specific Comments:

- The use of barcoded bacteria and the STAMPR analytic framework is a strong methodological approach that allows for detailed tracking of pathogen dissemination in the CNS. The experimental design appears sound, with appropriate controls and replication to support the conclusions drawn. However, I found the study lacking novelty and excitement in certain areas. For example, the role of the immune system is never touched upon in this study, and I think it should be at least explored. Additionally, 10^{10} bacteria in the CSF seems like a very high number and might need further justification.
- The manuscript would benefit from more detailed explanations of certain experimental procedures. For example, the methods used to assess glucose levels and the expression of PTS_{man} in various environments could be more thoroughly described. Additionally, the statistical analyses, while present, need to be explicitly connected to the conclusions in a clearer manner.
- The discussion occasionally drifts into speculative territory, particularly regarding the potential ecological advantages of the PTS_{man} system outside of an animal host. These claims could be more cautiously presented or supported by additional data. The role of the stringent response in SEZ, while mentioned, could be more thoroughly explored in relation to other regulatory systems.
- Some references are clustered together without a clear connection to specific points in the text. These citations could be better integrated into the narrative to strengthen the arguments made.

Minor comments:

Citations: The introduction section includes some sentences with numerous citations (e.g., lines 39-40), which could be distracting. It might be helpful to either rephrase or split such sentences to reduce the density of citations.

Line 43: pneumoniae.

Lines 51-54: The sentence about contemporary studies focusing on bacterial invasion strategies could be expanded slightly to explain why this is relevant to the current study (i.e., contrasting with the lack of research on post-invasion adaptation).

Lines 103-161: The section detailing the SEZ accumulation in murine CSF is well explained, but it might benefit from more explicit connections between the experimental results and the overarching hypothesis of the study.

Potential Revisions:

- Further Exploration of Therapeutic Implications: Expand the discussion on how targeting PTS_{man} could be applied in therapeutic settings, perhaps considering challenges and future directions for research in this area.
- Broader Contextualization: While the focus on SEZ is appropriate, drawing connections to other pathogens and conditions could enhance the manuscript's relevance to a wider audience.

The manuscript presents novel and valuable insights into the mechanisms of SEZ proliferation in the CNS, with potential implications for therapeutic development. However, it would benefit from a deeper exploration of the therapeutic implications, and a clearer focus in the discussion. Addressing these points would strengthen the manuscript and make it more suitable for publication in Nature.

Reviewer #3 (Remarks to the Author):

This article by Yuan et al. presents a study that investigates the unique ability of zoonotic *Streptococci equi* (SEZ) to grow to high abundance in cerebrospinal fluid (CSF) and attributes this ability to the activity of the mannose phosphotransferase system (PTS_{man}) system in SEZ, specifically at low glucose concentrations. Additionally, they attempt to make a connection between the activity of the PTS_{man} system and the lack of stringent response activity.

To understand the kinetics of colonization of the brain and further growth, the authors use a mouse model of IV infection of uniquely barcoded SEZ. This revealed that the founding population in the brain is very small (~10 cells) and distinct from the population involved in systemic infection and the blood.

Using a SEZ transposon library, a Tnseq screen identified bacterial genes important for growth in the brain, including those involved in proline synthesis and the PTS mannose system. The authors go on to investigate how the SEZ promoter for manXYZ (P_{man}), unlike other Streptococcal manXYZ promoters, is still active even at low glucose (2g/L) concentrations using a promoter fused to beta-galactosidase. They show that the P_{man} is specifically responsible for this by testing in a GBS species background as well. They show that with a GBS P_{man}, SEZ is not able to grow to a higher optical density like WT SEZ in low glucose conditions and, in fact, phenocopies the manY mutant.

Perhaps the most unclear conclusion in the paper is that the PTS_{man} system in SEZ prevents the stringent response, resulting in an increased abundance in the brain and CSF in low glucose conditions. The authors show that the activity of the SEZ PTS mannose system at low glucose concentrations and the stringent response are associated by identifying the differentially expressed genes in SEZΔmanY in comparison to WT SEZ and identify these genes as also being regulated by the stringent response.

****Main critiques:****

- Of the genes identified, the authors focused on manY as the manY mutant resulted in no change in mortality but a significant decrease in bacterial abundance in the brain. Although this gene may describe why SEZ can grow to such high abundance, the significance of this insight to potential meningitis remains weak, as there is no link between disease severity, complications, or outcome shown from increased bacterial abundance. It is interesting that the Δ6210 mutant was not chosen for further study as it is also related to the PTS_{man} system and in an i.v. inoculation resulted in a defect in growth in the brain only and also had decreased mortality.
- Validation of Tn-seq screen should have been performed in the same way as the screen with the stereotaxic model to confirm that the mutants had a defect in growth in the brain in case the mutant SEZ could not pass through the blood-brain barrier in the same way, which may also result in decreased abundance in the brain.
- The GBS P_{man} experiment data seem overinterpreted. The in vitro growth differences seem very slight with the optical density only increasing by a few hundredths with no statistics applied. The difference in abundance in vivo was more convincing, with i.v. injected mice having lower SEZ abundance in the brain with the GBS P_{man} promoter compared to WT. However, these differences were apparently not significant with an alpha level of 0.05.
- The stringent response conclusions are not well-supported. There are many regulators of metabolism genes. Additional evidence for this link to the stringent response included a suboptimal qualitative detection of ppGpp via thin-layer chromatography with no complement of the manY mutant and quantification of relA transcript in SEZΔmanY. relA transcript may increase in manY mutants, but this is not specifically indicative of relA activity. Fig 6e might suggest that the stringent response is limiting bacterial abundance, but how that is related to the PTS_{man} system is not defined. Mechanistically, this assertion seems underdeveloped, as there is minimal evidence to suggest that the activity of P_{man} decreases the stringent response. If it does, the mechanism is very unclear.

****Other specific critiques:****

- ****Figures:****
- Fig 1: Would benefit from CFU data from other meningeal pathogens to really prove that this observation is remarkable.
- Fig 3: Unclear why results were verified in i.v. injection model and not model used in Tnseq screen.
- Fig 5: Would have appreciated an alignment of Streptococcal promoters. What is different about the SEZ promoter specifically?
- Fig 5d: Very hard to see graphs inside other graphs. I understand the authors were trying to keep the axes the same, but it makes it harder to understand the conclusion.
- Fig 6a: Title is misleading. Your Venn diagram is not ΔmanY vs. SEZ.
- Fig 6b: This figure is very confusing and does not support the association between the stringent response and PTS mannose system well.
- Fig 6f: TLC very blotchy, at least for 0.2g/L. No complement included.

****Text:****

- Line 37: change "with bacterial invasion" to "when bacteria invade."
- Line 42: italicize "Streptococcus."
- Line 44: change "whereas" to "and" – it is odd to contrast two pathogens you are trying to persuade your audience are both significant.
- Line 46: change "is" to "has."
- Line 46-47: take out "emerging pandemic" or "outbreak in death."
- Line 62: italicize "Streptococci."
- Line 351: change "promoted by a strong promoter."
- Line 368: change "entrance into" to "enter."

The final claim of significance for your findings is not well supported: If there were therapeutics developed, your findings suggest

that lethal meningitis would still develop (Fig 3c tail vein injection of SEZ Δ manY), which is not “mitigating the deleterious consequences of CNS infection.”

****General writing critiques:****

Make sure *Streptococcus* and *Streptococci* are capitalized and italicized.
Ensure gene names are italicized (e.g., manY), including Δ manY.

****Nature Communications** is the Nature Portfolio flagship Open Access journal. If you would like this work to be considered for publication there, you can easily transfer the manuscript by following the instructions below. It is not necessary to reformat your paper. Once all files are received, the editors at *Nature Communications* will assess your manuscript's suitability for potential publication; they aim to provide feedback quickly, with a median decision time of 8 days for first editorial decisions on suitability. Since your paper has been peer reviewed at this journal, the referee reports will also be transferred and assessed by the editorial team. In some cases, papers are accepted without further peer review, providing a rapid path to publication. The journal is also proud to offer double blind and transparent peer review options. For 2021, the 2-year impact factor for *Nature Communications* is 17.694 and the 2-year median is 10 (for further information on journal impact factors, please visit our [Nature journals metrics page](http://www.nature.com/npg_company_info/journal_metrics.html)). Our [open access pages](http://www.nature.com/ncomms/open_access/index.html) contain information about article processing charges, open access funding, and advice and support from Springer Nature.

**** Although we cannot offer to publish your manuscript, we believe the editors at our sister journal, *Communications Biology*, will find it interesting and recommend you transfer it there.**

Communications Biology is a selective Nature Portfolio title publishing Open Access research that brings new insight in all areas of the biological sciences. [Additional journal metrics and information can be found here](https://www.nature.com/commsbio/journal-information/journal-impact). Their editors prioritise good author service, fast peer review (in 2021, the median time to decision after first review was 40 days), and are happy to answer any questions you may have commsbio@nature.com. The journal has an Impact Factor of 6.548, a CiteScore of 6.0 and a Scimago quartile ranking of Q1.

Please note that *Communications Biology* is a fully open-access journal and an article processing charge will apply to any papers accepted for publication. Our [open access pages](https://www.nature.com/commsbio/about/open-access) contain information about article processing charges, open access funding, and advice and support from Springer Nature.

If you wish to transfer your manuscript to *Communications Biology*, please use our manuscript transfer portal using the link below to initiate the transfer to this journal (or to another journal of your choice in the Nature Research portfolio). If you transfer to Nature-branded journals or to the *Communications* journals, you will not have to re-supply manuscript metadata and files. This link can only be used once and remains active until used. For more information, please see our [manuscript transfer FAQ](https://www.nature.com/nature-portfolio/for-authors/transfer) page.

****Scientific Reports** publishes primary research from all areas of the natural and clinical sciences that is judged to be scientifically valid and technically sound, whatever the considered significance. If you would like this work to be considered for publication in *Scientific Reports*, you can easily transfer the manuscript by following the instructions below. Your manuscript will be handled by an academic scientist who is an Editorial Board Member and will manage the peer review process and decide whether a paper should be accepted for publication. Most submissions are peer reviewed by one or more referees as well as the editorial board member and you can expect to receive an editorial decision within 56 days. Over 55% of the papers are published following peer review.

To discover more about this journal and, should you wish, have your paper considered the Editorial Board of *Scientific Reports*, please use the link to the manuscript transfer service provided in the footnote below. Please see our [open access pages](http://www.nature.com/ncomms/open_access/index.html) for information about article processing charges, open access funding, and advice and support from Springer Nature.

Version 1:

Reviewer comments:

Reviewer #1

(Remarks to the Author)

All comments were properly addressed. I acknowledge the excellent work done by the authors.

Minor comments:

1. Fig 5: Authors state that the increased virulence is due to a higher growth rate locally (in the brain). Yet with a bacteremia-derived meningitis model is not really possible to assess whether an increase of bacterial load in the brain is due to higher levels of bacteremia (more bacteria in the blood = more bacteria invading the brain). The only way to assess whether higher disease severity in the brain is due to a higher bacterial growth in the brain, is either to use antibiotics (that do not pass the BBB) for clearing bacteremia, or to use an ic model.

2. Western blot figures: the western blot data should be accompanied by quantification analysis of the protein bands (based on the loading controls).

Reviewer #2

(Remarks to the Author)

While the study presents an interesting hypothesis regarding the ability of *Streptococcus equi* subspecies *zooepidemicus* (SEZ) to proliferate in cerebrospinal fluid (CSF) and explores its glucose acquisition mechanisms—particularly through the phosphotransferase system (PTS_{man})—I have significant concerns regarding the validity of the findings and the robustness of the analysis.

Major Concerns:

1. Bacterial Load Estimates

The authors responded to my concern by stating that only one mouse had a bacterial load of 10^{10} CFU, while the rest ranged from 10^7 to 10^{10} , with an average of 10^9 CFU. However, this raises concerns about plausibility and statistical consistency. If the average CFU count is 10^9 , most values should cluster around this number. However, with a range spanning three orders of magnitude (10^7 to 10^{10}), this suggests substantial variability. If most values were closer to 10^7 , then the average would likely be lower than 10^9 . Conversely, if most values were near 10^{10} , then 10^7 would be an extreme outlier. A 10^3 -fold variation in bacterial load among a small group of mice is unexpected unless specific factors—such as infection severity or immune response—differ significantly.

2. Lack of Immune System Analysis

The study does not analyze the immune response to such a high bacterial burden. Given that the brain has a well-regulated immune environment, including microglial activation and cytokine responses, the absence of any discussion or data on immune system involvement weakens the study. The authors should include a thorough immunological analysis to support their claims. The authors responded to my concern by stating that the brain is considered an immune-privileged organ. While the brain has historically been considered an immune-privileged organ with limited immune cell recruitment, this concept has evolved significantly. It is now well established that the brain has an active and dynamic immune environment, with resident immune cells such as microglia playing key roles in immune surveillance and response. Additionally, under pathological conditions or infection, peripheral immune cells—including macrophages and T cells infiltrate the brain, challenging the notion of strict immune privilege. Thus, any study involving bacterial presence in the brain should thoroughly assess immune activation and potential inflammatory responses.

Therefore, I do not find this study suitable for publication in its current form.

Reviewer #3

(Remarks to the Author)

Overall, the authors have done a solid job responding to critiques. Replies to their edits/rebuttals in response to my critiques from the first submission are below.

(25) Histopathology addressed the comment about the pathological significance of the manY knockout given there is no change in mortality. Quantification would be appreciated as it is difficult to determine significance of differences in pathology between manY knockout and WT and it is not clear if images are representative of all individuals in the experiment. If penicillin is the first line antibiotic for Streptococcal bacterial meningitis, then additional data is convincing to show significance of manY to CNS infection.

(26) Practical challenges of stereotaxic route noted. Appreciated the validation of manY knockout through this route at minimum.

(27) Additional data and applied statistics seemed appropriate and addressed concerns that were brought up.

(28) Additional data 14 :

NEED to define what WT::pSET2 and WT::pSET2-reIA are. This is not defined explicitly in the text or figure legend just in methods but not clearly.

Lines 368-371 not accurately describing what is seen in the data.

I can agree with the first part of the statement: "Moreover, an increase in ppGpp was observed in Δ manY cultured under low glucose conditions"

I do not agree with the second part of the statement. There are spots that seem like ppGpp in those columns at the location that ppGpp is labeled. Maybe less ppGpp than the low glucose concentrations though. Also there was some fluorescence in those conditions with the reporter too (even if it is less): "whereas ppGpp was hardly detectable in WT or Δ manY when grown in higher glucose concentrations by either TLC (Fig. 6c) or an RNA-based fluorescent sensor (Fig. 6d)."

Additional data 9 and 10 made HPr phosphorylation at low glucose concentrations, crp involvement, and reIA connection via cAMP clear. This provided more supporting evidence that the SEZ manY system prevents the stringent response at low glucose concentrations in WT SEZ. Additional data 15 was very much appreciated as it was hard to tell the difference between the ppGpp on the TLC. I think you could see the pppGpp difference on the TLC.

Schematic (additional data 8) is good.

Other critiques:

(29) Addressed well in 23. Interesting that the same concentration of bacteria was not used between *S. agalactiae*, *S. pneumoniae*, and SEZ. However, the bacterial inoculum for SEZ was lower than *S. agalactiae* and *S. pneumoniae* and still exhibited a significant difference between abundance in the brain only in SEZ and not *S. agalactiae* and *pneumoniae*, so I think it

is okay.

(30) Addressed in 26 above.

(31) Initially was confused about the 155 notations. Looks really good but please add an indication of the specific nucleotide you are looking at in panel B in panel A. It took me a while to find that one nucleotide and the "155" does not line up with the negative numbers (I think it is -133). Adding a box to indicate the nucleotide you are looking at in A would be helpful.

(32) Addressed concern very well. Different layout of this data was appreciated.

(33-35) Concerns addressed well.

(36-44) All text edits addressed.

(45) Addressed with histopathology in 25.

(46) Addressed.

Reviewer #4

(Remarks to the Author)

For me the authors show convincing data that ManY transport system is required for glucose uptake in CSF and essential for growth in the brain. Later on the author link this observation to the stringent response. Here are some comments on this part of manuscript: It is likely that in the CSF stringent response is activated when the bacteria become limited for glucose or amino acids e.g. in the manY mutant. For *Streptococcus suis*, it was already shown that low glucose somehow results in stringent response activation (<https://www.ncbi.nlm.nih.gov/pubmed/27255540>).

The data presented do not actually show if and how (p)ppGpp is required for the observed growth defect of the manY mutant (no data using a pppGpp0 strain are provided).

Specific comments:

The authors claim that their transcriptome data (Wt versus manY under low glucose conditions) mimic stringent response gene expression. This assumption is hard to validate based on the presented data. First, no data (E.g. Excell sheet) including annotations and significance of genes are provided. How are stringent genes divined? Based on previous RNAseq-Data? It is required to show the overlap of gene expression pattern (many versus pppGpp0 strains) in a more quantitative way to judge the assumption. One hallmark of the stringent response is the downregulation of ribosomal proteins. However, these genes might also be down-regulated via other mechanisms under starving conditions. Thus, the real (p)ppGpp dependent (stringent) genes should be evaluated by comparison with transcriptome data from WT and pppGpp0 strains. This data might be available from previous publication from other streptococcal species. E.g (<https://www.ncbi.nlm.nih.gov/pubmed/27255540>).

Line 365: The link to stringent response is mainly made by the observation that rel (coding for the bifunctional Rel) was upregulated in the manY mutant. However, Rel is a bifunctional enzyme with (p)ppGpp synthase and hydrolysis activity. Thus, the (p)ppGpp level is determined via post-translational activation. (Through interaction with the ribosome and uncharged tRNA) of the synthetase activity and concurrent inhibition of the hydrolase activity http://www.ncbi.nlm.nih.gov/entrez/query.fcgi?cmd=Retrieve&db=PubMed&dopt=Citation&list_uids=15066282). Thus, transcriptional regulation of rel may lead to more or less (p)ppGpp dependent on the activation status of the enzyme.

Fig 6B shows that Rel overexpression results in growth defect. Assuming that Rel synthetase is active under low glucose conditions the data is trivial. (p)ppGpp is one of the main regulators to halt growth (inhibition of translation, replication...). However, the data does not allow direct conclusion that the growth defect in the many mutant is caused by (p)ppGpp. The growth defect can easily be explained via the decrease of glucose in the cell e.g via ATP depletion.

Fig 6C: Data are provided to show that glucose results in stringent response activation by direct detection of (p)ppGpp under low glucose conditions (Fig 6C). This data is not quantified. Only one TLC result is shown. Quantitative data based on at least three independent experiments should be shown.

Figure 6E: The meaning of the schematic figure is not very clear. What would be the role of ManY in this context

Reviewer #5

(Remarks to the Author)

The manuscript by Yuan et al. describes a mechanism behind how pathogenic *Streptococcus equi* zoopedemicus (SEZ) can replicate to high levels in CSF. Using a genetic barcoding approach, the authors reveal that the CSF is colonized by a small founding population (FP) following bloodstream infection. Once in the CSF, that small FP replicates to high levels. While other pathogenic streptococci can access the CSF, they don't replicate to as high of bacterial burden. The ability to replicate to these high levels, in part, is because of the unique increased expression of the PTSman system. The increased expression results from specific polymorphisms in the PTSman promoter. The authors finish by demonstrating this increased production of the PTSman system allows for increased replication in minimal nutrient conditions with low glucose, like those found in the CSF. Increased glucose import by the PTSman system provides enough nutrients for the cell to prevent activation of the stringent response. I found the manuscript well written, the figures easy to interpret, and the experiments thorough.

After reviewing the document and rebuttal to prior reviews, I feel the authors have addressed all of the previous reviewer's concerns. I only have minor comments that need to be addressed.

1. Line 197 – “genes that were linked to metabolism (marked with asterisks...) – I didn’t see any asterisks in the figure.
2. Line 203 – This sentence refers to 24hr CFU counts of the Δ proB and Δ proC strains and references Fig 1a and Fig. S4b. While the statement made in the sentence may be true, Fig. S4b is looking 14 days post infection, not 24 hr. Fig. 1a only shows WT data at 24 hr
3. Line 235 – This sentence refers to bacterial burden outside the CNS upon antibiotic treatment. I did not see any data where bacterial burdens were determined with antibiotic treatment, only survival curves. I am also confused as to the greater survival of the Δ manY mutant with penicillin treatment compared to WT. The authors state that both WT and the Δ manY mutant are equally susceptible to penicillin (line 235). If without penicillin treatment, both WT and Δ manY burdens are the same outside the CNS (Fig 3C), and both cause the same mortality... then why is there a difference in survival with penicillin treatment? Do the authors think this difference in mortality is due to likely differences in bacterial burden in the brain (as these are different to begin with in the absence of antibiotics)? If not, it is unclear how the sentence beginning in line 236 regarding the benefits of antibiotic treatment being enhanced by targeting nutrient uptake systems can be true. This should be clarified.
4. Line 253 should be Fig. S4c not Fig. 4c
5. Line 397 should be “promoter”

Decision Letter:

27th March 2025

Dear Professor Ma,

Thank you for your patience while your manuscript "Constitutive glucose import in zoonotic Streptococci enables proliferation in cerebrospinal fluid" was under peer-review at Nature Microbiology. It has now been seen by 5 referees, whose expertise and comments you will find at the of this email. Three of these were the original referees, while R#4 and R#5 were brought in to look into the pppGpp/stringent response and STAMPR aspects, respectively. You will see from their comments below that while they find your work of interest and recognise the significant efforts that went into revising many of the points made, some important points remain. We are very interested in the possibility of publishing your study in Nature Microbiology, but would like to consider your response to these concerns in the form of a revised manuscript before we make a final decision on publication.

In particular, you will see that Referee #1 notes that it remains unclear whether within-CNS virulence stems from growth within the CNS or more translocation across the BBB. We would ask you to address this point whether with new or existing data. Referee #4 notes that it remains unclear whether pppGpp underlies the growth defect in the absence of manY and that experiments with pppGpp null mutant strains are required to support this conclusion, alongside some other analyses to further support this link. Reviewer #2 also notes that the CFU data remain very variable which may have implications for the robustness of the data. We feel that these are critical points which must be addressed in a revised manuscript. The rest of the referees' reports are clear and the remaining issues should be straightforward to address. We note that Referee #2 was concerned that analysis of the immune component of the response had been overlooked. Though we agree that this would develop and expand the manuscript and our understanding of the host response to streptococcal CNS infection, it would not be an essential component and its lack would not preclude us returning the revised manuscript to reviewers.

If you have not done so already please begin to revise your manuscript so that it conforms to our Article format instructions at <http://www.nature.com/nmicrobiol/info/final-submission/>

The usual length limit for a Nature Microbiology Article is six display items (figures or tables) and 3,500 words. We have some flexibility, and can allow a revised manuscript at 4000 words, but please consider this a firm upper limit.

Please include a data availability statement as a separate section after Methods but before references, under the heading "Data Availability". This section should inform readers about the availability of the data used to support the conclusions of your study. This information includes accession codes to public repositories (data banks for protein, DNA or RNA sequences, microarray, proteomics data etc...), references to source data published alongside the paper, unique identifiers such as URLs to data repository entries, or data set DOIs, and any other statement about data availability. At a minimum, you should include the following statement: "The data that support the findings of this study are available from the corresponding author upon request", mentioning any restrictions on availability. If DOIs are provided, we also strongly encourage including these in the Reference list (authors, title, publisher (repository name), identifier, year). For more guidance on how to write this section please see: <http://www.nature.com/authors/policies/data/data-availability-statements-data-citations.pdf>

To improve the accessibility of your paper to readers from other research areas, please pay particular attention to the wording of the paper's opening bold paragraph, which serves both as an introduction and as a brief, non-technical summary in about 150

words. If, however, you require one or two extra sentences to explain your work clearly, please include them even if the paragraph is over-length as a result. The opening paragraph should not contain references. Because scientists from other sub-disciplines will be interested in your results and their implications, it is important to explain essential but specialised terms concisely. We suggest you show your summary paragraph to colleagues in other fields to uncover any problematic concepts.

If your paper is accepted for publication, we will edit your display items electronically so they conform to our house style and will reproduce clearly in print. If necessary, we will re-size figures to fit single or double column width. If your figures contain several parts, the parts should form a neat rectangle when assembled. Choosing the right electronic format at this stage will speed up the processing of your paper and give the best possible results in print. We would like the figures to be supplied as vector files - EPS, PDF, AI or postscript (PS) file formats (not raster or bitmap files), preferably generated with vector-graphics software (Adobe Illustrator for example). Please try to ensure that all figures are non-flattened and fully editable. All images should be at least 300 dpi resolution (when figures are scaled to approximately the size that they are to be printed at) and in RGB colour format. Please do not submit Jpeg or flattened TIFF files. Please see also 'Guidelines for Electronic Submission of Figures' at the end of this letter for further detail.

Figure legends must provide a brief description of the figure and the symbols used, within 350 words, including definitions of any error bars employed in the figures.

When submitting the revised version of your manuscript, please pay close attention to our [href="https://www.nature.com/nature-research/editorial-policies/image-integrity">Digital Image Integrity Guidelines.](https://www.nature.com/nature-research/editorial-policies/image-integrity) and to the following points below:

EXTENDED DATA FIGURES

Please include a statement before the acknowledgements naming the author to whom correspondence and requests for materials should be addressed.

Finally, we require authors to include a statement of their individual contributions to the paper -- such as experimental work, project planning, data analysis, etc. -- immediately after the acknowledgements. The statement should be short, and refer to authors by their initials. For details please see the Authorship section of our joint Editorial policies at http://www.nature.com/authors/editorial_policies/authorship.html

- * include a point-by-point response to any editorial suggestions and to our referees. Please include your response to the editorial suggestions in your cover letter, and please upload your response to the referees as a separate document.
- * ensure it complies with our format requirements for Letters as set out in our guide to authors at www.nature.com/nmicrobiol/info/gta/
- * state in a cover note the length of the text, methods and legends; the number of references; number and estimated final size of figures and tables
- * resubmit electronically if possible using the link below to access your home page:

Link Redacted

*This url links to your confidential homepage and associated information about manuscripts you may have submitted or be reviewing for us. If you wish to forward this e-mail to co-authors, please delete this link to your homepage first.

Please ensure that all correspondence is marked with your Nature Microbiology reference number in the subject line.

Nature Microbiology is committed to improving transparency in authorship. As part of our efforts in this direction, we are now requesting that all authors identified as 'corresponding author' on published papers create and link their Open Researcher and Contributor Identifier (ORCID) with their account on the Manuscript Tracking System (MTS), prior to acceptance. This applies to primary research papers only. ORCID helps the scientific community achieve unambiguous attribution of all scholarly contributions. You can create and link your ORCID from the home page of the MTS by clicking on 'Modify my Springer Nature

account'. For more information please visit www.springernature.com/orcid.

We hope to receive your revised paper within 2 months. If you cannot send it within this time, please let us know.

Yours sincerely,

Reviewers Comments:

Reviewer #1 (Remarks to the Author):

All comments were properly addressed. I acknowledge the excellent work done by the authors.

Minor comments:

1. Fig 5: Authors state that the increased virulence is due to a higher growth rate locally (in the brain). Yet with a bacteremia-derived meningitis model is not really possible to assess whether an increase of bacterial load in the brain is due to higher levels of bacteremia (more bacteria in the blood = more bacteria invading the brain). The only way to assess whether higher disease severity in the brain is due to a higher bacterial growth in the brain, is either to use antibiotics (that do not pass the BBB) for clearing bacteremia, or to use an ic model.
2. Western blot figures: the western blot data should be accompanied by quantification analysis of the protein bands (based on the loading controls).

Reviewer #2 (Remarks to the Author):

While the study presents an interesting hypothesis regarding the ability of *Streptococcus equi* subspecies *zooepidemicus* (SEZ) to proliferate in cerebrospinal fluid (CSF) and explores its glucose acquisition mechanisms—particularly through the phosphotransferase system (PTS_{man})—I have significant concerns regarding the validity of the findings and the robustness of the analysis.

Major Concerns:

1. Bacterial Load Estimates

The authors responded to my concern by stating that only one mouse had a bacterial load of 10^{10} CFU, while the rest ranged from 10^7 to 10^{10} , with an average of 10^9 CFU. However, this raises concerns about plausibility and statistical consistency. If the average CFU count is 10^9 , most values should cluster around this number. However, with a range spanning three orders of magnitude (10^7 to 10^{10}), this suggests substantial variability. If most values were closer to 10^7 , then the average would likely be lower than 10^9 . Conversely, if most values were near 10^{10} , then 10^7 would be an extreme outlier. A 10^3 -fold variation in bacterial load among a small group of mice is unexpected unless specific factors—such as infection severity or immune response—differ significantly.

2. Lack of Immune System Analysis

The study does not analyze the immune response to such a high bacterial burden. Given that the brain has a well-regulated immune environment, including microglial activation and cytokine responses, the absence of any discussion or data on immune system involvement weakens the study. The authors should include a thorough immunological analysis to support their claims. The authors responded to my concern by stating that the brain is considered an immune-privileged organ. While the brain has historically been considered an immune-privileged organ with limited immune cell recruitment, this concept has evolved significantly. It is now well established that the brain has an active and dynamic immune environment, with resident immune cells such as microglia playing key roles in immune surveillance and response. Additionally, under pathological conditions or infection, peripheral immune cells—including macrophages and T cells infiltrate the brain, challenging the notion of strict immune privilege. Thus, any study involving bacterial presence in the brain should thoroughly assess immune activation and potential inflammatory responses.

Therefore, I do not find this study suitable for publication in its current form.

Reviewer #3 (Remarks to the Author):

Overall, the authors have done a solid job responding to critiques. Replies to their edits/rebuttals in response to my critiques from the first submission are below.

(25) Histopathology addressed the comment about the pathological significance of the manY knockout given there is no change in mortality. Quantification would be appreciated as it is difficult to determine significance of differences in pathology between manY knockout and WT and it is not clear if images are representative of all individuals in the experiment. If penicillin is the first line antibiotic for Streptococcal bacterial meningitis, then additional data is convincing to show significance of manY to CNS infection.

(26) Practical challenges of stereotaxic route noted. Appreciated the validation of manY knockout through this route at minimum.

(27) Additional data and applied statistics seemed appropriate and addressed concerns that were brought up.

(28) Additional data 14 :

NEED to define what WT::pSET2 and WT::pSET2-reIA are. This is not defined explicitly in the text or figure legend just in methods but not clearly.

Lines 368-371 not accurately describing what is seen in the data.

I can agree with the first part of the statement: "Moreover, an increase in ppGpp was observed in Δ manY cultured under low glucose conditions"

I do not agree with the second part of the statement. There are spots that seem like ppGpp in those columns at the location that ppGpp is labeled. Maybe less ppGpp than the low glucose concentrations though. Also there was some fluorescence in those conditions with the reporter too (even if it is less): "whereas ppGpp was hardly detectable in WT or Δ manY when grown in higher glucose concentrations by either TLC (Fig. 6c) or an RNA-based fluorescent sensor (Fig. 6d)."

Additional data 9 and 10 made HPr phosphorylation at low glucose concentrations, crp involvement, and relA connection via cAMP clear. This provided more supporting evidence that the SEZ manY system prevents the stringent response at low glucose concentrations in WT SEZ. Additional data 15 was very much appreciated as it was hard to tell the difference between the ppGpp on the TLC. I think you could see the pppGpp difference on the TLC.

Schematic (additional data 8) is good.

Other critiques:

(29) Addressed well in 23. Interesting that the same concentration of bacteria was not used between *S. agalactiae*, *S. pneumoniae*, and SEZ. However, the bacterial inoculum for SEZ was lower than *S. agalactiae* and *S. pneumoniae* and still exhibited a significant difference between abundance in the brain only in SEZ and not *S. agalactiae* and *pneumoniae*, so I think it is okay.

(30) Addressed in 26 above.

(31) Initially was confused about the 155 notations. Looks really good but please add an indication of the specific nucleotide you are looking at in panel B in panel A. It took me a while to find that one nucleotide and the "155" does not line up with the negative numbers (I think it is -133). Adding a box to indicate the nucleotide you are looking at in A would be helpful.

(32) Addressed concern very well. Different layout of this data was appreciated.

(33-35) Concerns addressed well.

(36-44) All text edits addressed.

(45) Addressed with histopathology in 25.

(46) Addressed.

Reviewer #4 (Remarks to the Author):

For me the authors show convincing data that ManY transport system is required for glucose uptake in CSF and essential for growth in the brain. Later on the author link this observation to the stringent response. Here are some comments on this part of manuscript: It is likely that in the CSF stringent response is activated when the bacteria become limited for glucose or amino acids e.g. in the manY mutant. For *Streptococcus suis*, it was already shown that low glucose somehow results in stringent response activation (<https://www.ncbi.nlm.nih.gov/pubmed/27255540>).

The data presented do not actually show if and how (p)ppGpp is required for the observed growth defect of the manY mutant (no data using a pppGpp0 strain are provided).

Specific comments:

The authors claim that their transcriptome data (Wt versus manY under low glucose conditions) mimic stringent response gene expression. This assumption is hard to validate based on the presented data. First, no data (E.g. Excell sheet) including annotations and significance of genes are provided. How are stringent genes divined? Based on previous RNAseq-Data? It is required to show the overlap of gene expression pattern (many versus pppGpp0 strains) in a more quantitative way to judge the assumption. One hallmark of the stringent response is the downregulation of ribosomal proteins. However, these genes might also be down-regulated via other mechanisms under starving conditions. Thus, the real (p)ppGpp dependent (stringent) genes should be evaluated by comparison with transcriptome data from WT and pppGpp0 strains. This data might be available from previous publication from other streptococcal species. E.g (<https://www.ncbi.nlm.nih.gov/pubmed/27255540>).

Line 365: The link to stringent response is mainly made by the observation that rel (coding for the bifunctional Rel) was upregulated in the manY mutant. However, Rel is a bifunctional enzyme with (p)ppGpp synthase and hydrolysis activity. Thus, the (p)ppGpp level is determined via post-translational activation. (Through interaction with the ribosome and uncharged tRNA) of the synthetase activity and concurrent inhibition of the hydrolase activity http://www.ncbi.nlm.nih.gov/entrez/query.fcgi?cmd=Retrieve&db=PubMed&dopt=Citation&list_uids=15066282). Thus, transcriptional regulation of rel may lead to more or less (p)ppGpp dependent on the activation status of the enzyme.

Fig 6B shows that Rel overexpression results in growth defect. Assuming that Rel synthetase is active under low glucose conditions the data is trivial. (p)ppGpp is one of the main regulators to halt growth (inhibition of translation, replication...).

However, the data does not allow direct conclusion that the growth defect in the many mutant is caused by (p)ppGpp. The growth

defect can easily be explained via the decrease of glucose in the cell e.g via ATP depletion.

Fig 6C: Data are provided to show that glucose results in stringent response activation by direct detection of (p)ppGpp under low glucose conditions (Fig 6C). This data is not quantified. Only one TLC result is shown. Quantitative data based on at least three independent experiments should be shown.

Figure 6E: The meaning of the schematic figure is not very clear. What would be the role of ManY in this context

Reviewer #5 (Remarks to the Author):

The manuscript by Yuan et al. describes a mechanism behind how pathogenic *Streptococcus equi zooepidemicus* (SEZ) can replicate to high levels in CSF. Using a genetic barcoding approach, the authors reveal that the CSF is colonized by a small founding population (FP) following bloodstream infection. Once in the CSF, that small FP replicates to high levels. While other pathogenic streptococci can access the CSF, they don't replicate to as high of bacterial burden. The ability to replicate to these high levels, in part, is because of the unique increased expression of the PTSman system. The increased expression results from specific polymorphisms in the PTSman promoter. The authors finish by demonstrating this increased production of the PTSman system allows for increased replication in minimal nutrient conditions with low glucose, like those found in the CSF. Increased glucose import by the PTSman system provides enough nutrients for the cell to prevent activation of the stringent response. I found the manuscript well written, the figures easy to interpret, and the experiments thorough.

After reviewing the document and rebuttal to prior reviews, I feel the authors have addressed all of the previous reviewer's concerns. I only have minor comments that need to be addressed.

1. Line 197 – “genes that were linked to metabolism (marked with asterisks...) – I didn't see any asterisks in the figure.
2. Line 203 – This sentence refers to 24hr CFU counts of the Δ proB and Δ proC strains and references Fig1a and Fig. S4b. While the statement made in the sentence may be true, Fig. S4b is looking 14 days post infection, not 24 hr. Fig. 1a only shows WT data at 24 hr
3. Line 235 – This sentence refers to bacterial burden outside the CNS upon antibiotic treatment. I did not see any data where bacterial burdens were determined with antibiotic treatment, only survival curves. I am also confused as to the greater survival of the Δ manY mutant with penicillin treatment compared to WT. The authors state that both WT and the Δ manY mutant are equally susceptible to penicillin (line 235). If without penicillin treatment, both WT and Δ manY burdens are the same outside the CNS (Fig 3C), and both cause the same mortality... then why is there a difference in survival with penicillin treatment? Do the authors think this difference in mortality is due to likely differences in bacterial burden in the brain (as these are different to begin with in the absence of antibiotics)? If not, it is unclear how the sentence beginning in line 236 regarding the benefits of antibiotic treatment being enhanced by targeting nutrient uptake systems can be true. This should be clarified.
4. Line 253 should be Fig. S4c not Fig. 4c
5. Line 397 should be “promoter”

Version 2:

Reviewer comments:

Reviewer #1

(Remarks to the Author)

It is clear that, with a cleared bacteremia, there is a significant (two logs) of bacterial load in the brain between the wt and the mutant. Yet is this due to better bacterial growth of the wt in the brain, or to a better penetration capabilities across the BBB? Authors should grow the two strains (wt, mutant) using brain tissue homogenate as culture medium and assess bacterial growth over time.

Additionally, if wt and mutant are administered onto brain endothelial cells in vitro, what strain is capable of adhering more/better to the cells?

These are quick experiments to be performed and would represent an important evidence in support (or not) of the conclusions drawn so far by the authors. I propose one last minor revision.

Reviewer #4

(Remarks to the Author)

The authors now provided more convincing evidence regarding the stringent response. I only suggest to tone down the conclusion provided in the summary figure. There are no in vivo data using a pppGpp mutant. Thus, it remains unproven whether the circuit is working in vivo.

Minor: Figr 8E seems to miss the description in the legend.

Decision Letter:

12th September 2025

Dear Professor Ma,

Thank you for your patience while your manuscript "Constitutive glucose import in zoonotic Streptococci enables proliferation in cerebrospinal fluid" was under peer-review at Nature Microbiology. It has now been seen by 2 referees, whose expertise and comments you will find at the of this email. You will see from their comments below that while they find your work of interest, some concerns remained. As discussed over email with you, you also know that we engaged in further discussion with some of the reviewers, as in additional discussion, continued concerns remained about the variability in the bacterial loads (CFU data) obtained in the CSF and brain smaples from the in vivo work in mice. After this process of consultation, we remain very interested in the possibility of publishing your study in Nature Microbiology, but would like to consider your response to the points listed below in the form of a revised manuscript (and accompanying rebuttal) before we make a final decision on publication.

First of all, Reviewer #1 continued to raise concerns and ask for additional experiments to show that bacteria were growing more in the brain rather than translocating across. After additional discussion, we have decided to overrule this request for additional experimental work. The bar-coding analyses provide evidence that seeding of the brain involves few clones, showing that expansion must occur subsequently – thus there is no need to do the additional experiments suggested at this later stage of review.

Reviewer #4 notes that there is no in vivo work with a (p)ppGpp mutant strain, and asks that the resulting conclusions and figure presenting the overall model be toned down.

Regarding the remaining concerns from Reviewers #1 and Reviewer #2 (from the earlier rounds of review) regarding variability in CFUs, we consulted the previous Reviewer #3 to get an additional point of view. They suggested though the data was variable, your responses to this were plausible. However they noted that from looking at the graphs, the data might not be normally distributed, and that tests to confirm whether data are parametric (normally distributed) should be done. If the data are not normally distributed, the use of parametric ANOVAs would not be appropriate and non-parametric statistical tests should be applied. We would ask you to address this point: confirm whether the in vivo CFU data are normally distributed, if not, re-do the statistical analysis with appropriate tests to confirm whether the differences observed remain statistically significant.

Lastly, we would ask you to double check that the updated version of the manuscript includes the full methods of the scoring for the histopathology, including the use of blinding and a trained pathologist, as confirmed by yourselves in previous emails.

We believe that these changes should be feasible, but are important to ensure robustness of the study and its main conclusions.

If you have not done so already please begin to revise your manuscript so that it conforms to our Article format instructions at <http://www.nature.com/nmicrobiol/info/final-submission/>

The usual length limit for a Nature Microbiology Article is six display items (figures or tables) and 3,000 words. We have some flexibility, and can allow a revised manuscript at 3,500 words, but please consider this a firm upper limit. There is a trade-off of ~250 words per display item, so if you need more space, you could move a Figure or Table to Supplementary Information.

Some reduction could be achieved by focusing any introductory material and moving it to the start of your opening 'bold' paragraph, whose function is to outline the background to your work, describe in a sentence your new observations, and explain your main conclusions. The discussion should also be limited. Methods should be described in a separate section following the discussion, we do not place a word limit on Methods.

Nature Microbiology titles should give a sense of the main new findings of a manuscript, and should not contain punctuation. Please keep in mind that we strongly discourage active verbs in titles, and that they should ideally fit within 90 characters each (including spaces).

Please include a data availability statement as a separate section after Methods but before references, under the heading "Data Availability". This section should inform readers about the availability of the data used to support the conclusions of your study. This information includes accession codes to public repositories (data banks for protein, DNA or RNA sequences, microarray, proteomics data etc...), references to source data published alongside the paper, unique identifiers such as URLs to data repository entries, or data set DOIs, and any other statement about data availability. At a minimum, you should include the following statement: "The data that support the findings of this study are available from the corresponding author upon request", mentioning any restrictions on availability. If DOIs are provided, we also strongly encourage including these in the Reference list (authors, title, publisher (repository name), identifier, year). For more guidance on how to write this section please see: <http://www.nature.com/authors/policies/data/data-availability-statements-data-citations.pdf>

To improve the accessibility of your paper to readers from other research areas, please pay particular attention to the wording of the paper's opening bold paragraph, which serves both as an introduction and as a brief, non-technical summary in about 150

words. If, however, you require one or two extra sentences to explain your work clearly, please include them even if the paragraph is over-length as a result. The opening paragraph should not contain references. Because scientists from other sub-disciplines will be interested in your results and their implications, it is important to explain essential but specialised terms concisely. We suggest you show your summary paragraph to colleagues in other fields to uncover any problematic concepts.

If your paper is accepted for publication, we will edit your display items electronically so they conform to our house style and will reproduce clearly in print. If necessary, we will re-size figures to fit single or double column width. If your figures contain several parts, the parts should form a neat rectangle when assembled. Choosing the right electronic format at this stage will speed up the processing of your paper and give the best possible results in print. We would like the figures to be supplied as vector files - EPS, PDF, AI or postscript (PS) file formats (not raster or bitmap files), preferably generated with vector-graphics software (Adobe Illustrator for example). Please try to ensure that all figures are non-flattened and fully editable. All images should be at least 300 dpi resolution (when figures are scaled to approximately the size that they are to be printed at) and in RGB colour format. Please do not submit Jpeg or flattened TIFF files. Please see also 'Guidelines for Electronic Submission of Figures' at the end of this letter for further detail.

Figure legends must provide a brief description of the figure and the symbols used, within 350 words, including definitions of any error bars employed in the figures.

When submitting the revised version of your manuscript, please pay close attention to our [href="https://www.nature.com/nature-research/editorial-policies/image-integrity">Digital Image Integrity Guidelines.](https://www.nature.com/nature-research/editorial-policies/image-integrity) and to the following points below:

EXTENDED DATA FIGURES

Please include a statement before the acknowledgements naming the author to whom correspondence and requests for materials should be addressed.

Finally, we require authors to include a statement of their individual contributions to the paper -- such as experimental work, project planning, data analysis, etc. -- immediately after the acknowledgements. The statement should be short, and refer to authors by their initials. For details please see the Authorship section of our joint Editorial policies at http://www.nature.com/authors/editorial_policies/authorship.html

- * include a point-by-point response to any editorial suggestions and to our referees. Please include your response to the editorial suggestions in your cover letter, and please upload your response to the referees as a separate document.
- * ensure it complies with our format requirements for Letters as set out in our guide to authors at www.nature.com/nmicrobiol/info/gta/
- * state in a cover note the length of the text, methods and legends; the number of references; number and estimated final size of figures and tables
- * resubmit electronically if possible using the link below to access your home page:

Link Redacted

*This url links to your confidential homepage and associated information about manuscripts you may have submitted or be reviewing for us. If you wish to forward this e-mail to co-authors, please delete this link to your homepage first.

Please ensure that all correspondence is marked with your Nature Microbiology reference number in the subject line.

Nature Microbiology is committed to improving transparency in authorship. As part of our efforts in this direction, we are now requesting that all authors identified as 'corresponding author' on published papers create and link their Open Researcher and Contributor Identifier (ORCID) with their account on the Manuscript Tracking System (MTS), prior to acceptance. This applies to primary research papers only. ORCID helps the scientific community achieve unambiguous attribution of all scholarly contributions. You can create and link your ORCID from the home page of the MTS by clicking on 'Modify my Springer Nature

account'. For more information please visit please visit www.springernature.com/orcid.

We hope to receive your revised paper within three weeks. If you cannot send it within this time, please let us know.

Yours sincerely,

Reviewers Comments:

Reviewer #1 (Remarks to the Author):

It is clear that, with a cleared bacteremia, there is a significant (two logs) of bacterial load in the brain between the wt and the mutant. Yet is this due to better bacterial growth of the wt in the brain, or to a better penetration capabilities across the BBB? Authors should grow the two strains (wt, mutant) using brain tissue homogenate as culture medium and assess bacterial growth over time.

Additionally, if wt and mutant are administered onto brain endothelial cells in vitro, what strain is capable of adhering more/better to the cells?

These are quick experiments to be performed and would represent an important evidence in support (or not) of the conclusions drawn so far by the authors. I propose one last minor revision.

Reviewer #4 (Remarks to the Author):

The authors now provided more convincing evidence regarding the stringent response. I only suggest to tone down the conclusion provided in the summary figure. There are no in vivo data using a pppGpp mutant. Thus, it remains unproven whether the circuit is working in vivo.

Minor: Figr 8E seems to miss the description in the legend.

Version 3:

Decision Letter:

Our ref: NMICROBIOL-24072029C

24th September 2025

Dear Dr. Ma,

Thank you for submitting your revised manuscript "Constitutive glucose import in zoonotic Streptococci enables proliferation in cerebrospinal fluid" (NMICROBIOL-24072029C). It has now been seen by the original referees and their comments are below. The reviewers find that the paper has improved in revision, and therefore we'll be happy in principle to publish it in Nature Microbiology, pending minor revisions to satisfy the referees' final requests and to comply with our editorial and formatting guidelines.

Thank you again for your interest in Nature Microbiology Please do not hesitate to contact me if you have any questions.

Sincerely,

Version 4:

Decision Letter:

17th October 2025

Dear Professor Ma,

I am pleased to accept your Article "Zoonotic Streptococcus imports glucose to inhibit stringent response and promote growth during meningitis" for publication in Nature Microbiology. Thank you for having chosen to submit your work to us and many congratulations.

Over the next few weeks, your paper will be copyedited to ensure that it conforms to Nature Microbiology style.

Authors may need to take specific actions to achieve compliance with funder and institutional open access mandates. If your research is supported by a funder that requires immediate open access (e.g. according to [Plan S principles](https://www.springernature.com/gp/open-science/plan-s-compliance) or the [NIH public access policy](https://www.springernature.com/gp/open-science/us-federal-agency-compliance)) then you should select the gold OA route, and we will direct you to the compliant route where possible. Because authors warrant under our subscription licensing terms that they haven't committed to licensing any version of their article under a licence inconsistent with the terms of our agreement – including the applicable embargo period – publication under the subscription model isn't suitable for authors whose funders require no embargo.

With kind regards,

P.S. Click on the following link if you would like to recommend Nature Microbiology to your librarian
<http://www.nature.com/subscriptions/recommend.html#forms>

** Visit the Springer Nature Editorial and Publishing website at http://editorial-jobs.springernature.com?utm_source=ejP_NMicro_email&utm_medium=ejP_NMicro_email&utm_campaign=ejp_NMicro for more information about our career opportunities. If you have any questions please click [here](mailto:editorial.publishing.jobs@springernature.com).**

We thank the reviewers for their insightful comments. We have conducted a number of additional experiments, listed below:

Additional Data Collections:

Additional Data #1 Comparison of the CFU in organs between perfused and non-perfused mice (New Fig. s1a)

Organ CFU burden at 18 hpi and at moribund stage following tail vein injection of 5×10^6 CFU SEZ. Cardiac perfusion was performed until the liver appeared pale (over 20 mL perfusate) or blood was drained as fully as possible after anesthesia with pentobarbital. Statistical significance was analyzed using two-way ANOVA with Šidák's multiple comparisons test.

Additional Data #2 Growth curve of SEZ WT and its derivatives in swine CSF with different carbohydrates (New Fig. s5c)

Growth curve of SEZ, $\Delta manY$ and $C\Delta manY$ cultured in swine CSF supplemented with five different carbohydrates in vitro (n = 3).

Additional Data #3 CFU burdens in the CSF of moribund mice (Formerly Fig. 1e, New Fig. 1d)

CFU burden in the CSF of moribund mice (n = 11).

Additional Data #4 H&E staining of brains infected with WT or $\Delta manY$ SEZ (New Fig. 3e).

H&E staining of brain sections from moribund mice intravenously infected with WT or $\Delta manY$ strains. The left panel shows low-magnification views of brain sections, with black squares indicating regions selected for detailed analysis across WT, $\Delta manY$, and vehicle (blank) inoculations. The magnified views in subsequent columns correspond to the meninges, ventricle, and olfactory bulb regions, arranged sequentially for consistent comparison. Red arrows indicate areas of hemorrhage. Blue arrows highlight neutrophil infiltration. WT exhibits obvious pathological features such as neutrophil infiltration and hemorrhage, while $\Delta manY$ shows mild pathology.

Additional Data #5 Survival curve of mice treated with SEZ or $\Delta manY$ following penicillin treatment (a, New Fig. 3f)

Survival curve of C57BL/6J mice after intravenous administration of 200 µg penicillin every 12 hours, beginning 24 hpi. Mice were challenged i.v. with 5×10^6 CFU WT or $\Delta manY$, and subsequently received penicillin or vehicle (PBS) treatment starting at 24 hpi (*p*-value were analyzed with the simple survival analysis following a log-rank test).

Additional Data #6 The distributions of transposon insertions within the PTS operon containing the 6210 gene (New Fig. s3c)

Comparison of the distributions of transposon insertions in another mannose PTS operon (RS06200~RS06215) and the upstream and downstream gene in the input library, as well as in three rounds of the output samples. The RS06210 gene is marked in red. The yellow background denotes the coding sequence (CDS) and intergenic regions. TA sites indicate potential transposon insertion site.

Additional Data #7 Growth assessment by CFU counting at stationary phase of SEZ WT and its derivatives (a, New Fig. 5f); the CFU burden of moribund mice after challenge with WT SEZ or SEZ::P_{manX(A909)} (b, Formerly Fig. 5e, New Fig. 5g)

a, The CFU of WT, $\Delta manY$, $C\Delta manY$ and SEZ::P_{manX(A909)} (n = 3) at stationary phase (cultured 12 h) in CDM with 0.2, 1, 2 and 10 g/L glucose respectively (one-way ANOVA followed by Dunnett's multiple comparisons test). **b**, C57BL/6J mice were i.v. injected with SEZ (n = 4) or SEZ::P_{manX(A909)} (n = 4, moribund between 31 and 37 hpi), CFU was counted from organs of moribund mice. (two-way ANOVA followed by Šídák's multiple comparisons test).

Additional Data #8 Graphical schematic of how PTS_{man} influences the stringent response in low glucose (New Fig. 6i)

Schematic model illustrating SEZ inhibition of the stringent response via the PTS_{man} system. Under low-glucose conditions, SEZ actively transports glucose through PTS_{man}, leading to phosphorylation of HPr, a component of the PTS system. The high ratio of phosphorylated HPr suppresses Crp expression, thereby inhibiting the activation of *relA* transcription by the cAMP complex. Reduced *relA* transcription limits ppGpp accumulation, effectively preventing the onset of the stringent response. However, $\Delta manY$ was unable to transport sufficient glucose to maintain a high ratio of P-HPr, resulted in Crp expression and then active the transcription of *relA*, ultimately resulting in ppGpp accumulation. Schematic is created with BioRender.com.

Additional Data #9 Detection of Crp binding to the promoter of *relA* with or without cAMP (a, New Fig. 6f); qPCR analysis of *relA* expression levels in SEZ WT and its derivatives cultured in CDM with 1 g/L glucose until log phase (b, New Fig. 6g); TLC detection of ppGpp in WT, Δcrp , $\Delta manY$ and $\Delta manY\Delta crp$ cultured in CDM with 1 g/L glucose (c, New Fig. 6h).

a, EMSA analysis assessing the binding of the *relA* promoter to Crp or the Crp:cAMP complex. **b**, Relative transcript levels of *relA* in WT, Δcrp , $\Delta manY$ and $\Delta manY\Delta crp$ strains, as detected by qRT-PCR. The data were normalized to the *recA* transcript (n = 3, one-way ANOVA followed by Dunnett's multiple comparisons test). All strains were collected at mid-exponential phase. **c**, Intracellular levels of (p)ppGpp were detected by TLC after nucleotide extraction of SEZ WT, Δcrp , $\Delta manY$, $\Delta manY\Delta crp$ strains. Bacteria were grown in CDM with 1 g/L glucose.

Additional Data #10 The HPr phosphorylation status (top) and the intracellular accumulation of Crp (bottom) in SEZ and its derivatives following incubation in CDM with 1 g/L or 10 g/L glucose (New Fig. 6e)

Western blot analysis of HPr phosphorylation ratio and Crp expression levels in WT, $\Delta manY$, $C\Delta manY$ and SEZ::P_{manX}(A909) strains grown in CDM with 1 g/L or 10 g/L glucose. GroL was used as a loading control. All samples were collected at mid-exponential phase.

Additional Data #11 The brain bacterial burden of another two meningeal pathogens, *Streptococcus agalactiae* A909 and *Streptococcus pneumoniae* D39 (New Fig. s1e, s1f)

a, Survival curve of C57BL/6J mice intravenously injected with *S. agalactiae* A909 (10^7 and 10^8 CFU) or *S. pneumoniae* D39 (10^6 or 10^7 CFU). **b**, Organ CFU counts in mice i.v. injected with 5×10^8 CFU A909 (moribund between 24 and 72 hpi, n=3) or 5×10^7 CFU of D39 (moribund between 24 and 120 hpi, n=3). Survival curves were analyzed using the simple survival analysis following a log-rank test vs. WT. CFU burden of each organ were analyzed using one-way ANOVA followed by Dunnett's multiple comparisons test.

Additional Data #12 Sequence alignment of the promoter regions of different streptococci, from 10 bp upstream of the -35 element to 10 bp downstream of -10 region (a, New Fig. s6c); Promoter activities of P_{manX} mutants (b, New Fig. 5c).

a, Nucleotide sequence homology of PTS_{man} promoter from 10 bp upstream of -35 region to 10 bp downstream of -10 region (-152 to -104 bp) across *Streptococcus*, with strains shown on the left. The -35 and -10 regions are marked with red squares. The nucleotide homology is indicated by different shades of blue, with a threshold set at 50%. A “-” indicated a gap base and “.” Indicated the consensus base exceeding the 50% threshold. **b**, The nucleotide sequence is the alignment of -152 to -104 region in P_{manX} of GAS and GCS, there are only two nucleotides mutant. Site-directed mutants were induced into the SEZ P_{manX} promoter according to the same region of GAS and cloned into the pTCV-lacZ vector to identify the essential of SEZ P_{manX} conservation to its high transcriptional activity.

Additional Data #13 Growth curve (n = 4) of WT, Δ manY, C Δ manY and SEZ::P_{manX}(A909) in CDM with 0.2, 1, 2 and 10 g/L glucose. (New Fig. 5e).

Growth curve (n = 4) of WT, Δ manY, C Δ manY and SEZ::P_{manX}(A909) in CDM with 0.2, 1, 2 and 10 g/L glucose. The p-values were analyzed using a one-way ANOVA with Dunnett’s multiple comparisons test applied to growth curves fitted by the Gompertz nonlinear regression model.

Additional Data #14 New TLC detection of ppGpp in WT, $\Delta manY$ and $C\Delta manY$ cultured in CDM with different glucose concentrations (New Fig. 6c).

Intracellular levels of (p)ppGpp were detected by TLC after nucleotide extraction of SEZ WT, $\Delta manY$ and $C\Delta manY$ strains. Bacteria were grown in CDM with low (1 g/L) or high (10 g/L) concentrations of glucose.

Additional Data #15 Quantification of in vivo ppGpp concentration with the S2 sensor plasmid (New Fig. 6d).

Fluorescence imaging of (p)ppGpp biosynthesis in SEZ, $\Delta manY$, $C\Delta manY$ and SEZ::P_{manX(A909)} strains under 1 or 10 g/L glucose. All strains were grown to mid-exponential phase in CDM containing 1 or 10 g/L glucose, and fluorescence images were captured with a 100 ms exposure time at 30% shift. GFP signal ranged from 200 to 500 and Bright field ranged from 800 to 4000. Identical imaging conditions and settings were applied across all figures, as detailed in the supplementary materials.

Response to Reviewers:

Reviewer #1 (Remarks to the Author). Author responses are in blue.

GENERAL COMMENT

An interesting study by Yuan et al investigating the pathogenesis of *S. equi* subsp. *Zooepidemicus* meningitis, a disease of great societal and health impact in farm animals (pigs) and humans. Authors have interestingly studied the interplay between the metabolic properties of the bacteria within the host and disease pathogenesis. Yet the study presents technical and scientific unclear points. My comments, remarks and suggestions are the following:

Thank you for the positive and constructive comments on our paper.

TEXTUAL COMMENTS

(1) Line 42: Streptococcus IS a common cause

We have revised this sentence to "Several species within the genus *Streptococcus* cause bacterial meningitis." (Lines 42).

(2) Line 43: Streptococcus pneumonia (the pneumococcus)

We have changed "Streptococcus pneumonia" to "*Streptococcus pneumoniae*" (Lines 42).

(3) Lines 51-52: blood-CNS-barrier, do the author mean blood-brain barrier here?

The blood-CNS barrier is comprised of both the blood-brain and blood-CSF barriers (Coureuil M., et al. Nat Rev Microbiol, 2017). Our paper does not address how SEZ traverses either barrier specifically, so we generalize to blood-CNS.

(4) Line 52: authors state what contemporary studies focus on, but biologically speaking, what are the differences between these two routes?

The Blood-CSF/Brain barriers are structurally heterogeneous according to Coureuil Mathieu et al Nature Reviews Microbiology 2017. The Blood-CSF barrier is generally more permeable than the Blood-Brain barrier due to structural differences, such as the fenestrated capillaries of the choroid plexus. Thus, the mechanisms by which bacteria cross the Blood-CSF or Blood-Brain barriers could be distinct. In our study, SEZ accumulates in the CSF, and we cannot be certain as to the specific penetration route, so we used Blood-CNS barriers in this study.

METHODS

(5) Animal experiments. Experimental meningitis can be induced in vivo either systemically (bacteremia-derived meningitis mouse model) or through intracisternal administration of bacteria directly into the csf. Even though bacteria can travel to the brain from the nose through the olfactory bulb, a minority of the bacteria in the nasal cavity reaches the CNS through this route. Therefore, it is not clear to me why the authors have chosen this in vivo model.

Thank you for helping us clarify this point. For most animal experiments, we used a bacteremia-derived meningitis mouse model with intravenous (i.v) injection, as well as administration of bacteria directly into the brain in the Tn-seq experiment. The intranasal inoculation model was only used in supplementary Figure S1c. Since its inclusion does not impact our conclusions, we have removed Figure S1c.

RESULTS

(6) Fig 1a. Were animals perfused before organ collection? If not, it is impossible to distinguish between bacteria in the blood stream from bacteria within the tissue. Same remarks goes for the IVIS imaging results in 1b.

Thank you for helping us clarify this point. In general, while it is true that we did not perform perfusion prior to organ collection for CFU counting experiments, perfusion was conducted in the STAMP experiment. For CFU counting, mice were deeply anesthetized and exsanguinated as fully as possible prior to organ collection. This procedure was not clearly outlined in the Methods and Materials section and we have revised this section for improved clarity (Lines 49-55).

To assess the impact of perfusion on CFU counts in our study, we conducted additional experiments using perfused mice. As shown in the figure below, the CFU counts did not differ between perfused mice and those that were only exsanguinated (**Additional Data #1**). This suggests that perfusion does not impact bacterial CFU.

IVIS imaging requires live mice, so perfusion could not be performed.

Additional Data #1 Comparison of the CFU in organs between perfused and non-perfused mice (New Fig. s1a)

Organ CFU burden at 18 hpi and at moribund stage following tail vein injection of 5×10^6 CFU SEZ. Cardiac perfusion was performed until the liver appeared pale (over 20 mL perfusate) or blood was drained as fully as possible after anesthesia with pentobarbital. Statistical significance was analyzed using two-way ANOVA with Šídák's multiple comparisons test.

(7) Fig 2. Why was the SEZstamp library injected iv, while the infection mouse model is based on an intranasal challenge? Again, if organs were collected from non-perfused mice, it is very hard to analyze SEZ infection dynamics with barcoded bacteria since it is not possible to distinguish the blood-borne bacteria that have infiltrated into tissues.

As noted above, with the exceptions of the TnSeq screen (and some validations) and Figure S1C, all animal experiments in this study were conducted using i.v. injection. In the SEZ_{STAMP} library experiments, perfusion was performed before organ collection, and bacteria in the perfusate were counted as bloodborne bacteria. This method is described in the “Animal Experiments” section of the Materials and Methods (Lines 61-63). We apologize for any confusion and have included this information more clearly in the revised manuscript.

(8) Fig 3. With a more typical meningitis model (iv or ic infection), instead of the intranasal model, would have the gene identification analysis led to different results?

We used the i.v. meningitis model throughout this study (survival curves and CFU counting). The i.c. route was used for the TnSeq screen. We did not use the intranasal model in Figure 3, and the only intranasal model in formerly Figure s1c has been deleted. We apologized that our description was not clear enough.

(9) Fig 4. Growth experiments should also be performed in swine CSF (like performed in 4g) with increasing concentrations of the five carbohydrates (each carbohydrate in separate growth experiments) to assess differences in growth phenotypes.

Thank you for your suggestion. We conducted the experiment as recommended, using the five carbohydrates relevant to PTS_{man} in SEZ. The wild-type SEZ and its derivatives exhibited similar growth phenotypes in swine CSF as in Chemically Defined Media (CDM), supporting our conclusion that PTS_{man} utilizes these carbohydrates (**Additional Data #2**).

Additional Data #2 Growth curve of SEZ WT and its derivatives in swine CSF with different carbohydrates (New Fig. s5c)

Growth curve of SEZ, $\Delta manY$ and $C\Delta manY$ cultured in swine CSF supplemented with five different carbohydrates in vitro (n = 3).

(10) Fig 5. Higher or lower presence of bacteria in tissues is not necessarily linked to a higher or lower growth capability of the bacteria. A higher expression of virulence factors can also lead to a higher invasiveness without affecting growth rate. Have authors assessed this?

Thank you for this insightful comment. We agree that the ability of SEZ to invade the brain is an important factor that contributes to bacterial burden in the brain. Our barcoding experiments were designed in part to address this question. We measured founding population sizes in the brain (Figure 2A-C) and found that they were largely constant over time, which argues against the idea that new clones from outside the CNS are invading the brain. Therefore, although additional invasion events via virulence factor expression may occur, these clones do not explain the high burden in the brain, indicating that in situ replication is the primary driver of brain CFU.

Additionally, complementary experiments were performed where SEZ, $\Delta manY$ and $C\Delta manY$ strains were injected into the brain using a stereotaxic instrument, and there was no bacterial CFU

outside of the CNS. Therefore, high CFU counts were primarily due to proliferation locally. The wild-type strain reached up to 10^9 CFU, while the $\Delta manY$ CFU was significantly lower ($\sim 10^7$). Therefore, we conclude that ManY-dependent burdens in the brain are mainly attributed to the proliferative capability of SEZ rather than continued invasion.

(11) Fig S1. Bacteria in the brain has to be properly defined. If animals were not perfused, bacteria could have either been in the blood circulation within the BBB (so, not within the brain tissue yet) or within the brain tissue. IVIS imaging does not have such high resolution to distinguish the two groups.

Thank you for this helpful suggestion. We carried out additional experiments to determine the impact of perfusion on bacterial CFUs. As demonstrated in **Additional data #1**, CFU counts did not differ between perfused mice and those that were only exsanguinated. Moreover, in Figure S1, blood CFUs are 3 orders of magnitude less than the CFU in the brain, strongly indicating that blood CFU does not contribute to the 1000-fold excess of bacteria in the brain.

We agree that IVIS imaging does not have high enough resolution to distinguish the bacterial population in the blood and brain tissue. However, IVIS imaging requires live mice and perfusion cannot be performed. However, these images reveal a large amount of SEZ accumulated in the brain area, which was highly unlikely to be caused by bacteria in the blood.

Additional Data #1 Comparison of the CFU in organs between perfused and non-perfused mice (New Fig. s1a)

Organ CFU burden at 18 hpi and at moribund stage following tail vein injection of 5×10^6 CFU SEZ. Cardiac perfusion was performed until the liver appeared pale (over 20 mL perfusate) or blood was drained as fully as possible after anesthesia with pentobarbital. Statistical significance was analyzed using two-way ANOVA with Šidák's multiple comparisons test.

(12) Fig S3. Again, without animal perfusion, it is not possible to analyze the GD values since each organ could have had bacteria that were inside the blood vasculature (since each organ is vascularized). Bacteria that are in the blood stream are not yet in any other tissue, and gene expression of bacteria during the invasion process (from blood into other tissues) can change a lot.

In STAMP experiments, where GD values were calculated, perfusion was performed before organ collection. We apologize for any misunderstandings.

(13) Growth experiments, general comment. Have authors performed statistical analysis to assess differences in growths between the experimental groups?

Thank you for your suggestion. We performed statistical analysis to assess growth between different strains in the new version of manuscript.

Reviewer #2 (Remarks to the Author): Author responses are in blue.

This manuscript by Yuan et al. explores the ability of *Streptococcus equi* subspecies *zooepidemicus* (SEZ) to proliferate in cerebrospinal fluid (CSF) and uncovers the mechanisms underlying its glucose acquisition strategies, particularly through the phosphotransferase system (PTS_{man}). The study provides valuable insights into SEZ's adaptations in nutrient-limited environments and proposes potential therapeutic interventions targeting PTS_{man} to mitigate CNS infections.

The study addresses a critical gap in our understanding of how certain bacterial pathogens survive and proliferate in the nutrient-limited environment of the CSF. The focus on SEZ's glucose acquisition system, specifically the constitutive expression of PTS_{man}, is novel and offers important therapeutic implications.

While the findings are significant, the manuscript could benefit from a broader discussion on how these insights might be applied to other bacterial pathogens. The study's implications for therapeutic development, while mentioned, are not explored in sufficient depth. Additionally, comparisons with other *Streptococcus* species could be expanded to strengthen the claims about SEZ's unique adaptations.

Thank you for your positive comments on our manuscript

Specific Comments:

(14) - The use of barcoded bacteria and the STAMPR analytic framework is a strong methodological approach that allows for detailed tracking of pathogen dissemination in the CNS. The experimental design appears sound, with appropriate controls and replication to support the conclusions drawn. However, I found the study lacking novelty and excitement in certain areas. For example, the role of the immune system is never touched upon in this study, and I think it should be at least explored. Additionally, 10^{10} bacteria in the CSF seems like a very high number and might need further justification.

Thank you for your constructive suggestions. Immune responses undoubtedly play critical roles in shaping the bottlenecks to CNS infection. Indeed, early experiments using a microglial cell line BV2 revealed that *manY* did not play a role in intracellular survival, and we therefore chose to focus our study on bacterial metabolic pathways required for growth in the brain, where bacteria

are primarily extracellular. Furthermore, the brain is considered an immune-privileged organ with limited immune cell recruitment. We agree that defining the role of the immune system in CNS infection is an exciting direction for our further research.

The 10^{10} CFU/mL is indeed an extremely high burden in the CSF. However, this was the highest CFU obtained and was only observed in one mouse. We added 4 additional mice and now, with a total of 11 mice, observed that the average CFU was 10^9 , and CFUs ranged from 10^7 to 10^{10} . **(Additional Data #3).**

Additional Data #3 CFU burdens in the CSF of moribund mice (Formerly Fig. 1e, New Fig. 1d)

CFU burden in the CSF of moribund mice (n = 11).

(15) - The manuscript would benefit from more detailed explanations of certain experimental procedures. For example, the methods used to assess glucose levels and the expression of PTSman in various environments could be more thoroughly described. Additionally, the statistical analyses, while present, need to be explicitly connected to the conclusions in a clearer manner.

We have provided more details in the M&M to improve readability and clarity (Lines 270-286; Lines 328-340). We have revised the statistical analysis section to connect it more clearly to the conclusions in new version manuscript.

(16) - The discussion occasionally drifts into speculative territory, particularly regarding the potential ecological advantages of the PTSman system outside of an animal host. These claims could be more cautiously presented or supported by additional data. The role of the stringent

response in SEZ, while mentioned, could be more thoroughly explored in relation to other regulatory systems.

Thank you for your comments, we now removed the speculative language about ecological advantages. Additionally, we have conducted additional experiments to investigate additional key regulatory molecules in the stringent response and their relationship to PTS_{man} (see response to comment (28)). Our results demonstrate that the absence of PTS_{man} reduced HPr phosphorylation levels, which in turn increased Crp protein levels. The Crp::cAMP complex, acting as a transcriptional regulator, was shown activate transcription of *relA* and drive the stringent response. These findings provide additional support for the regulatory role of PTS_{man} in the stringent response.

(17) - Some references are clustered together without a clear connection to specific points in the text. These citations could be better integrated into the narrative to strengthen the arguments made.

Thank you for pointing this out. We have redistributed our citations to better integrate the references, ensuring that each citation clearly supports the specific points being made.

Minor comments:

(18) Citations: The introduction section includes some sentences with numerous citations (e.g., lines 39-40), which could be distracting. It might be helpful to either rephrase or split such sentences to reduce the density of citations.

We have reduced the density of citations. Every point has now no more than 2 citations

(19) Line 43: pneumoniae.

Spelling has been corrected. (Line 42)

(20) Lines 51-54: The sentence about contemporary studies focusing on bacterial invasion strategies could be expanded slightly to explain why this is relevant to the current study (i.e., contrasting with the lack of research on post-invasion adaptation).

We have added some information about BBB penetration to link bacterial invasion strategies to our study (Lines 52-60).

(21) Lines 103-161: The section detailing the SEZ accumulation in murine CSF is well explained, but it might benefit from more explicit connections between the experimental results and the overarching hypothesis of the study.

Response:

Thank you very much for your comments. We have improved the language in these sections (Lines 107-169)

Potential Revisions:

(22) - Further Exploration of Therapeutic Implications: Expand the discussion on how targeting PTS_{man} could be applied in therapeutic settings, perhaps considering challenges and future directions for research in this area.

As a way to gauge whether targeting PTS_{man} could have therapeutic implications, we examined the influence of antibiotic therapy in the context of *manY*-dependent infection. We found that treatment of mice with penicillin prolongs survival to a greater extent in the $\Delta manY$ strain, even though both strains are equally susceptible to penicillin in vitro. These observations suggest that disrupting bacterial replication in the CNS can improve the therapeutic efficacy and extend the therapeutic window of antibiotics. Further information is provided in response to Reviewer comment (25).

(23)- Broader Contextualization: While the focus on SEZ is appropriate, drawing connections to other pathogens and conditions could enhance the manuscript's relevance to a wider audience.

We have added some additional data from *Streptococcus pneumoniae* and *Streptococcus agalactiae* (Group B Streptococcus, GBS) (**Additional data #11**). Moribund mice infected by these two pathogens did not show higher CFU burdens in the brain compared to other tissues, suggesting that this phenotype may not be universal among streptococci.

Additional Data #11 The brain bacterial burden of another two meningeal pathogens, *Streptococcus agalactiae* A909 and *Streptococcus pneumoniae* D39 (New Fig. s1e, s1f)

a, Survival curve of C57BL/6J mice intravenously injected with *S. agalactiae* A909 (10^7 and 10^8 CFU) or *S. pneumoniae* D39 (10^6 or 10^7 CFU). **b**, Organ CFU counts in mice i.v. injected with 5×10^8 CFU A909 (moribund between 24 and 72 hpi, n=3) or 5×10^7 CFU of D39 (moribund between 24 and 120 hpi, n=3). Survival curves were analyzed using the simple survival analysis following a log-rank test vs. WT. CFU burden of each organ were analyzed using one-way ANOVA followed by Dunnett's multiple comparisons test.

(24) The manuscript presents novel and valuable insights into the mechanisms of SEZ proliferation in the CNS, with potential implications for therapeutic development. However, it would benefit from a deeper exploration of the therapeutic implications, and a clearer focus in the discussion. Addressing these points would strengthen the manuscript and make it more suitable for publication in Nature.

Thank you for your encouraging comments. Please see the response to comment (25)

Reviewer #3 (Remarks to the Author): Author responses are in blue.

This article by Yuan et al. presents a study that investigates the unique ability of zoonotic *Streptococci equi* (SEZ) to grow to high abundance in cerebrospinal fluid (CSF) and attributes this ability to the activity of the mannose phosphotransferase system (PTS_{man}) system in SEZ, specifically at low glucose concentrations. Additionally, they attempt to make a connection between the activity of the PTS_{man} system and the lack of stringent response activity.

To understand the kinetics of colonization of the brain and further growth, the authors use a mouse model of IV infection of uniquely barcoded SEZ. This revealed that the founding population in the brain is very small (~10 cells) and distinct from the population involved in systemic infection and the blood.

Using a SEZ transposon library, a Tnseq screen identified bacterial genes important for growth in the brain, including those involved in proline synthesis and the PTS mannose system. The authors go on to investigate how the SEZ promoter for *manXYZ* (P_{man}), unlike other Streptococcal *manXYZ* promoters, is still active even at low glucose (2g/L) concentrations using a promoter fused to beta-galactosidase. They show that the P_{man} is specifically responsible for this by testing in a GBS species background as well. They show that with a GBS P_{man}, SEZ is not able to grow to a higher optical density like WT SEZ in low glucose conditions and, in fact, phenocopies the *manY* mutant.

Perhaps the most unclear conclusion in the paper is that the PTS_{man} system in SEZ prevents the stringent response, resulting in an increased abundance in the brain and CSF in low glucose conditions. The authors show that the activity of the SEZ PTS mannose system at low glucose concentrations and the stringent response are associated by identifying the differentially expressed genes in SEZ Δ *manY* in comparison to WT SEZ and identify these genes as also being regulated by the stringent response.

Thank you for your comments and summary. We have conducted additional experiments to connect PTS_{man} to the stringent response, which are outlined below.

****Main critiques:****

(25) - Of the genes identified, the authors focused on *manY* as the *manY* mutant resulted in no change in mortality but a significant decrease in bacterial abundance in the brain. Although this gene may describe why SEZ can grow to such high abundance, the significance of this insight to potential meningitis remains weak, as there is no link between disease severity, complications, or outcome shown from increased bacterial abundance. It is interesting that the $\Delta 6210$ mutant was not chosen for further study as it is also related to the PTS_{man} system and in an i.v. inoculation resulted in a defect in growth in the brain only and also had decreased mortality.

Thank you for these insightful comments.

To address the reviewer's concern about the significance of *manY* during CNS infection, we have conducted additional experiments. First, we performed H&E staining on brains of mice infected with WT and $\Delta manY$ (**Additional Data #4**). These experiments demonstrate that brains infected with WT SEZ had significantly more severe pathology than those infected with $\Delta manY$, including neutrophilic infiltration and hemorrhage in the meninges, choroid plexus, and olfactory bulb. These results suggested that reduced bacterial burden contributed to alleviating brain injury in bacterial meningitis, despite the fact that overall survival is not altered. Pathology may therefore result from PTS_{man}-dependent SEZ growth in the brain.

Additional Data #4 H&E staining of brains infected with WT or $\Delta manY$ SEZ (New Fig. 3e).

H&E staining of brain sections from moribund mice intravenously infected with WT or $\Delta manY$ strains. The left panel shows low-magnification views of brain sections, with black squares indicating regions selected for detailed analysis across WT, $\Delta manY$, and vehicle (blank) inoculations. The magnified views in subsequent columns correspond to the meninges, ventricle, and olfactory bulb regions, arranged sequentially for consistent comparison. Red arrows indicate areas of hemorrhage. Blue arrows highlight neutrophil infiltration. WT exhibits obvious pathological features such as neutrophil infiltration and hemorrhage, while $\Delta manY$ shows mild pathology.

To further underscore the significance of our findings, we examined whether the influence of antibiotic therapy in the context of *manY*-dependent infection. In these experiments, we infected mice with WT SEZ or the $\Delta manY$ mutant and administered penicillin. Despite no difference in the *in vitro* susceptibility of these strains, penicillin treatment markedly increased the survival of mice infected with the $\Delta manY$ (55.5% survival by 6 days post infection). In contrast, although penicillin also prolonged survival in mice infected with WT SEZ, 100% of mice died by 4.5 days post infection. **(Additional Data #5)**. Therefore, inactivating PTS_{man}-dependent growth has a marked effect in enhancing the therapeutic benefits of antibiotic therapy.

Additional Data #5 Survival curve of mice treated with SEZ or $\Delta manY$ following penicillin treatment (a, New Fig. 3f)

Survival curve of C57BL/6J mice after intravenous administration of 200 μg penicillin every 12 hours, beginning 24 hpi. Mice were challenged i.v. with 5×10^6 CFU WT or $\Delta manY$, and subsequently received penicillin or vehicle (PBS) treatment starting at 24 hpi (*p*-value were analyzed with the simple survival analysis following a log-rank test).

We chose to focus on *manY*, not 6210, since the CFU burden of the $\Delta 6210$ mutant in the brains of moribund mice was highly variable, ranging from approximately 10^5 to 10^9 CFU/organ (Figure S6d). The reduction in average CFU in both the $\Delta manY$ and $\Delta 6210$ strains were highly specific to the brain. However, the $\Delta manY$ strain was markedly more consistent than the $\Delta 6210$ strain. Furthermore, examination of the distributions of transposon insertions within the PTS operon containing the 6210 gene suggests that this PTS system as a whole is not essential for proliferation in the brain, since the other components (e.g., EIIA) as well as the promoter region contained numerous transposon insertions (**Additional Data #6**). In contrast to 6210, insertions were depleted from the entire PTS_{man} operon. Therefore, we selected the *manY* gene within the PTS_{man} operon for further investigation.

Additional Data #6 The distributions of transposon insertions within the PTS operon containing the 6210 gene (New Fig. s3c)

Comparison of the distributions of transposon insertions in another mannose PTS operon (RS06200~RS06215) and the upstream and downstream gene in the input library, as well as in three rounds of the output samples. The RS06210 gene is marked in red. The yellow background denotes the coding sequence (CDS) and intergenic regions. TA sites indicate potential transposon insertion site.

(26) - Validation of Tn-seq screen should have been performed in the same way as the screen with the stereotaxic model to confirm that the mutants had a defect in growth in the brain in case the mutant SEZ could not pass through the blood-brain barrier in the same way, which may also result in decreased abundance in the brain.

We agree that ideally, the same method (stereotaxic i.c. injection) used in the Tn-seq screen should also be applied for validation. However, we primarily chose to use i.c. injection for the TnSeq screen to circumvent the bottleneck into the brain, which would prohibit genome-scale analyses. Due to the practical challenges of using the stereotaxic model for large numbers of mice, we opted for tail vein injection to assess whether select bacterial mutants exhibit growth defects in the brain. Intravenous injection is also more analogous to natural infection, so we used it for initial validation. Importantly, we did carry out stereotaxic injections to confirm the phenotypes of WT, $\Delta manY$, and $C\Delta manY$ strains in Formerly Figure 6G and New Figure 3d.

(27) - The GBS P_{man} experiment data seem overinterpreted. The in vitro growth differences seem very slight with the optical density only increasing by a few hundredths with no statistics applied. The difference in abundance in vivo was more convincing, with i.v. injected mice having lower SEZ abundance in the brain with the GBS P_{man} promoter compared to WT. However, these differences were apparently not significant with an alpha level of 0.05.

We have performed additional experiments assessing growth differences by counting CFU at the stationary phase. The GBS P_{manX} promoter replacement strain (SEZ::P_{manX(A909)}) showed a 50% reduction in CFU compared to the WT and $C\Delta manY$ strains at 12 hours in CDM with 0.2 g/L glucose ($p < 0.001$), while displaying similar CFU counts to the $\Delta manY$ strain (**Additional Data #7, a**). Statistical analyses have been updated

We have also added 2 more mice challenged with SEZ::P_{manX(A909)}. The CFU in the brain and CSF of SEZ::P_{manX(A909)}-challenged mice was significantly lower than WT (two-way ANOVA, $p = 0.0034$ for brain and $p = 0.012$ for CSF, **Additional Data #7, b**).

Additional Data #7 Growth assessment by CFU counting at stationary phase of SEZ WT and its derivatives (a, New Fig. 5f); the CFU burden of moribund mice after challenge with WT SEZ or SEZ::P_{manX(A909)} (b, Formerly Fig. 5e, New Fig. 5g)

a, The CFU of WT, $\Delta manY$, $C\Delta manY$ and SEZ::P_{manX(A909)} (n = 3) at stationary phase (cultured 12 h) in CDM with 0.2, 1, 2 and 10 g/L glucose respectively (one-way ANOVA followed by Dunnett's multiple comparisons test). **b**, C57BL/6J mice were i.v. injected with SEZ (n = 4) or SEZ::P_{manX(A909)} (n = 4, moribund between 31 and 37 hpi), CFU was counted from organs of moribund mice. (two-way ANOVA followed by Šídák's multiple comparisons test).

(28) - The stringent response conclusions are not well-supported. There are many regulators of metabolism genes. Additional evidence for this link to the stringent response included a suboptimal qualitative detection of ppGpp via thin-layer chromatography with no complement of the *manY* mutant and quantification of *relA* transcript in SEZ $\Delta manY$. *relA* transcript may increase in *manY* mutants, but this is not specifically indicative of *relA* activity. Fig 6e might suggest that the stringent response is limiting bacterial abundance, but how that is related to the PTS_{man} system is not defined. Mechanistically, this assertion seems underdeveloped, as there is minimal evidence to suggest that the activity of P_{man} decreases the stringent response. If it does, the mechanism is very unclear.

Thank you for these constructive comments. We agree the transcription level of *relA* may not support a role for the stringent response. We have performed several additional experiments to clarify the connection between PTS_{man} and the stringent response.

First, we have added the complemented strain in the thin-layer chromatography detection of ppGpp (**Additional Data #14**). The complementation restores the concentration of ppGpp to those observed in WT.

Additional Data #14 New TLC detection of ppGpp in WT, $\Delta manY$ and $C\Delta manY$ cultured in CDM with different glucose concentrations (New Fig. 6c).

Intracellular levels of (p)ppGpp were detected by TLC after nucleotide extraction of SEZ WT, $\Delta manY$ and $C\Delta manY$ strains. Bacteria were grown in CDM with low (1 g/L) or high (10 g/L) concentrations of glucose.

Second, we have performed an orthogonal validation of ppGpp quantification using a reporter system, where increased fluorescence reflects increased levels of ppGpp. We observed that the $\Delta manY$ and SEZ::P_{manX(A909)} had higher ppGpp levels than WT or $C\Delta manY$

Additional Data #15 Quantification of in vivo ppGpp concentration with the S2 sensor plasmid (New Fig. 6d).

Fluorescence imaging of (p)ppGpp biosynthesis in SEZ, $\Delta manY$, $C\Delta manY$ and SEZ::P_{manX(A909)} strains under 1 or 10 g/L glucose. All strains were grown to mid-exponential phase in CDM containing 1 or 10 g/L

glucose, and fluorescence images were captured with a 100 ms exposure time at 30% shift. GFP signal ranged from 200 to 500 and Bright field ranged from 800 to 4000. Identical imaging conditions and settings were applied across all figures, as detailed in the supplementary materials.

Third, we have assessed the role of several key intermediate proteins that link PTS_{man} to the stringent response. The PTS_{man} component is part of the EII complex within the PTS system and is functionally associated with the EI complex, which includes HPr and the EI enzyme. HPr phosphorylation has been reported to reduce Crp expression in *Listeria monocytogenes* (Mertins et al., 2007. *J Bacteriol*). The Crp protein can bind to cAMP, forming the cAMP-Crp complex, which acts as a transcriptional regulator. To provide more mechanistic insights connecting PTS_{man} and the stringent response in SEZ under low glucose conditions, we have performed additional experiments focused on Crp, *relA*, and HPr. We found that deletion of the *crp* significantly reduced *relA* transcription in the $\Delta manY$ mutant under 1 g/L glucose conditions. EMSA results showed that Crp binds to the *relA* promoter in the presence of cAMP (**Additional Data #9**), linking *relA* regulation to the cAMP-Crp complex. We further investigated the relationship between HPr phosphorylation and Crp expression in SEZ. In 1 g/L glucose CDM, both $\Delta manY$ and $SEZ::P_{manX(A909)}$ displayed low phosphorylated HPr and high Crp expression levels. In contrast, the WT and $C\Delta manY$ strains showed the opposite phenotypes. In 10 g/L glucose CDM, all four strains exhibited high phosphorylated HPr and low Crp expression (**Additional Data #10**). These data suggest an inverse correlation between Crp levels and HPr phosphorylation, indicating a relationship between Crp and PTS_{man} and mechanistically connecting PTS_{man} to the stringent response (**Additional Data #8**)

Additional Data #8 Graphical schematic of how PTS_{man} influences the stringent response in low glucose (New Fig. 6i)

Schematic model illustrating SEZ inhibition of the stringent response via the PTS_{man} system. Under low-glucose conditions, SEZ actively transports glucose through PTS_{man}, leading to phosphorylation of HPr, a component of the PTS system. The high ratio of phosphorylated HPr suppresses Crp expression, thereby inhibiting the activation of *relA* transcription by the cAMP complex. Reduced *relA* transcription limits ppGpp accumulation, effectively preventing the onset of the stringent response. However, $\Delta manY$ was unable to transport sufficient glucose to maintain a high ratio of P-HPr, resulted in Crp expression and then active the transcription of *relA*, ultimately resulting in ppGpp accumulation. Schematic is created with BioRender.com.

Additional Data #9 Detection of Crp binding to the promoter of *relA* with or without cAMP (a, New Fig. 6f); qPCR analysis of *relA* expression levels in SEZ WT and its derivatives cultured in CDM with 1 g/L glucose until log phase (b, New Fig. 6g); TLC detection of ppGpp in WT, Δcrp , $\Delta manY$ and $\Delta manY\Delta crp$ cultured in CDM with 1 g/L glucose (c, New Fig. 6h).

a, EMSA analysis assessing the binding of the *relA* promoter to Crp or the Crp:cAMP complex. **b**, Relative transcript levels of *relA* in WT, Δcrp , $\Delta manY$ and $\Delta manY\Delta crp$ strains, as detected by qRT-PCR. The data were normalized to the *recA* transcript (n = 3, one-way ANOVA followed by Dunnett's multiple comparisons test). All strains were collected at mid-exponential phase. **c**, Intracellular levels of (p)ppGpp were detected by TLC after nucleotide extraction of SEZ WT, Δcrp , $\Delta manY$, $\Delta manY\Delta crp$ strains. Bacteria were grown in CDM with 1 g/L glucose.

Additional Data #10 The HPr phosphorylation status (top) and the intracellular accumulation of Crp (bottom) in SEZ and its derivatives following incubation in CDM with 1 g/L or 10 g/L or 10 g/L glucose (New Fig. 6e)

Western blot analysis of HPr phosphorylation ratio and Crp expression levels in WT, $\Delta manY$, $C\Delta manY$ and SEZ::P_{manX(A909)} strains grown in CDM with 1 g/L or 10 g/L glucose. GroL was used as a loading control. All samples were collected at mid-exponential phase.

****Other specific critiques:****

- ****Figures:****

(29) - Fig 1: Would benefit from CFU data from other meningeal pathogens to really prove that this observation is remarkable.

Thank you for your suggestion. We assessed bacterial burden in the brain of two other meningeal pathogens, *Streptococcus agalactiae* A909 and *Streptococcus pneumoniae* D39. See response to comment (23) **(Additional Data #11)**

(30) - Fig 3: Unclear why results were verified in i.v. injection model and not model used in Tnseq screen.

Thank you for your suggestion. See the response to comment (26).

(31) - Fig 5: Would have appreciated an alignment of Streptococcal promoters. What is different about the SEZ promoter specifically?

Thank you for your suggestion. We have updated the alignment of PTS_{man} promoters to better highlight the differences among the promoters of *Streptococci* (**Additional Data #12 a, New Fig. s6c**). Based on the promoter activity analysis in **New Figure 5b**, the essential region for transcriptional activity spans from -152 to -98. Since GAS had the most similar promoter region (P_{manX}) to SEZ (differing by only two nucleic acids, **New Fig. S6c**), we constructed single site mutants in the -35 and -10 regions of SEZ P_{manX} to match the sequences in GAS P_{manX} . All mutants reduced SEZ P_{manX} activity (**Additional Data #12 b, New Fig. 5c**), suggesting that the high transcriptional activity of SEZ P_{manX} is species specific and is dependent on the two residues that distinguish SEZ and GAS P_{manX} sequences.

Additional Data #12 Sequence alignment of the promoter regions of different streptococci, from 10 bp upstream of the -35 element to 10 bp downstream of -10 region (a, New Fig. s6c); Promoter activities of P_{manX} mutants (b, New Fig. 5c).

a, Nucleotide sequence homology of PTS_{man} promoter from 10 bp upstream of -35 region to 10 bp downstream of -10 region (-152 to -104 bp) across *Streptococcus*, with strains shown on the left. The -35 and -10 regions are marked with red squares. The nucleotide homology is indicated by different shades of blue, with a threshold set at 50%. A “-” indicated a gap base and “.” Indicated the consensus base exceeding the 50% threshold. **b**, The nucleotide sequence is the alignment of -152 to -104 region in P_{manX} of GAS and SEZ, there are only two nucleotides mutant. Site-directed mutants were induced into the SEZ P_{manX} promoter according to the same region of GAS and cloned into the pTCV-lacZ vector to identify the essential of SEZ P_{manX} conservation to its high transcriptional activity.

(32) - Fig 5d: Very hard to see graphs inside other graphs. I understand the authors were trying to keep the axes the same, but it makes it harder to understand the conclusion.

We have updated former Fig. 5d to New Fig. 5e (**Additional Data #13**). Additionally, we added the CFU count data at stationary phase to make the differences more apparent (**Additional Data #7, New Fig. 5f**).

Additional Data #13 Growth curve (n = 4) of WT, $\Delta manY$, $C\Delta manY$ and SEZ:: $P_{manX(A909)}$ in CDM with 0.2, 1, 2 and 10 g/L glucose. (New Fig. 5e).

Growth curve (n = 4) of WT, $\Delta manY$, $C\Delta manY$ and SEZ:: $P_{manX(A909)}$ in CDM with 0.2, 1, 2 and 10 g/L glucose. The *p*-values were analyzed using a one-way ANOVA with Dunnett's multiple comparisons test applied to growth curves fitted by the Gompertz nonlinear regression model.

Additional Data #7 Growth assessment by CFU counting at stationary phase of SEZ WT and its derivatives (a, New Fig. 5f)

a, The CFU of WT, $\Delta manY$, $C\Delta manY$ and SEZ:: $P_{manX(A909)}$ (n = 3) at stationary phase (cultured 12 h) in CDM with 0.2, 1, 2 and 10 g/L glucose respectively (one-way ANOVA followed by Dunnett's multiple comparisons test).

(33) - Fig 6a: Title is misleading. Your Venn diagram is not Δ manY vs. SEZ.

Thank you for your meticulous review. We have revised the title to improve clarity.

(34) - Fig 6b: This figure is very confusing and does not support the association between the stringent response and PTS mannose system well.

Thank you for your suggestion. Please see the response to comment (28).

(35) - Fig 6f: TLC very blotchy, at least for 0.2g/L. No complement included.

We have updated these images (**Additional Data 14**)

- **Text:**

(36) - Line 37: change “with bacterial invasion” to “when bacteria invade.”

We have revised this sentence from “with bacterial invasion” to “when bacteria invade.” (Lines 37)

(37) - Line 42: italicize “Streptococcus.”

We have revised Streptococcus to italicized format and checked the entire manuscript to ensure all instances of Streptococcus are consistently italicized.

(38) - Line 44: change “whereas” to “and” – it is odd to contrast two pathogens you are trying to persuade your audience are both significant.

We have made the revision (Lines 44)

(39) - Line 46: change “is” to “has.”

We have revised this word from “is” to “has” (Lines 46)

(40) - Line 46-47: take out “emerging pandemic” or “outbreak in death.”

We have made the revision (Lines 46-47)

(41) - Line 62: italicize “Streptococci.”

We have revised *Streptococci* to italicized format and checked the entire manuscript.

(42) - Line 351: change “promoted by a strong promoter.”

We have made the revision (Lines 397)

(44) - Line 368: change “entrance into” to “enter.”

We have revised this word from “entrance” to “enter” (Lines 414)

(45) The final claim of significance for your findings is not well supported: If there were therapeutics developed, your findings suggest that lethal meningitis would still develop (Fig 3c tail vein injection of SEZ Δ manY), which is not “mitigating the deleterious consequences of CNS infection.”

Thank you for the suggestion. We have added some data to support the therapeutic implications. Please see the response to comment (25).

****General writing critiques:****

(46) Make sure Streptococcus and Streptococci are capitalized and italicized.

Ensure gene names are italicized (e.g., manY), including Δ manY.

I apologize for these writing errors. We will revise accordingly.

Thank you very much for organizing the second-round review of our manuscript, “Constitutive Glucose Import in Zoonotic Streptococci Enables Proliferation in Cerebrospinal Fluid”, and for giving us the opportunity to revise it. We have now addressed the additional comments provided by the original reviewers, as well as the new critiques raised by the two newly invited reviewers. In response, we have incorporated nine additional experiments and revised the manuscript accordingly. We believe these efforts have significantly strengthened the work. Thank you again for your time and thoughtful consideration.

In your decision letter, you mentioned five major concerns:

- 1) In particular, you will see that Referee #1 notes that it remains unclear whether within-CNS virulence stems from growth within the CNS or more translocation across the BBB. We would ask you to address this point whether with new or existing data.

Thank you for the opportunity to clarify this point. Our new and existing data support the idea that within-CNS virulence stems from growth within the CNS, based on two key pieces of evidence.

Our existing STAMP data reveals the presence of clones that dominate populations in the CNS that are present at very low abundance or absent from other sites (**Figure 2e, Extended Data 2f and 2g**). These observations support the notion that only a subset of bacteria in the brain originates from the bloodstream, with the majority proliferating locally within the brain. These data imply that high bacterial loads in the brain likely arise from in situ replication rather than ongoing translocation, and that this robust intracerebral proliferation is at least partially responsible for the observed virulence.

We provide new data to show that mice with worse clinical outcomes have high bacterial loads in the brain but very little or absent bacteria in the blood. To demonstrate that CNS replication is sufficient for SEZ virulence, we provide **Additional Data #1 (new Figure 3f and 3g)** that demonstrates that when animals are infected with WT SEZ and then given penicillin (does not penetrate the BBB) at 12-hour intervals beginning 24 hours after infection, they uniformly succumbed to infection and exhibited high CFU in the brain despite the absence of detectable bacteria in the blood. In contrast, a subset of $\Delta manY$ -infected mice survived under the same regimen of penicillin treatment. Furthermore, by comparing brain CFU counts between surviving and moribund $\Delta manY$ -infected mice that had undetectable bacterial loads in the blood (0 CFU), we found that moribund mice consistently exhibited significantly higher

bacterial burdens in the brain than those that survived. These findings suggest that elevated brain CFU is more closely associated with virulence, particularly because in the absence of detectable bacteremia, SEZ translocation across the BBB is unlikely.

2) Referee #4 notes that it remains unclear whether pppGpp underlies the growth defect in the absence of *manY* and that experiments with pppGpp null mutant strains are required to support this conclusion, alongside some other analyses to further support this link.

Thank you for this suggestion. We have generated a series of mutants to assess the link between (p)ppGpp included this data in Figure 6b and 6c. Note that we have also corrected a naming inconsistency regarding the gene nomenclature for (p)ppGpp synthetase enzymes in *Streptococcus*. In many Gammaproteobacteria species, the term *relA* is commonly used to refer to monofunctional (p)ppGpp synthetases. However, in Firmicutes, *relA* typically denotes bifunctional synthetase/hydrolase enzymes (PMID: 20508246). Therefore, *relA* should not have been used to describe streptococcal monofunctional (p)ppGpp synthetases in this study. We have corrected this nomenclature error in the revised manuscript: the primer previously labeled as “*relA*” has been renamed *relQ*, and “RSH” has been renamed *relA*. Please refer to our response to comment (24) for additional details.

We constructed the (p)ppGpp⁰ strain ($\Delta relA\Delta relQ$) by deleting the *relA* and *relQ* genes in the SEZ WT background (**Additional Data #4**). Subsequently, the RNA-seq analysis with SEZ WT, (p)ppGpp⁰ ($\Delta relA\Delta relQ$), $\Delta manY$ strains was performed. The $\Delta relA\Delta relQ$ vs. WT group was used as a reference to define stringent response associated gene expression, which are truly regulated dependent on the (p)ppGpp. Of the 773 growth related DEGs under low-glucose conditions in a PTS_{man} dependent manner, 432 were classified as (p)ppGpp-dependent (**New Extended Data Fig. 7c**). Expression analysis of these 432 genes showed predominantly opposite regulatory trends in $\Delta manY$ and $\Delta relA\Delta relQ$, consistent with their contrasting (p)ppGpp levels, elevated in $\Delta manY$ and depleted in $\Delta relA\Delta relQ$ (**New Extended Data Fig. 7e**; PMID: 30916318, 32345719). The updated RNA-seq data and analysis pipeline provide support for the notion that *manY* deletion-induced DEGs under low-glucose conditions are functionally linked to the stringent response.

As the $\Delta relA\Delta relQ$ has growth defective and it is not an ideal model to measure the bacterial growth, we constructed $\Delta relQ$, $\Delta manY\Delta relQ$ (low (p)ppGpp), and $\Delta manYC\Delta relQ$ (restore (p)ppGpp in $\Delta manY\Delta relQ$ background) strains (**Additional Data #4**) to reveal whether (p)ppGpp underlies the growth defect in the absence of *manY*. We found that deletion of the *relQ* gene significantly reduced the intracellular (p)ppGpp levels and rescued the growth defect caused by the absence of *manY* under

low-glucose conditions. Conversely, reintroducing *relQ* into the $\Delta manY\Delta relQ$ background restored (p)ppGpp production and led to the reappearance of the growth defect in the $\Delta manYC\Delta relQ$ strain. These results suggest that the intracellular (p)ppGpp level is closely associated with the growth of SEZ under low-glucose conditions (**Additional Data #7**).

3. Reviewer #2 also notes that the CFU data remain very variable which may have implications for the robustness of the data. We feel that these are critical points which must be addressed in a revised manuscript.

Thank you for the opportunity to clarify this point. As stated in the Materials and Methods section, all mice used in this study were female C57BL/6J mice aged between 6 and 8 weeks. To further minimize experimental variation, we specifically requested that animals be age-matched (6 or 8 weeks old, respectively) and obtained from the same litter, thereby reducing potential differences related to husbandry conditions and developmental stage. Within each group, CFU values varied by approximately 100-fold, whereas across both age groups, the range between the highest and lowest values approached 1,000-fold (**Additional Data #2**). The substantial variability observed in CSF CFU may be partly attributable to the use of 6–8-week-old mice. We believe this level of variation falls within the expected range for infection studies conducted in mice of this age group. Notably, comparable levels of variability in CFU measurements have also been reported in previously published studies (e.g., PMID: 30011336; PMID: 32943639; PMID: 40089471), suggesting that such variation is not uncommon in similar experimental settings.

4. The rest of the referees' reports are clear and the remaining issues should be straightforward to address.

We are grateful for these constructive comments and have provided detailed responses to each point.

5. We note that Referee #2 was concerned that analysis of the immune component of the response had been overlooked. Though we agree that this would develop and expand the manuscript and our understanding of the host response to streptococcal CNS infection, it would not be an essential component and its lack would not preclude us returning the revised manuscript to reviewers.

Thank you for your understanding. We recognize that immune-related mechanisms may be more complex and challenging to fully elucidate within the scope of the current study. Therefore, we did not extensively investigate host immune responses in this work. Nonetheless, elucidating how SEZ adapts to the immune microenvironment of the brain remains an intriguing direction for future research.

New data to address referees' comments:

Additional Data #1: Survival curves and CFU of WT or $\Delta manY$ -infected mice administered penicillin 24 hours post infecton. (A, New Fig.3f; B, New Fig.3g)

A, Survival curve of C57BL/6J mice after intravenous administration of 200 μ g penicillin every 12 hours. Mice were challenged i.v. with 5×10^6 CFU WT or $\Delta manY$, and subsequently given penicillin or vehicle (PBS) treatment starting at 24 hpi (Simple survival analysis following a log-rank test). **B**, CFU burden in the organs and blood of moribund and surviving mice post infection and penicillin treatment.

Additional data #2: CSF CFU counts were obtained from age-matched littermate mice to ensure consistency.

The CSF CFU of mice at 6-week-old or 8-week-old

Additional data #3: Quantification of histopathological brain sections (Table, Supplementary Table S10; Figure, New Fig. 3e).

	Meninges		Choroid Plexus		Cerebral Parenchyma		Cerebellar Parenchyma		Total score
	Hemorrhage	Inflammatory cell infiltration	Hemorrhage	Vacuolar degeneration	Inflammatory cell infiltration	Hemorrhage	Inflammatory cell infiltration	Hemorrhage	
Uninfected-1	0	0	0	0	0	0	0	0	0
Uninfected-2	0	0	0	0	0	0	0	0	0
WT-1	3	0	1	0	1	1	0	1	7
WT-2	2	2	1	1	1	1	0	1	9
WT-3	1	1	3	0	2	1	0	1	9
$\Delta manY$ -1	0	0	1	1	0	0	0	0	2
$\Delta manY$ -2	0	0	0	0	1	0	0	0	1
$\Delta manY$ -3	0	0	0	0	0	1	0	0	1

The table presents the histopathological scores of the brain sections of mice. Pathological features were graded based on severity, with scores assigned as follows: 3 for severe, 2 for moderate, 1 for mild, and 0 for no detectable lesions.

The figure represents the total histopathological scores of brain sections from indicated mice. Statistical significance was analyzed with one-way ANOVA followed by Dunnett's multiple comparisons test.

Additional data #4: Construction of the (p)ppGpp⁰ (Δ relA Δ relQ), Δ relQ, Δ manY Δ relQ and Δ manYC Δ relQ strains of SEZ and detection of their ratio of (p)ppGpp with TLC. (A, New Extended Data Fig.7b; B, New Fig.6c)

Intracellular levels of (p)ppGpp were detected by TLC after nucleotide extraction. **A**, SEZ WT, Δ relA Δ relQ strains were grown in CDM with low (1 g/L) or high (10 g/L) concentrations of glucose. **B**, Strains were grown in CDM with 1 g/L glucose.

Additional data #5: RNA-seq analysis of SEZ WT, (p)ppGpp⁰ (Δ relA Δ relQ), and Δ manY strains. (New Extended Data Fig.7 exclude b and d)

A, RNA-seq analysis shows the number of DEGs between $\Delta manY$ and WT cultured in CDM with 1 and 10 g/L glucose at mid-exponential phase. The average $\log_2 Fc = 1$ was used as the threshold of DEGs. **B**, The Venn diagram illustrates the intersection of DEGs between $\Delta manY$ vs. SEZ cultured in 1 g/L glucose CDM, non-DEGs of $\Delta manY$ vs. SEZ cultured in 10 g/L glucose CDM, and DEGs of $\Delta relA \Delta relQ$ vs. SEZ. Growth related genes are marked in the purple lens-shaped region (intersection of $\Delta manY$ vs. SEZ cultured in 1 g/L glucose CDM and non-DEGs of $\Delta manY$ vs. SEZ cultured in 10 g/L glucose CDM), while the (p)ppGpp dependent genes are shown in the green circle ($\Delta relA \Delta relQ$ vs. SEZ DEGs). The central overlap among all three sets (432 genes) indicates candidates involved in both growth regulation and (p)ppGpp. **C**, The heatmap displays the $\log_2 Fc$ of the intersection genes in panel c in $\Delta manY$ vs. WT (influenced by *manY* in low glucose) and $\Delta relA \Delta relQ$ vs. WT (influenced by (p)ppGpp) groups. The combined group consists of four distinct clusters, with the different colored squares on the right side of the heatmap representing each cluster. **D**, The boxplots indicate normalized Transcripts Per Million (TPM) of *relA* and *relQ* in SEZ and $\Delta manY$ cultured in CDM with 1 g/L (SEZ1 and $\Delta manY1$) or 10 g/L (SEZ10 and $\Delta manY10$) glucose.

Additional data #6: The (p)ppGpp levels were measured by TLC, and HPr and Crp were detected by western blot. Band intensities were quantified using ImageJ. (A, New Fig.6d; B, New Fig.6c; C, New Fig.6h; D, New Fig.6e. All column plots of gray intensity analysis are displayed in New Extended Data Fig.8)

A-C. Intracellular levels of (p)ppGpp detected by TLC plates. The relative intensity of (p)ppGpp is calculated as $(ppGpp + pppGpp) / (GTP + ppGpp + pppGpp)$. The bubble plots below each panel represent the gray intensity corresponding to the results. **D.** Gray intensity analysis of Crp expression levels and the percentage of P-HPr. Crp expression level is calculated as the ratio of Crp to GroEL. The percentage of P-HPr is calculated as the ratio of P-HPr to total HPr. The bubble plots below each panel represent the gray intensity corresponding to the results.

The column plots on the right of each panel present the replicated gray intensity results.

Statistical significance of A and D was analyzed with two-way ANOVA followed by Dunnett's multiple comparisons test. Statistical significance of B and C was analyzed with one-way ANOVA followed by Dunnett's multiple comparisons test.

Additional data #7: Growth curve of WT, $\Delta manY$, $C\Delta manY$, $\Delta relA\Delta relQ$ ((p)ppGpp⁰) $\Delta relQ$, $\Delta manY\Delta relQ$, $\Delta manYC\Delta relQ$ (complement strain of $\Delta manY\Delta relQ$ with pSET2-*relQ* overexpression plasmid) and *relQ*⁺ (WT strain with pSET2-*relQ* overexpression plasmid). (B, New Fig.6b)

A, the growth curve of WT and (p)ppGpp⁰ strain. (p)ppGpp⁰ strain exhibits growth defects compared to WT SEZ. **B**, the CFU of WT, $\Delta manY$, $C\Delta manY$, $\Delta relQ$, $\Delta manY\Delta relQ$, $\Delta manYC\Delta relQ$, and *relQ*⁺ strains in CDM with 1 g/L glucose at 0, 4 and 12 h; **C**, the growth curve of WT, $\Delta manY$, $C\Delta manY$, $\Delta relQ$, $\Delta manY\Delta relQ$, $\Delta manYC\Delta relQ$, and *relQ*⁺ strains in CDM with 1 g/L glucose. Overexpression of the *relQ* gene in the WT SEZ inhibits bacterial growth, while its deletion has no effect. In the $\Delta manY$ strain, *relQ* deletion restores growth, but reintroducing *relQ* in the $\Delta manYC\Delta relQ$ strain results in growth defects once again.

Additional data #8: The blood CFU counts of the $\Delta proB$ and $\Delta proC$ strains at 24 hours post-infection in the new Extended Data Fig.4b.

C57BL/6J mice were intravenously injected with 5×10^6 CFU of either $\Delta proB$ or $\Delta proC$, and bacterial counts in the blood were measured.

Additional data #9: Intracellular ATP concentration analysis of WT, $\Delta manY$ and $C\Delta manY$ in CDM with 1 g/L glucose.

A, Bacterial growth of WT, $\Delta manY$, and $C\Delta manY$ was measured at different time points by CFU counts, Strains are grown in CDM with 1 g/L glucose. **B**, Intracellular ATP concentration in WT, $\Delta manY$, and $C\Delta manY$ grown in CDM with 1 g/L glucose, corresponding to the time points in panel A, was analyzed using the BacTiter-Glo Microbial Cell Viability Assay Kit, with ATP levels quantified by luminescence intensity.

Reviewers' comments:

Reviewer #1 (Remarks to the Author):

All comments were properly addressed. I acknowledge the excellent work done by the authors.

We sincerely thank the reviewer for their positive comments and constructive feedback. We are grateful for the recognition of our work.

Minor comments:

(1) Fig 5: Authors state that the increased virulence is due to a higher growth rate locally (in the brain). Yet with a bacteremia-derived meningitis model is not really possible to assess whether an increase of bacterial load in the brain is due to higher levels of bacteremia (more bacteria in the blood = more bacteria invading the brain). The only way to assess whether higher disease severity in the brain is due to a higher bacterial growth in the brain, is either to use antibiotics (that do not pass the BBB) for clearing bacteremia, or to use an ic model.

We sincerely appreciate the reviewer's constructive suggestion. Following the recommendation, we administered antibiotics to eliminate bacteremia and assessed both the bacterial burden in the brain and disease severity in this modified infection model.

Additional Data #1 (new Figure 3f and 3g) shows that WT infected mice administered 200 µg of penicillin at 12-hour intervals uniformly succumbed to infection, exhibiting high CFU in the brain despite the absence of detectable bacteria in the blood. In contrast, a subset of $\Delta manY$ -infected mice survived under the same penicillin treatment. By comparing brain CFU counts between surviving and moribund mice that had undetectable bacterial loads in the blood (0 CFU), we found that moribund mice consistently exhibited significantly higher bacterial burdens in the brain than those that survived. These findings suggest that elevated brain CFU is more closely associated with virulence, particularly because in the absence of detectable bacteremia, SEZ translocation across the BBB is unlikely. Furthermore, the current STAMP data support the notion that only a small fraction of bacteria originates from the bloodstream, with the majority proliferating locally within the brain. Together, these data imply that high bacterial loads in the brain likely arise from in situ replication rather than ongoing translocation, and that this robust intracerebral proliferation is at least partially responsible for the observed virulence.

Additional Data #1: Survival curves and CFU of WT or $\Delta manY$ -infected mice administered penicillin 24 hours post infecton. (A, New Fig.3f; B, New Fig.3g)

A, Survival curve of C57BL/6J mice after intravenous administration of 200 μ g penicillin every 12 hours. Mice were challenged i.v. with 5×10^6 CFU WT or $\Delta manY$, and subsequently given penicillin or vehicle (PBS) treatment starting at 24 hpi (Simple survival analysis following a log-rank test). **B**, CFU burden in the organs and blood of moribund and surviving mice post infection and penicillin treatment.

(2) Western blot figures: the western blot data should be accompanied by quantification analysis of the protein bands (based on the loading controls).

Thank you for your suggestion, we have added more replicates and performed gray intensity analysis. (**Additional data #6, Source file #2, New Extended Data Fig.8**)

Additional data #6: The (p)ppGpp levels were measured by TLC, and HPr and Crp were detected by western blot. Band intensities were quantified using ImageJ. (A, New Fig.6d; B, New Fig.6c; C, New Fig.6h; D, New Fig.6e. All column plots of gray intensity analysis are displayed in New Extended Data Fig.8)

A

B

C

D

A-C. Intracellular levels of (p)ppGpp detected by TLC plates. The relative intensity of (p)ppGpp is calculated as $(ppGpp + pppGpp) / (GTP + ppGpp + pppGpp)$. The bubble plots below each panel represent the gray intensity corresponding to the results. **D.** Gray intensity analysis of Crp expression levels and the percentage of P-HPr. Crp expression level is calculated as the ratio of Crp to GroEL. The percentage of P-HPr is calculated as the ratio of P-HPr to total HPr. The bubble plots below each panel represent the gray intensity corresponding to the results.

The column plots on the right of each panel present the replicated gray intensity results. Statistical significance of A and D was analyzed with two-way ANOVA followed by Dunnett's multiple comparisons test. Statistical significance of B and C was analyzed with one-way ANOVA followed by Dunnett's multiple comparisons test.

Reviewer #2 (Remarks to the Author):

While the study presents an interesting hypothesis regarding the ability of *Streptococcus equi* subspecies *zooepidemicus* (SEZ) to proliferate in cerebrospinal fluid (CSF) and explores its glucose acquisition mechanisms, particularly through the phosphotransferase system (PTS_{man}). I have significant concerns regarding the validity of the findings and the robustness of the analysis.

Thank you for your thoughtful feedback. We appreciate your comments about the bacterial burden and immune system analysis, as well as your interest in our study. We also acknowledge your concerns and value your perspective, which we have taken into careful consideration during our revision.

Major Concerns:

(3) Bacterial Load Estimates

The authors responded to my concern by stating that only one mouse had a bacterial load of 10^{10} CFU, while the rest ranged from 10^7 to 10^{10} , with an average of 10^9 CFU. However, this raises concerns about plausibility and statistical consistency. If the average CFU count is 10^9 , most values should cluster around this number. However, with a range spanning three orders of magnitude (10^7 to 10^{10}), this suggests substantial variability. If most values were closer to 10^7 , then the average would likely be lower than 10^9 . Conversely, if most values were near 10^{10} , then 10^7 would be an extreme outlier. A 10^3 -fold variation in bacterial load among a small group of mice is unexpected unless specific factors—such as infection severity or immune response—differ significantly.

Thank you for this comment. As described in the Materials and Methods section, all mice used were female C57BL/6J, aged between 6 and 8 weeks. To evaluate whether age contributed to the variability in CSF CFU, we compared bacterial burdens in 6- and 8-week-old mice.

We used age-matched mice from the same litter, thereby reducing potential variability due to age and husbandry conditions. Notably, we found that mice of different ages exhibited distinct bacterial burdens in the CSF, with 8-week-old mice generally showing higher CFU counts than 6-week-old mice. Within the same age group, a ~100-fold variation was observed, while across both age groups, the CFU range spanned approximately three orders of magnitude ($\sim 10^7$ to 10^{10}), as shown in **Additional data #2**, suggesting that the observed substantial variability in CSF CFU is partly attributable to the age of mice. In our experience, ~1000-fold variations in CSF CFU burdens of 6-8-week-old mice is expected.

Additional data #2: CSF CFU counts were obtained from age-matched littermate mice to ensure consistency.

The CSF CFU of mice at 6-week-old or 8-week-old

(4) Lack of Immune System Analysis

The study does not analyze the immune response to such a high bacterial burden. Given that the brain has a well-regulated immune environment, including microglial activation and cytokine responses, the absence of any discussion or data on immune system involvement weakens the study. The authors should include a thorough immunological analysis to support their claims. The authors responded to my concern by stating that the brain is considered an immune-privileged organ. While the brain has historically been considered an immune-privileged organ with limited immune cell recruitment, this concept has evolved significantly. It is now well established that the brain has an active and dynamic immune environment, with resident immune cells such as microglia playing key roles in immune surveillance and response. Additionally, under pathological conditions or infection, peripheral immune cells—including macrophages and T cells infiltrate the brain, challenging the notion of strict immune privilege. Thus, any study involving bacterial presence in the brain should thoroughly assess immune activation and potential inflammatory responses. Therefore, I do not find this study suitable for publication in its current form.

We sincerely thank the reviewer for the thoughtful concern. We regret that we are unable to fully incorporate this aspect at this time. We are truly grateful for the reviewer's insightful recommendation, which has inspired a valuable line of investigation that we intend to pursue in future work.

Reviewer #3 (Remarks to the Author):

Overall, the authors have done a solid job responding to critiques. Replies to their edits/rebuttals in response to my critiques from the first submission are below.

Thank you for your positive feedback and for taking the time to review our responses in detail. We appreciate your thoughtful evaluation and constructive input throughout the review process.

(5) Histopathology addressed the comment about the pathological significance of the manY knockout given there is no change in mortality. Quantification would be appreciated as it is difficult to determine significance of differences in pathology between manY knockout and WT and it is not clear if images are representative of all individuals in the experiment.

Thank you very much for your comment regarding the histopathological data. In response, we have added additional replicates of brain sections and performed quantitative analysis to facilitate the evaluation of pathological differences among the experimental groups. (Additional data #3)

Additional data #3: Quantification of histopathological brain sections (Table, Supplement Table S10; Figure, New Fig. 3e).

	Meninges		Hemorrhage	Choroid Plexus Vacuolar degeneration		Cerebral Parenchyma		Cerebellar Parenchyma		Total score
	Hemorrhage	Inflammatory cell infiltration		Inflammatory cell infiltration	Hemorrhage	Inflammatory cell infiltration	Hemorrhage	Inflammatory cell infiltration		
Uninfected-1	0	0	0	0	0	0	0	0	0	
Uninfected-2	0	0	0	0	0	0	0	0	0	
WT-1	3	0	1	0	1	1	0	1	7	
WT-2	2	2	1	1	1	1	0	1	9	
WT-3	1	1	3	0	2	1	0	1	9	
$\Delta manY$ -1	0	0	1	1	0	0	0	0	2	
$\Delta manY$ -2	0	0	0	0	1	0	0	0	1	
$\Delta manY$ -3	0	0	0	0	0	1	0	0	1	

The table presents the histopathological scores of the brain sections of mice. Pathological features were graded based on severity, with scores assigned as follows: 3 for severe, 2 for moderate, 1 for mild, and 0 for no detectable lesions.

The figure represents the total histopathological scores of brain sections from indicated mice. Statistical significance was analyzed with one-way ANOVA followed by Dunnett's multiple comparisons test.

(6) If penicillin is the first line antibiotic for Streptococcal bacterial meningitis, then additional data is convincing to show significance of manY to CNS infection.

(7) Practical challenges of stereotaxic route noted. Appreciated the validation of manY knockout through this route at minimum.

(8) Additional data and applied statistics seemed appropriate and addressed concerns that were brought up.

We appreciate that the additional data we provided were able to address most of your concerns in point (6), (7) and (8).

(9) Additional data 14 :

NEED to define what WT::pSET2 and WT::pSET2-relA are. This is not defined explicitly in the text or figure legend just in methods but not clearly.

We have used the term *relQ*⁺ to replace WT::pSET-relA and more details about *relQ*⁺ are in the legend of Fig. 6b. The *relQ*⁺ refers to SEZ containing the *relQ* over-expressing plasmid (pSET2-relQ) under the P_{manX} promoter control.

(10) Lines 368-371 not accurately describing what is seen in the data.

I can agree with the first part of the statement: "Moreover, an increase in ppGpp was observed in Δ manY cultured under low glucose conditions"

I do not agree with the second part of the statement. There are spots that seem like ppGpp in those columns at the location that ppGpp is labeled. Maybe less ppGpp than the low glucose concentrations though. Also there was some fluorescence in those conditions with the reporter too (even if it is less): "whereas ppGpp was hardly detectable in WT or Δ manY when grown in higher glucose concentrations by either TLC (Fig. 6c) or an RNA-based fluorescent sensor (Fig. 6d)."

Thank you for helping us clarify this point. We have revised the statement from "hardly detectable" to "less", and add replicates (**Source file #2**) for gray intensity analysis (**Additional data #6**).

Gray Intensity Ratio of (p)ppGpp=(ppGpp+pppGpp)/(ppGpp+pppGpp+GTP)

Additional data #6: The (p)ppGpp levels were measured by TLC, and HPr and Crp were detected by western blot. Band intensities were quantified using

ImageJ. (A, New Fig.6d; B, New Fig.6c; C, New Fig.6h; D, New Fig.6e. All column plots of gray intensity analysis are displayed in New Extended Data Fig.8)

A-C. Intracellular levels of (p)ppGpp detected by TLC plates. The relative intensity of (p)ppGpp is calculated as $(ppGpp + pppGpp) / (GTP + ppGpp + pppGpp)$. The bubble plots below each panel represent the gray intensity corresponding to the results. **D.** Gray intensity analysis of Crp expression levels and the percentage of P-HPr. Crp expression level is calculated as the ratio of Crp to GroEL. The percentage of P-HPr is calculated as the ratio of P-HPr to total HPr. The bubble plots below each panel represent the gray intensity corresponding to the results.

The column plots on the right of each panel present the replicated gray intensity results. Statistical significance of A and D was analyzed with two-way ANOVA followed by Dunnett's multiple comparisons test. Statistical significance of B and C was analyzed with one-way ANOVA followed by Dunnett's multiple comparisons test.

(11) Additional data 9 and 10 made HPr phosphorylation at low glucose concentrations, crp involvement, and relA connection via cAMP clear. This provided more supporting evidence that the SEZ manY system prevents the stringent response at low glucose concentrations in WT SEZ. Additional data 15 was very much appreciated as it was hard to tell the difference between the ppGpp on the TLC. I think you could see the pppGpp difference on the TLC.

Thank you so much for your positive feedback. We also added more experimental replications to the HPr and crp part, all these western data was quantified by grey intensity.

(12) Schematic (additional data 8) is good.

Thank you for your positive feedback on the schematic.

Other critiques:

(13) Addressed well in 23. Interesting that the same concentration of bacteria was not used between *S. agalactiae*, *S. pneumoniae*, and SEZ. However, the bacterial inoculum for SEZ was lower than *S. agalactiae* and *S. pneumoniae* and still exhibited a significant difference between abundance in the brain only in SEZ and not *S. agalactiae* and *pneumoniae*, so I think it is okay.

(14) Addressed in 26 above.

Thank you for your positive feedback in point (13) and (14).

(15) Initially was confused about the 155 notations. Looks really good but please add an indication of the specific nucleotide you are looking at in panel B in panel A. It took me a while to find that one nucleotide and the "155" does not line up with the negative numbers (I think it is -133). Adding a box to indicate the nucleotide you are looking at in A would be helpful.

The "155" indicates the length of the promoter, not its positional information. We have added an asterisk (*) to align the corresponding nucleotide in panel B (**Extended Data Fig.6c**) with that in panel A (**Fig. 5c**). We apologize for the confusion caused by the earlier explanation.

(16) Addressed concern very well. Different layout of this data was appreciated.

(17) Concerns addressed well.

(18) All text edits addressed.

(19) Addressed with histopathology in 25.

(20) Addressed.

We are grateful to see that our data have addressed most of your comments from point (16) to (20). Thank you very much for your insightful feedback, which has been extremely helpful in improving the overall quality of our manuscript.

Reviewer #4 (Remarks to the Author):

For me the authors show convincing data that ManY transport system is required for glucose uptake in CSF and essential for growth in the brain. Later on the author link this observation to the stringent response. Here are some comments on this part of manuscript:

We are grateful for your thorough and insightful suggestions. These comments have significantly contributed to enhancing the rigor and clarity of our study. We sincerely apologize for the incorrect use of gene nomenclature for (p)ppGpp synthetase enzymes in *Streptococcus*. The primer previously labeled as “*relA*” has been renamed *relQ*, and “RSH” has been renamed *relA*. Please refer to our response to comment (24) for additional details.

(21) It is likely that in the CSF stringent response is activated when the bacteria become limited for glucose or amino acids e.g. in the manY mutant. For *Streptococcus suis*, it was already shown that low glucose somehow results in stringent response activation (<https://www.ncbi.nlm.nih.gov/pubmed/27255540>).

In **Fig. 5a and Extended Data 6c**, we compared the transcriptional activity and promoter sequences of PTS_{man} from various streptococcal species. The PTS_{man} promoter of SEZ exhibits uniquely high transcriptional activity. Even the most homologous PTS_{man} promoter from GAS, which differs by only two nucleotide substitutions, fails to prevent activation of the stringent response. Therefore, it is reasonable to infer that *Streptococcus suis*, which harbors a more divergent PTS_{man} promoter compared to SEZ, exhibits activation of the stringent response under low-glucose conditions.

(22) The data presented do not actually show if and how (p)ppGpp is required for the observed growth defect of the manY mutant (no data using a pppGpp0 strain are provided).

As the $\Delta relA \Delta relQ$ has growth defective and it is not an ideal model to measure the bacterial growth, we constructed $\Delta relQ$, $\Delta manY \Delta relQ$ (low (p)ppGpp), and $\Delta manYC \Delta relQ$ (restore (p)ppGpp in $\Delta manY \Delta relQ$ background) strains (**Additional Data #4**). By analyzing $\Delta relQ$, $\Delta manY \Delta relQ$, and $\Delta manYC \Delta relQ$ strains, we found that *relQ* deletion reduced (p)ppGpp levels and rescued the $\Delta manY$ growth defect under low glucose conditions. Reintroduction of *relQ* restored (p)ppGpp production and the growth defect. These findings link (p)ppGpp levels to SEZ growth in low-glucose environments (**Additional Data #7**). Please find more details in reply to comment (25).

Additional data #4: Construction of the (p)ppGpp⁰ ($\Delta relA\Delta relQ$), $\Delta relQ$, $\Delta manY\Delta relQ$ and $\Delta manYC\Delta relQ$ strains of SEZ and detection of their ratio of (p)ppGpp with TLC. (A, New Extended Data Fig.7b; B, New Fig.6c)

Intracellular levels of (p)ppGpp were detected by TLC after nucleotide extraction. **A**, SEZ WT, $\Delta relA\Delta relQ$ strains were grown in CDM with low (1 g/L) or high (10 g/L) concentrations of glucose. **B**, Strains were grown in CDM with 1 g/L glucose.

Additional data #7: Growth curve of WT, $\Delta manY$, $C\Delta manY$, $\Delta relA\Delta relQ$ ((p)ppGpp⁰) $\Delta relQ$, $\Delta manY\Delta relQ$, $\Delta manYC\Delta relQ$ (complement strain of $\Delta manY\Delta relQ$ with pSET2-*relQ* overexpression plasmid) and *relQ*⁺ (WT strain with pSET2-*relQ* overexpression plasmid). (B, New Fig.6b)

A, the growth curve of WT and (p)ppGpp⁰ strain. (p)ppGpp⁰ strain exhibits growth defects compared to WT SEZ. **B**, the CFU of WT, $\Delta manY$, $C\Delta manY$, $\Delta relQ$, $\Delta manY\Delta relQ$, $\Delta manYC\Delta relQ$, and $relQ^+$ strains in CDM with 1 g/L glucose at 0, 4 and 12 h; **C**, the growth curve of WT, $\Delta manY$, $C\Delta manY$, $\Delta relQ$, $\Delta manY\Delta relQ$, $\Delta manYC\Delta relQ$, and $relQ^+$ strains in CDM with 1 g/L glucose. Overexpression of the *relQ* gene in the WT SEZ inhibits bacterial growth, while its deletion has no effect. In the $\Delta manY$ strain, *relQ* deletion restores growth, but reintroducing *relQ* in the $\Delta manYC\Delta relQ$ strain results in growth defects once again.

Specific comments:

(23) The authors claim that their transcriptome data (Wt versus *manY* under low glucose conditions) mimic stringent response gene expression. This assumption is hard to validate based on the presented data. First, no data (E.g. Excell sheet) including annotations and significance of genes are provided. How are stringent genes defined? Based on previous RNAseq-Data? It is required to show the overlap of gene expression pattern (many versus pppGpp⁰ strains) in a more quantitative way to judge the assumption. One hallmark of the stringent response is the downregulation of ribosomal proteins. However, these genes might also be down-regulated via other mechanisms under starving conditions. Thus, the real (p)ppGpp dependent (stringent) genes should be evaluated by comparison with transcriptome data from WT and pppGpp⁰ strains. This data might be available from previous publication from other streptococcal species. E.g (<https://www.ncbi.nlm.nih.gov/pubmed/27255540>).

Thank you so much for the constructive comment.

We constructed the (p)ppGpp⁰ strain ($\Delta relA\Delta relQ$) by deleting the *relA* and *relQ* genes in the SEZ WT background (**Additional Data #4**). Subsequently, the RNA-seq analysis with SEZ WT, (p)ppGpp⁰ ($\Delta relA\Delta relQ$), $\Delta manY$ strains was re-performed. With increased number of biologic replicates, we were able to achieve statistical significance. Gene annotations have also been provided in the updated results (**Supplementary Table S9**).

The $\Delta relA\Delta relQ$ vs. WT group was used as a reference to define stringent response associated genes expression, thereby identifying genes that are truly regulated dependent on the (p)ppGpp. By comparing the DEGs in the 1 g/L glucose group (i.e., influenced by *manY* in low glucose) with the non-DEGs in the 10 g/L glucose group (i.e., not influenced by *manY* in high glucose) (**New Extended Data Fig.7c**), we identified 773 DEGs, 408 greater in WT (down-regulated in $\Delta manY$) and 365 greater in $\Delta manY$ (up-regulated in $\Delta manY$), that may influence bacterial growth specifically under low glucose conditions in a PTS_{man} dependent manner.

When these 773 DEGs were compared to the list of (p)ppGpp-dependent genes identified in the $\Delta relA\Delta relQ$ vs. WT comparison, 432 genes were found to overlap. Examination of the expression patterns of these 432 genes revealed that the majority displayed opposite regulation trends in the $\Delta manY$ and $\Delta relA\Delta relQ$ strains, consistent with the opposing levels of (p)ppGpp in these two mutants, elevated in $\Delta manY$ and depleted in $\Delta relA\Delta relQ$. (**Additional Data #4**) (PMID: 30916318, PMID: 32345719). We believe that the updated RNA-seq data and analysis pipeline provide support for the notion that *manY* deletion-induced DEGs under low-glucose conditions are functionally linked to the stringent response.

The data of DEGs were showed according the format of the *Streptococcus suis* serotype 2 paper (<https://www.ncbi.nlm.nih.gov/pubmed/27255540>). (**Additional Data #5, New Extended Data Fig.7e**)

Additional data #4: Construction of the (p)ppGpp⁰ ($\Delta relA\Delta relQ$), $\Delta relQ$, $\Delta manY\Delta relQ$ and $\Delta manYC\Delta relQ$ strains of SEZ and detection of their ratio of (p)ppGpp with TLC. (A, New Extended Data Fig.7b; B, New Fig.6c)

Intracellular levels of (p)ppGpp were detected by TLC after nucleotide extraction. **A**, SEZ WT, $\Delta relA\Delta relQ$ strains were grown in CDM with low (1 g/L) or high (10 g/L) concentrations of glucose. **B**, Strains were grown in CDM with 1 g/L glucose.

Additional data #5: RNA-seq analysis of SEZ WT, (*p*)ppGpp⁰ (Δ relA Δ relQ), and Δ manY strains. (New Extended Data Fig.7 exclude b and d)

A, RNA-seq analysis shows the number of DEGs between Δ manY and WT cultured in CDM with 1 and 10 g/L glucose at mid-exponential phase. The average log₂Fc = 1 was used as the threshold of DEGs. **B**, The Venn diagram illustrates the intersection of DEGs between Δ manY vs. SEZ cultured in 1 g/L glucose CDM, non-DEGs of Δ manY vs. SEZ cultured in 10 g/L glucose CDM, and DEGs of Δ relA Δ relQ vs. SEZ. Growth related genes are marked in the purple lens-shaped region (intersection of Δ manY vs. SEZ cultured in 1 g/L glucose CDM and non-DEGs of Δ manY vs. SEZ cultured in 10 g/L glucose CDM), while the (*p*)ppGpp dependent genes are shown in the green circle (Δ relA Δ relQ vs. SEZ DEGs). The central overlap among all three sets (432 genes) indicates candidates involved in both growth

regulation and (p)ppGpp. **C**, The heatmap displays the Log₂Fc of the intersection genes in panel c in $\Delta manY$ vs. WT (influenced by *manY* in low glucose) and $\Delta relA \Delta relQ$ vs. WT (influenced by (p)ppGpp) groups. The combined group consists of four distinct clusters, with the different colored squares on the right side of the heatmap representing each cluster. **D**, The boxplots indicate normalized Transcripts Per Million (TPM) of *relA* and *relQ* in SEZ and $\Delta manY$ cultured in CDM with 1 g/L (SEZ1 and $\Delta manY1$) or 10 g/L (SEZ10 and $\Delta manY10$) glucose.

(24) Line 365: The link to stringent response is mainly made by the observation that *rel* (coding for the bifunctional Rel) was upregulated in the *manY* mutant. However, Rel is a bifunctional enzyme with (p)ppGpp synthase and hydrolysis activity. Thus, the (p)ppGpp level is determined via post-translational activation. (Through interaction with the ribosome and uncharged tRNA) of the synthetase activity and concurrent inhibition of the hydrolase activity (http://www.ncbi.nlm.nih.gov/entrez/query.fcgi?cmd=Retrieve&db=PubMed&dopt=Citation&list_uids=15066282). Thus, transcriptional regulation of *rel* may lead to more or less (p)ppGpp dependent on the activation status of the enzyme.

We apologized for using the wrong gene name for the (p)ppGpp synthetase enzymes in streptococcus. In many Gammaproteobacteria species, the term *relA* is used to refer to monofunctional (p)ppGpp synthetases, whereas in Firmicutes, *relA* typically refers to bifunctional synthetase/hydrolase enzymes (PMID: 20508246). Apparently, *relA* should not be used to name streptococcal monofunctional (p)ppGpp synthetases in this study.

When we performed a BLAST search using the SESEC_RS05505 gene from SEZ (672 bp), it aligned with the SPD_RS05280 gene from *Streptococcus pneumoniae* (also 672 bp), annotated as “GTP pyrophosphokinase”. This 672 bp gene possessing only (p)ppGpp synthetase activity (SAS) and should be more accurately annotated as *relQ* (SESEC_RS05505) in streptococcus. As the reviewer correctly points out, the bifunctional Rel enzyme (RSH), typically referred to as *relA* in streptococcus, is approximately 2220 bp in length (corresponding to SESEC_RS09455 in SEZ). Accordingly, the gene we referred to as *relA* in our manuscript should have been identified as *relQ*, and the gene labeled as RSH should be corrected to *relA*. We apologize for this oversight, and to avoid misleading of the (p)ppGpp synthetase/hydrolase enzymes names, we marked SAS to *relQ* and RSH to *relA*.

We have made the necessary corrections: the primer “*relA*” has been renamed to *relQ*, and “RSH” has been renamed to *relA*. Please refer to Supplementary Table

S6: in the plasmid pSET2_PmanXYZ_relA (now corrected to pSET2-relQ), the primers 2_relA-F and 2_relA-R have been renamed 2_relQ-F and 2_relQ-R, respectively, corresponding to the 672 bp SESEC_RS05505 gene (*relQ*) of SEZ.

There was research indicated that the activity of *relQ* (SAS) was primarily regulated at the transcriptional level (PMID: 25853779).

Gene information of SPD_RS05280 (*relQ*)

```
gene      994585..995256
          /locus_tag="SPD_RS05280"
          /old_locus_tag="SPD_0982"
CDS       994585..995256
          /locus_tag="SPD_RS05280"
          /old_locus_tag="SPD_0982"
          /EC_number="2.7.6.3"
          /inference="COORDINATES: similar to AA
sequence:RefSeq:WP_000171686.1"
          /GO_function="GO:0005524 - ATP binding [Evidence IEA];
GO:0008728 - GTP diphosphokinase activity [Evidence IEA];
GO:0046872 - metal ion binding [Evidence IEA]"
          /GO_process="GO:0015969 - guanosine tetraphosphate
metabolic process [Evidence IEA]; GO:0016310 -
phosphorylation [Evidence IEA]"
          /note="Derived by automated computational analysis using
gene prediction method: Protein Homology."
          /codon_start=1
          /transl_table=11
          /product="GTP pyrophosphokinase"
          /protein_id="WP_000171686.1"
          /translation="MTLEWEEFLDPYIQAVGELKIKLRGIRKQYRKNKHSPIEFVTG
RVKPIESIEKEMARRGITYATLEHLDQDIAGLRVMVQFVDDVKEVVDILHKRQDMRII
QERYDITHRKASGYSYHVVEYTVDTINGAKTILAEIQIRTLAMNFWATIEHSLNYK
YGGDFPDEIKKRLTARIAHQLEDEMRKIREDDIQEAQLFDPLSRKLDVGVNSDDT
DEEYR"
```

Gene information of SESEC_RS05505 (*relQ*)

```
gene      complement(1157042..1157713)
          /locus_tag="SESEC_RS05505"
          /old_locus_tag="SeseC_01371"
CDS       complement(1157042..1157713)
          /locus_tag="SESEC_RS05505"
          /old_locus_tag="SeseC_01371"
          /EC_number="2.7.6.5"
          /inference="COORDINATES: similar to AA
sequence:RefSeq:WP_017771989.1"
          /GO_function="GO:0005524 - ATP binding [Evidence IEA];
GO:0008728 - GTP diphosphokinase activity [Evidence IEA];
GO:0046872 - metal ion binding [Evidence IEA]"
          /GO_process="GO:0015969 - guanosine tetraphosphate
metabolic process [Evidence IEA]; GO:0016310 -
phosphorylation [Evidence IEA]"
          /note="Derived by automated computational analysis using
gene prediction method: Protein Homology."
          /codon_start=1
          /transl_table=11
          /product="GTP pyrophosphokinase"
          /protein_id="WP_014622912.1"
          /translation="MALDWEFLDPYIQTGELKIKLRGIRKQFRKNRYSPIEFVTG
RVKSVESIEKMLRGLVLEENIAQDIQDIAGLRIMVQFVDDIEDVLSLLRQRDMTIV
YERDYIRNKKSGYSYHVVEYTVDTIEGQKVLAEIQIRTLAMNFWATIEHSLNYK
YGGDFPDEIKKRLTARIAHQLEDEMRKIREDDIQEAQLFDPLSRKLDVGVNSDDT
DELYR"
```

(25) Fig 6B shows that Rel overexpression results in growth defect. Assuming that Rel synthetase is active under low glucose conditions the data is trivial. (p)ppGpp is one of the main regulators to halt growth (inhibition of translation, replication...). However, the data does not allow direct conclusion that the growth defect in the many mutant is caused by (p)ppGpp. The growth defect can easily be explained via the decrease of glucose in the cell e.g via ATP depletion.

By constructing the $\Delta relQ$, $\Delta manY\Delta relQ$, and $\Delta manYC\Delta relQ$ strains (**Additional Data #4**), we found that deletion of the *relQ* gene significantly reduced the intracellular (p)ppGpp levels and rescued the growth defect caused by the absence of *manY* under low-glucose conditions. Conversely, reintroducing *relQ* into the $\Delta manY\Delta relQ$ background restored (p)ppGpp production and led to the reappearance of the growth defect in the $\Delta manYC\Delta relQ$ strain. These results suggest that the intracellular (p)ppGpp level is closely associated with the growth of SEZ under low-glucose conditions (**Additional Data #7**). Additionally, we measured ATP levels in WT, $\Delta manY$, and $C\Delta manY$ strains (**Additional Data #9**). Surprisingly, rather than ATP depletion, the growth-defective $\Delta manY$ strain exhibited higher ATP levels compared to WT and $C\Delta manY$ after 8 hours of growth, suggesting that its growth defect was not due to energy limitation.

Additional data #4: Construction of the (p)ppGpp⁰ ($\Delta relA\Delta relQ$), $\Delta relQ$, $\Delta manY\Delta relQ$ and $\Delta manYC\Delta relQ$ strains of SEZ and detection of their ratio of (p)ppGpp with TLC. (A, New Extended Data Fig.7b; B, New Fig.6c)

Intracellular levels of (p)ppGpp were detected by TLC after nucleotide extraction. **A**, SEZ WT, $\Delta relA\Delta relQ$ strains were grown in CDM with low (1 g/L) or high (10 g/L) concentrations of glucose. **B**, Strains were grown in CDM with 1 g/L glucose.

Additional data #7: Growth curve of WT, $\Delta manY$, $C\Delta manY$, $\Delta relA\Delta relQ$ ((p)ppGpp⁰) $\Delta relQ$, $\Delta manY\Delta relQ$, $\Delta manYC\Delta relQ$ (complement strain of $\Delta manY\Delta relQ$ with pSET2-*relQ* overexpression plasmid) and *relQ*⁺ (WT strain with pSET2-*relQ* overexpression plasmid). (B, New Fig.6b)

A, the growth curve of WT and (p)ppGpp⁰ strain. (p)ppGpp⁰ strain exhibits growth defects compared to WT SEZ. **B**, the CFU of WT, $\Delta manY$, $C\Delta manY$, $\Delta relQ$, $\Delta manY\Delta relQ$, $\Delta manYC\Delta relQ$, and *relQ*⁺ strains in CDM with 1 g/L glucose at 0, 4 and 12 h; **C**, the growth curve of WT, $\Delta manY$, $C\Delta manY$, $\Delta relQ$, $\Delta manY\Delta relQ$, $\Delta manYC\Delta relQ$, and *relQ*⁺ strains in CDM with 1 g/L glucose. Overexpression of the *relQ* gene in the WT SEZ inhibits bacterial growth, while its deletion has no effect. In the $\Delta manY$ strain, *relQ* deletion restores growth, but reintroducing *relQ* in the $\Delta manYC\Delta relQ$ strain results in growth defects once again.

Additional data #9: Intracellular ATP concentration analysis of WT, $\Delta manY$ and $C\Delta manY$ in CDM with 1 g/L glucose.

A, Bacterial growth of WT, $\Delta manY$, and $C\Delta manY$ was measured at different time points by CFU counts, Strains are grown in CDM with 1 g/L glucose. **B**, Intracellular ATP concentration in WT, $\Delta manY$, and $C\Delta manY$ grown in CDM with 1 g/L glucose, corresponding to the time points in panel A, was analyzed using the BacTiter-Glo Microbial Cell Viability Assay Kit, with ATP levels quantified by luminescence intensity.

(26) Fig 6C: Data are provided to show that glucose results in stringent response activation by direct detection of (p)ppGpp under low glucose conditions (Fig 6C). This data is not quantified. Only one TLC result is shown. Quantitative data based on at least three independent experiments should be shown.

We agree that increasing the number of replicates and performing quantitative analysis would improve the rigor and readability of the data. Accordingly, we have included additional biological replicates (**Source file #2**) and conducted gray intensity quantification (**Additional data #6, New Extended Data Fig.8**) of the TLC results.

Additional data #6: The (p)ppGpp levels were measured by TLC, and HPr and Crp were detected by western blot. Band intensities were quantified using ImageJ. (A, New Fig.6d; B, New Fig.6c; C, New Fig.6h; D, New Fig.6e. All column plots of gray intensity analysis are displayed in New Extended Data Fig.8)

A-C. Intracellular levels of (p)ppGpp detected by TLC plates. The relative intensity of (p)ppGpp is calculated as $(ppGpp + pppGpp) / (GTP + ppGpp + pppGpp)$. The bubble plots below each panel represent the gray intensity corresponding to the results. **D.** Gray intensity analysis of Crp expression levels and the percentage of P-HPr. Crp expression level is calculated as the ratio of Crp to GroEL. The percentage of P-HPr is calculated as the ratio of P-HPr to total HPr. The bubble plots below each panel represent the gray intensity corresponding to the results.

The column plots on the right of each panel present the replicated gray intensity results. Statistical significance of A and D was analyzed with two-way ANOVA followed by Dunnett's multiple comparisons test. Statistical significance of B and C was analyzed with one-way ANOVA followed by Dunnett's multiple comparisons test.

(27) Figure 6E: The meaning of the schematic figure is not very clear. What would be the role of ManY in this context

As suggested, we have updated the schematic figure and highlighted the role of the *manY* gene in the process.

Reviewer #5 (Remarks to the Author):

The manuscript by Yuan et al. describes a mechanism behind how pathogenic *Streptococcus equi zooepidemicus* (SEZ) can replicate to high levels in CSF. Using a genetic barcoding approach, the authors reveal that the CSF is colonized by a small founding population (FP) following bloodstream infection. Once in the CSF, that small FP replicates to high levels. While other pathogenic streptococci can access the CSF, they don't replicate to as high of bacterial burden. The ability to replicate to these high levels, in part, is because of the unique increased expression of the PTSman system. The increased expression results from specific polymorphisms in the PTSman promoter. The authors finish by demonstrating this increased production of the PTSman system allows for increased replication in minimal nutrient conditions with low glucose, like those found in the CSF. Increased glucose import by the PTSman system provides enough nutrients for the cell to prevent activation of the stringent response. I found the manuscript well written, the figures easy to interpret, and the experiments thorough.

After reviewing the document and rebuttal to prior reviews, I feel the authors have addressed all of the previous reviewer's concerns. I only have minor comments that need to be addressed.

Thank you so much for your positive feedback on this paper.

(28) Line 197 – “genes that were linked to metabolism (marked with asterisks...)” – I didn't see any asterisks in the figure.

We have repositioned the asterisks to the left side of the figure to improve visibility and make them easier to locate.

(29) Line 203 – This sentence refers to 24hr CFU counts of the $\Delta proB$ and $\Delta proC$ strains and references Fig1a and Fig. S4b. While the statement made in the sentence may be true, Fig. S4b is looking 14 days post infection, not 24 hr. Fig. 1a only shows WT data at 24 hr

We have added the blood CFU data for the \$\Delta proB\$ and \$\Delta proC\$ strains at 24 hours post-infection in the **New Extended Data Fig.4b (Additional data #8)**. These strains exhibited markedly lower CFU levels compared to the WT at the same time point. Furthermore, both \$\Delta proB\$ and \$\Delta proC\$ were completely cleared from the blood by day 14, indicating impaired survival in the bloodstream. Consistent with the survival curve, these results suggest that both mutants have completely lost virulence.

Additional data #8: The blood CFU counts of the $\Delta proB$ and $\Delta proC$ strains at 24 hours post-infection in the new Extended Data Fig.4b.

C57BL/6J mice were intravenously injected with 5×10^6 CFU of either $\Delta proB$ or $\Delta proC$, and bacterial counts in the blood were measured.

(30) Line 235 – This sentence refers to bacterial burden outside the CNS upon antibiotic treatment. I did not see any data where bacterial burdens were determined with antibiotic treatment, only survival curves. I am also confused as to the greater survival of the $\Delta manY$ mutant with penicillin treatment compared to WT. The authors state that both WT and the $\Delta manY$ mutant are equally susceptible to penicillin (line 235). If without penicillin treatment, both WT and $\Delta manY$ burdens are the same outside the CNS (Fig 3C), and both cause the same mortality... then why is there a difference in survival with penicillin treatment? Do the authors think this difference in mortality is due to likely differences in bacterial burden in the brain (as these are different to begin with in the absence of antibiotics)? If not, it is unclear how the sentence beginning in line 236 regarding the benefits of antibiotic treatment being enhanced by targeting nutrient uptake systems can be true. This should be clarified.

We thank the reviewer for this insightful comment. We have added CFU burden data from mice administered 200 μ g of penicillin at 12-hour intervals following infection with either the WT or $\Delta manY$ strain (**Additional Data #1**). Analysis of brain CFU revealed that surviving mice challenged with $\Delta manY$ exhibited low bacterial burdens in the brain after bacteremia was controlled by antibiotic treatment. In contrast, although blood CFUs dropped to undetectable levels in WT-infected mice following penicillin administration, all mice ultimately succumbed to infection due to persistently high bacterial loads in the brain. These findings suggest that while penicillin effectively clears bacteria from the bloodstream, additional targeting of the PTS_{man} system limits bacterial proliferation in the brain and significantly improves survival.

Additional Data #1: Survival curves and CFU of WT or $\Delta manY$ -infected mice administered penicillin 24 hours post infecton. (A, New Fig.3f; B, New Fig.3g)

A, Survival curve of C57BL/6J mice after intravenous administration of 200 μ g penicillin every 12 hours. Mice were challenged i.v. with 5×10^6 CFU WT or $\Delta manY$, and subsequently given penicillin or vehicle (PBS) treatment starting at 24 hpi (Simple survival analysis following a log-rank test). **B**, CFU burden in the organs and blood of moribund and surviving mice post infection and penicillin treatment.

(31) Line 253 should be Fig. S4c not Fig. 4c

(32) Line 397 should be "promoter"

Thank you very much for your thorough review. We have revised the manuscript accordingly.

Response to Reviewers:

Reviewer #1 (Remarks to the Author). Author responses are in blue.

It is clear that, with a cleared bacteremia, there is a significant (two logs) of bacterial load in the brain between the wt and the mutant. Yet is this due to better bacterial growth of the wt in the brain, or to a better penetration capabilities across the BBB?

-- Authors should grow the two strains (wt, mutant) using brain tissue homogenate as culture medium and assess bacterial growth over time.

-- Additionally, if wt and mutant are administered onto brain endothelial cells in vitro, what strain is capable of adhering more/better to the cells?

These are quick experiments to be performed and would represent an important evidence in support (or not) of the conclusions drawn so far by the authors. I propose one last minor revision.

Response: We sincerely thank the reviewer for the constructive suggestions and for thoughtfully proposing additional experiments. We fully agree that assessing bacterial growth in brain tissue homogenates and adhesion to brain endothelial cells in vitro would provide further mechanistic insights, and we will pursue these important directions in our future work. At present, we believe our barcoding data already address the central concern, the increased bacterial loads observed in the brain are not primarily due to enhanced translocation across the BBB, but rather to extensive proliferation within the cerebrospinal fluid. This was precisely the rationale behind the design of our barcoding experiments, and the results strongly support this conclusion.

Reviewer #4 (Remarks to the Author): Author responses are in blue.

The authors now provided more convincing evidence regarding the stringent response. I only suggest to tone down the conclusion provided in the summary figure. There are no in vivo data using a pppGpp mutant. Thus, it remains unproven whether the circuit is working in vivo.

Response: We thank the reviewer for the positive assessment of our results on the stringent response and for the helpful suggestion. Conducting in vivo experiments with the (p)ppGpp mutant strain poses technical challenges, particularly due to the difficulty in detecting intracellular (p)ppGpp levels within animal models. In line with the reviewer's advice, we have toned down the conclusion in the summary figure. Specifically, we have clarified which findings are derived from in vitro experiments and have avoided overextending the mechanistic implications to the in vivo context. We hope these revisions improve the rigor of our manuscript.

Minor: Figr 8E seems to miss the description in the legend.

Response: We thank the reviewer for pointing this out. In the original version, the legend for Extended Data Figure 8e was embedded in the same paragraph as panels 8a and 8b, which made it less noticeable. In the revised manuscript, we have separated the description of panel 8e into its own entry to improve clarity.